# CBraMod: A Criss-Cross Brain Foundation Model for EEG Decoding

**Jiquan Wang**[1,2], **Sha Zhao**[1,2,*] **Zhiling Luo**[3], **Yangxuan Zhou**[1,2], **Haiteng Jiang**[4,5,1],
**Shijian Li**[1,2], **Tao Li**[4,5,1], **Gang Pan**[1,2,5*]

[1]State Key Laboratory of Brain-machine Intelligence, Zhejiang University
[2]College of Computer Science and Technology, Zhejiang University
[3]Alibaba Group
[4]Department of Neurobiology, Affiliated Mental Health Center & Hangzhou
Seventh People's Hospital, Zhejiang University School of Medicine
[5]MOE Frontier Science Center for Brain Science and Brain-machine Integration,
Zhejiang University
```
{wangjiquan, szhao, zyangxuan, h.jiang}@zju.edu.cn;
{shijianli, litaozjusc, gpan}@zju.edu.cn;
godot.lzl@alibaba-inc.com
```

## ABSTRACT

Electroencephalography (EEG) is a non-invasive technique to measure and record brain electrical activity, widely used in various BCI and healthcare applications. Early EEG decoding methods rely on supervised learning, limited by specific tasks and datasets, hindering model performance and generalizability. With the success of large language models, there is a growing body of studies focusing on EEG foundation models. However, these studies still leave challenges: Firstly, most of existing EEG foundation models employ full EEG modeling strategy. It models the spatial and temporal dependencies between all EEG patches together, but ignores that the spatial and temporal dependencies are heterogeneous due to the unique structural characteristics of EEG signals. Secondly, existing EEG foundation models have limited generalizability on a wide range of downstream BCI tasks due to varying formats of EEG data, making it challenging to adapt to. To address these challenges, we propose a novel foundation model called CBraMod. Specifically, we devise a criss-cross transformer as the backbone to thoroughly leverage the structural characteristics of EEG signals, which can model spatial and temporal dependencies separately through two parallel attention mechanisms. And we utilize an asymmetric conditional positional encoding scheme which can encode positional information of EEG patches and be easily adapted to the EEG with diverse formats. CBraMod is pre-trained on a very large corpus of EEG through patch-based masked EEG reconstruction. We evaluate CBraMod on up to 10 downstream BCI tasks (12 public datasets). CBraMod achieves the state-of-the-art performance across the wide range of tasks, proving its strong capability and generalizability. The source code is publicly available at https://github.com/wjq-learning/CBraMod.

## 1 INTRODUCTION

Brain-Computer Interface (BCI) refers to a system that allows direct communication between the brain and external devices or computers (Schalk et al., 2004; Wu et al., 2013; Zhang et al., 2019b). Electroencephalography (EEG) is a technique used to measure and record electrical activity in the brain, where electrodes are placed on the scalp to detect and amplify the brain's electrical signals. EEG plays a crucial role in BCI as it provides a non-invasive and real-time measure of brain activity that can be used to decode and interpret user's intentions or commands. By analyzing the patterns and features in EEG signals, algorithms and models have been developed to decode specific brain

---

[*]Corresponding authors: Sha Zhao and Gang Pan.

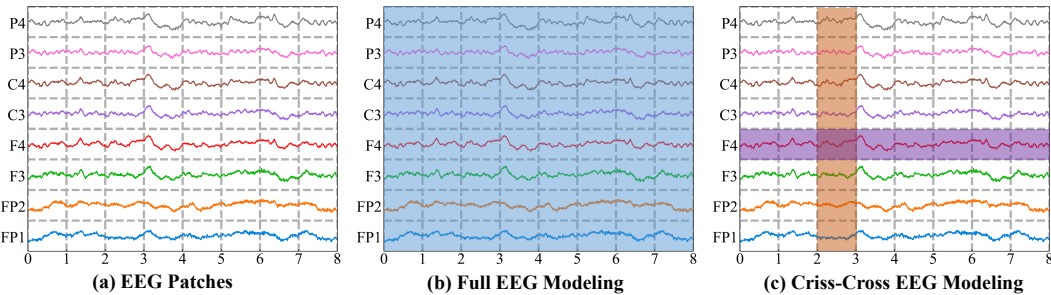

Figure 1: EEG patches and different EEG modeling strategies.

states, including but not limited to emotion recognition (Dadebayev et al., 2022; Gao et al., 2024), motor imagery classification (Altaheri et al., 2021; Dai et al., 2020), seizure detection (Ahmad et al., 2022; Yıldız et al., 2022) and sleep staging (Phan & Mikkelsen, 2022; Wang et al., 2024b; Zhou et al., 2024a).

Early studies on EEG decoding predominantly employed traditional machine learning methods (Lotte et al., 2007). With the rapid advancement of deep learning (LeCun et al., 2015), various deep neural networks have been developed to decode EEG signals (Craik et al., 2019) and perform downstream BCI tasks (Pan et al., 2018; Sekkal et al., 2022). These deep learning-based EEG models include Convolution Neural Network (CNN) (Lawhern et al., 2018; Ding et al., 2022), Long Short-Term Memory (LSTM) (Wang et al., 2018; Phan et al., 2019), CNN-LSTM (Zhang et al., 2019a; Wang et al., 2023a), Transformer (Song et al., 2021; Phan et al., 2022), CNN-Transformer (Song et al., 2022; Peh et al., 2022), Graph Neural Networks (GNN) (Jia et al., 2020; Ding et al., 2023) and etc. However, most of deep learning models employ supervised learning methods tailored for specific tasks or datasets, and lack generalization ability. There are still significant challenges in improving the generalizability and performance due to many factors, such as limited data amounts and substantial differences in EEG signal formats. The EEG is usually collected for specific BCI tasks, and the amount is relatively small, because collecting and labeling EEG is expensive and time-consuming. Meanwhile, the EEG signal formats significantly differ across different datasets, such as channel configurations and time lengths.

Impressed by self-supervised learning (SSL) (Liu et al., 2021c) on computer vision (CV) and natural language processing (NLP), some studies (Kostas et al., 2021; Chien et al., 2022; Yang et al., 2023; Foumani et al., 2024; Jiang et al., 2024; Wang et al., 2024a) propose EEG foundation models which are pre-trained on a vast amount of EEG data with self-supervised learning and fine-tuned on downstream datasets for some clinical or BCI applications. Most of these methods segment the original EEG signals into patches of EEG channels to ensure that the neural network can deal with EEG signals with diverse channels and time length, shown in Figure 1(a). Consistent with the approach of handling image patches used in ViT (Dosovitskiy et al., 2020), they flatten the EEG patches and feed them into transformers to model the dependencies among all EEG patches, called full EEG modeling strategy shown in Figure 1(b). Meantime, they adopted absolute positional encoding based on electrode numbering as a channel embedding to encode positional information. However, existing methods ignore two characteristics of EEG signals: **Unique structural characteristics**: EEG signals have different structural characteristics compared to image. It contain heterogeneous spatial and temporal dependencies, but images contain only spatial dependencies. And the dependencies among EEG patches within the same channel or time interval could be stronger than those among patches from different channels and time intervals, but such a prior assumption may not be valid in image. The full EEG modeling strategy models the dependencies among all EEG patches together, ignoring the unique structural characteristics of EEG signals. **Channel variation**: EEG channels are not solely defined by electrode position but are also influenced by the referencing scheme used (e.g., earlobe, average, REST, or bipolar references). Existing EEG foundation models, such as LaBraM (Jiang et al., 2024), employ absolute positional encoding based on electrode numbering as a channel embedding, but this method assumes a fixed relationship between EEG channels and electrode positions. As a result, absolute positional encodings tied directly to electrode numbering may limit the model's adaptability across tasks and datasets that vary in spatial and reference properties.

To address these issues, we first propose a criss-cross EEG modeling strategy to thoroughly leverage the structural characteristics of EEG signals, inspired by some studies (Huang et al., 2019; Dong et al., 2022) which adopt similar strategies on vision modeling. Based on this strategy, we then propose an EEG foundation model, which can model spatial and temporal dependencies in parallel, as shown in Figure 1(c). Meantime, we propose asymmetric conditional positional encoding (ACPE) as a more flexible approach to positional encoding. ACPE employs a convolutional network to dynamically learn spatial relationships among patches, allowing the model to capture relative positional information from various channel format. This dynamic position learning enhances the model's adaptability, making it better suited for downstream tasks with varying spatial configurations and reference contexts. Our detailed contributions are as follows:

- We propose **a novel EEG foundation model**, called **CBraMod**, for EEG decoding on various clinical and BCI application. CBraMod is pre-trained on **Temple University Hospital EEG Corpus (TUEG)** for learning generic representations from both **time-domain** and **frequent-domain** EEG signals through patch-based masked EEG reconstruction. To the best of our knowledge, TUEG is the largest public EEG corpus so far.

- For modeling effective **spatial-temporal dependencies** among EEG patches and adapting to various downstream datasets, we firstly devise a **criss-cross transformer**, based on **criss-cross EEG modeling** strategy, as the backbone of CBraMod. It models spatial and temporal dependencies separately through two parallel attention mechanisms, spatial and temporal attentions. Meanwhile, we utilize an **asymmetric conditional positional encoding (ACPE)** scheme to encode positional information. ACPE can dynamically generate positional encoding, which can well be adapted to **arbitrary EEG formats** of different downstream datasets.

- We evaluate the performance of CBraMod on up to **10 downstream BCI tasks** using **12 public datasets**. The majority of downstream datasets are sourced from institutions different from the pretraining dataset. The experimental results demonstrate that CBraMod achieves state-of-the-art performance across all the tasks, highlighting its successful **generalizability**. To the best of our knowledge, this is the first study to comprehensively evaluate EEG foundation models **across such a broad range of downstream BCI tasks**.

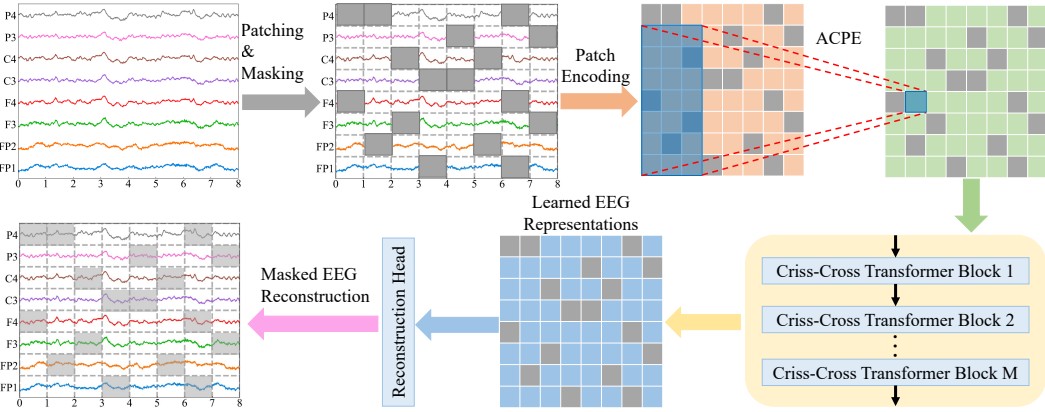

Figure 2: CBraMod pre-training overview.

## 2 METHOD

The pre-training framework of CBraMod is shown in Figure 2. Firstly, we segment EEG samples into patches using a fixed time window and randomly mask some patches with a mask token. Next, each EEG patch is fed into a patch encoding network to obtain corresponding patch embedding. Spatial-temporal positional embeddings are obtained through an asymmetric conditional positional encoding (ACPE) scheme and added to the patch embeddings. Then, the patch embeddings are fed into criss-cross transformer blocks with criss-cross attention mechanism to learn EEG representations. Finally, a reconstruction head is utilized to reconstruct the masked EEG patches from the learned representations. We will provide a detailed description of CBraMod.

**Patching & Masking**  We denote an EEG sample as $S \in \mathbb{R}^{C \times T}$, where $C$ is the number of electrode channels and $T$ is the number of timestamps. In order to ensure that the neural network can deal with EEG signals with diverse channels and time length, we **segment the original EEG sample into patches of EEG channels through a fixed-length time window**. Specifically, we divide $S$ with time window length $t$ to get a set of patches $X \in \mathbb{R}^{C \times n \times t}$, where $n = \lfloor \frac{T}{t} \rfloor$ is the number of patches in each channel, or sequence length. Thus an EEG patch can be denoted as $x \in \mathbb{R}^t$ and we can get $X = \{x_{i,j} | i \in [1, 2, ..., C], j \in [1, 2, ..., n]\}$. The total number of patches in $X$ is $|X| = Cn$. After EEG patching, we randomly generate a **mask** $\mathcal{M} = \{m_{i,j} | i \in [1, 2, ..., C], j \in [1, 2, ..., n]\}$ from a Bernoulli distribution of $r$ proportion, where $m_{i,j} \in \{0, 1\}$ denotes the mask indicator of $x_{i,j}$. Then we mask the EEG patches of $X$ by replacing the EEG patch $x_{i,j}$ with $x_M$ as follows:

$$\tilde{x}_{i,j} = \begin{cases} x_{i,j}, & m_{i,j} = 0 \\ x_M, & m_{i,j} = 1 \end{cases} \tag{1}$$

$$\tilde{X} = \{\tilde{x}_{i,j} | i \in [1, 2, ..., C], j \in [1, 2, ..., n]\} \tag{2}$$

where $x_M \in \mathbb{R}^t$ is the mask token and $\tilde{X} \in \mathbb{R}^{C \times n \times t}$ is the set of all EEG patches after masking.

**Time-Frequency Patch Encoding**  In order to extract the local features from each patch $\tilde{x}_{i,j}$ of $\tilde{X}$, we devise a patch encoder applied to each patch. Our patch encoder consists of two branches, **time-domain branch** and **frequency-domain branch**. The **time-domain branch** comprises multiple convolution blocks designed to extract time-domain features within each patch. A convolution block consists of an one-dimensional convolution layer, a group normalization layer, and a GELU activation function. We feed each EEG patch $\tilde{x}_{i,j}$ into the time-domain branch to obtain the time-domain embedding $e_{i,j}^t \in \mathbb{R}^d$, where $d$ is the dimension of the embedding. The **frequency-domain branch** comprises fast Fourier transform (FFT) and a fully-connected layer to extract frequency-domain features from each patch. Specifically, we leverage FFT to extract an energy vector for each patch $\tilde{x}_{i,j}$ where every dimension indicates the energy of a specific frequency. Then we feed the energy vector into a fully-connected layer to get frequency-domain embedding $e_{i,j}^f \in \mathbb{R}^d$. Afterwards, we add each time-domain embedding and the corresponding frequency-domain embedding as follows:

$$e_{i,j} = e_{i,j}^t + e_{i,j}^f \tag{3}$$

$$E = \{e_{i,j} | i \in [1, 2, ..., C], j \in [1, 2, ..., n]\} \tag{4}$$

where $e_{i,j} \in \mathbb{R}^d$ represents the patch embedding, $E \in \mathbb{R}^{C \times n \times d}$ is the set of patch embeddings.

**Asymmetric Conditional Positional Encoding**  Differing from existing EEG foundation models that primarily utilize an absolute positional encoding (APE) scheme, we employ an asymmetric conditional positional encoding (ACPE) scheme to encode spatial and temporal positional information of EEG patches. This is a modification upon conditional positional encoding (CPE) (Chu et al., 2021) that is applied to encode positional information of image patches. Compared to APE, ACPE can dynamically encode the temporal and spatial positional information of each EEG patch, improving adaptation to diverse channel configurations and time lengths. Compared to CPE, ACPE is designed in an asymmetric fashion to encode short-range temporal positional information and long-range spatial positional information. It is because EEG signals are multi-channels sequential neural signals, requiring the model to encode different-range positional information in spatial and temporal dimension.

Specifically, we devise a **convolution layer** as **positional encoder** to dynamically generate ACPE from the spatial-temporal neighborhoods of an EEG patch, shown in Figure 2. The convolution network is composed of a depthwise two-dimension convolution layer with kernel $(k_s, k_t)$ and $(\frac{k_s - 1}{2}, \frac{k_t - 1}{2})$ zero paddings, where $k_s$ is the kernel size of spatial (channel) dimension and $k_t$ is the kernel size of temporal dimension. Differing from CPE, we utilize an **asymmetric convolution kernel** $(k_s > k_t)$ to encode spatial-temporal positional information. The longer side in spatial dimension is used to encode longer-range spatial positional information, and the shorter in temporal dimension is used to encode shorter-range temporal positional information. We feed patch embeddings $E$ into the positional encoder to generate the ACPE $E^p = \{e_{i,j}^p | i \in [1, 2, ..., C], j \in [1, 2, ..., n]\}$, where $E^p \in \mathbb{R}^{C \times n \times d}$ and $e_{i,j}^p \in \mathbb{R}^d$. Then we add ACPE to patch embeddings:

$$E^o = E + E^p = \{e_{i,j} + e_{i,j}^p | i \in [1, 2, ..., C], j \in [1, 2, ..., n]\} \tag{5}$$

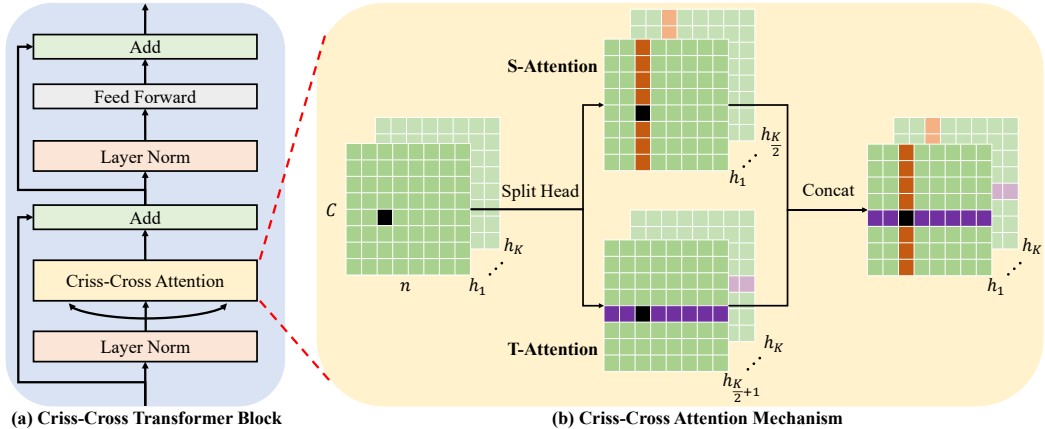

Figure 3: Criss-Cross Transformer Block.

where $E^o \in \mathbb{R}^{C \times n \times d}$ is the set of EEG patch embeddings with positional information.

**Criss-Cross Transformer** We propose criss-cross transformer to capture the heterogeneous spatial and temporal dependencies among EEG patches, whose architecture is shown in Figure 3.

*Criss-Cross Transformer Block.* As shown in Figure 3(a), a criss-cross transformer block consists of layer normalization layer, criss-cross attention, addition component, and feed forward layer. In order to ensure the stability and efficiency of the transformer training process, we utilize the pre-norm strategy. We incorporate layer normalization to the queries and keys prior to the attention mechanism to prevent the occurrence of excessively large values in attention logits. This approach helps to maintain more consistent gradients across layers, resulting in better convergence during the training process (Xiong et al., 2020; Dehghani et al., 2023). Here, we feed EEG patch embeddings $E^o$ into layer normalization to get normalized patch embedding $\tilde{E} \in \mathbb{R}^{C \times n \times d}$. Then we feed $\tilde{E}$ into the criss-cross attention to capture spatial-temporal dependencies among EEG patches.

*Criss-Cross Attention Mechanism.* The **criss-cross attention** is shown in Figure 3(b). It consists of parallel **spatial attention (S-Attention)** and **temporal attention (T-Attention)**. The S-Attention is used to capture spatial dependencies among EEG patches within the same time interval and the T-Attention is used to capture temporal dependencies among EEG patches within the same channel. Based on the multi-head self-attention mechanism, the input embeddings $\tilde{E} \in \mathbb{R}^{C \times n \times d}$ will undergo an initial linear projection to $K$ heads, and each head will subsequently apply either S-Attention or T-Attention. For S-Attention, we can partition $\tilde{E}$ into $n$ spatial stripes as $\tilde{E} = [\tilde{E}^1, \tilde{E}^2, ...\tilde{E}^n]$, where $\tilde{E}^1, ..., \tilde{E}^n \in \mathbb{R}^{C \times d}$. We suppose the projected queries, keys and values of the $k$-th head all have dimension $d_k$, then the process of S-Attention for the $k$-th head is as follows:

$$F_k^j = \text{Attention}(\tilde{E}^j W_k^Q, \tilde{E}^j W_k^K, \tilde{E}^j W_k^V) \qquad (6)$$

$$\text{S-Attention}_k(\tilde{E}) = [F_k^1, F_k^2, ..., F_k^n] \qquad (7)$$

where $j \in [1, 2, ..., n]$ represents the $j$-th spatial stripes, $W_k^Q, W_k^K, W_k^V \in \mathbb{R}^{d \times d_k}$ is the linear projection matrices of queries, keys and values for the $k$-th head respectively, and $d_k = d/K$. The process of T-Attention closely resembles that of S-Attention. Then we concatenate the outputs of S-Attention and T-Attention as follows:

$$\text{Criss-Cross-Attention}(\tilde{E}) = \text{Concat}(\text{head}_1, \text{head}_2, ..., \text{head}_K) \qquad (8)$$

$$\text{head}_k = \begin{cases} \text{S-Attention}_k(\tilde{E}), & k \in [1, 2, ..., K/2] \\ \text{T-Attention}_k(\tilde{E}), & k \in [K/2 + 1, K/2 + 2, ..., K] \end{cases} \qquad (9)$$

where $\text{head}_k$ represents the output of the $k$-th head.

There are $M$ criss-cross transformer blocks in CBraMod. The output of criss-cross transformer is denoted as $E^r = \{e_{i,j}^r | i \in [1, 2, ..., C], j \in [1, 2, ..., n]\}$, where $e_{i,j}^r \in \mathbb{R}^d$ is an EEG patch representation, and $E^r \in \mathbb{R}^{C \times n \times d}$ is the set of EEG patch representations.

**Masked EEG Reconstruction** In order to learn powerful generic representations from unlabeled EEG data, we need to utilize the task of self-supervised representation learning to pre-train CBraMod. Masked autoencoder (MAE) has been proved to be a simple and effective self-supervised pre-training approach in the fields of natural language understanding, computer vision, and time series forecasting (Devlin et al., 2018; He et al., 2022; Nie et al., 2022). We utilize **patch-based masked EEG reconstruction** to learn generic representation from EEG. Specifically, a reconstruction head, composed of a fully-connected layer, is used to project learned EEG representations $E^r$ into predicted EEG patches $\hat{X} = \{\hat{x}_{i,j} | i \in [1, 2, ..., C], j \in [1, 2, ..., n]\}$, where $\hat{x}_{i,j} \in \mathbb{R}^t$ is the predicted EEG patch corresponding to one original EEG patch $x_{i,j}$, and $\hat{X} \in \mathbb{R}^{C \times n \times t}$ is the set of predicted EEG patches corresponding to the set of original EEG patches $X$.

Consistent with most of existing MAE studies(He et al., 2022; Xie et al., 2022), we only reconstruct the masked EEG patches. Given the mentioned mask $\mathcal{M} = \{m_{i,j} | i \in [1, 2, ..., C], j \in [1, 2, ..., n]\}$, we get the set of the masked predicted EEG patches $\hat{X}^M$ and the set of the masked original EEG patches $X^M$ as follows:

$$\hat{X}^M = \{\hat{x}_{i,j} | m_{i,j} = 1, i \in [1, 2, ..., C], j \in [1, 2, ..., n]\} \tag{10}$$

$$X^M = \{x_{i,j} | m_{i,j} = 1, i \in [1, 2, ..., C], j \in [1, 2, ..., n]\} \tag{11}$$

where $m_{i,j} \in \{0, 1\}$ denotes the mask indicator of $x_{i,j}$.

Finally, we use the **mean square error (MSE)** as reconstruction loss function:

$$\mathcal{L} = \|\hat{X}^M - X^M\|^2 \tag{12}$$

## 3 EXPERIMENTS

### 3.1 PRE-TRAINING

**Pre-training Dataset** CBraMod is pre-trained on a very large public dataset, Temple University Hospital EEG corpus (TUEG) (Obeid & Picone, 2016). The TUEG dataset consists of a diverse archive of 69,652 clinical EEG recordings from 14,987 subjects across 26,846 sessions, with a total duration of 27,062 hours. The archive has over 40 different channel configurations and varying duration of recordings. Most of the recordings are sampled at 256 Hz. Unfortunately, the TUEG dataset suffers from significant data contamination, including a substantial amount of unmarked noise, artifacts, and faulty channels. Manual removal of these interference factors presents considerable challenges. Thus we utilize a range of automated techniques to preprocess the data.

**Preprocessing** Firstly, the recordings with a total duration of no more than 5 minutes are removed, then the first one minute and last one minute of each recording are also discarded, so that we can remove low-quality data as much as possible. Next, we select 19 common EEG channels (Fp1, Fp2, F7, F3, Fz, F4, F8, T3, C3, Cz, C4, T4, T5, P3, Pz, P4, T6, O1, O2) that meet a subset of the 10-20 international electrode placement system standards to obtain clean and uniformly formatted pre-training data. Then a band-pass filtered (0.3 Hz–75 Hz) is applied to remove the low-frequency and high-frequency noise. A notch filter (60 Hz) is used to remove the power line noise. All EEG signals **are resampled to 200Hz** and **segmented to 30-second non-overlapping EEG samples**. However, the aforementioned preprocessing still cannot completely solve the quality issues of EEG data. Therefore, we adopt a simple and automated EEG bad sample removal scheme to obtain clean pre-training EEG data. Specifically, we regard EEG samples as bad samples in which any data point has an absolute amplitude exceeding 100 μV, and remove them from the dataset. Finally, we normalize EEG by setting the unit to 100 μV to guarantee the value mainly between -1 to 1, consistent with LaBraM (Jiang et al., 2024). To sum up, there are 1,109,545 EEG samples retained for pre-training and longer than 9000 hours in total, significantly exceeding the amount of EEG samples used for pretraining LaBraM (2,534.78hours) (Jiang et al., 2024).

**Pre-training Settings** We implemented CBraMod based on the Python 3.11.7 and PyTorch 2.1.2 + CUDA 12.1. **We set the time duration of each EEG patch as 1 second (200 data points)**, and one 30-second EEG sample is segmented to $19 \times 30 = 570$ EEG patches. For the model configurations, the patch encoder consists of 3-layer 1D convolution with group normalization and GELU activation. The positional encoder is an one-layer 2D depthwise CNN. **The backbone of CBraMod is a 12-layer**

**criss-cross transformer with 200 hidden dimensions, 800 inner dimensions (feed-forward), and 8-head criss-cross attention (4 heads for S-Attention and 4 heads for V-Attention).** 50% of mask ratio is used to randomly mask the patches. The batch size was set to 128 and the number of epochs was set to 40. The model is trained using the AdamW optimizer with default settings, the learning rate is set to 5e-4 and the weight decay is set to 5e-2. CosineAnnealingLR was used to dynamically adjust the learning rate during the pre-training. CBraMod was pre-trained on one machine with Intel Xeon Gold 6226R CPU and four NVIDIA RTX A5000 GPU for about 5 days. More details can be found in Appendix B.

## 3.2 Experiment Setup of Downstream BCI Tasks

**Downstream BCI Tasks and Datasets**   To comprehensively evaluate the performance of our method, we select up to 10 downstream BCI tasks. All the downstream BCI tasks with the corresponding datasets are presented in Table 1 and Appendix D.1. For all the downstream datasets, we **resampled the EEG signals into 200 Hz** and **set the time duration of each EEG patch as 1 second (200 data points)**, consistent with the pre-training data. More details of each dataset and its preprocessing are introduced in Section 3.3 and Appendix E.

Table 1: Overview of downstream BCI tasks and datasets.

| BCI Tasks | Datasets | Rate | # Channels | Duration | # Samples | Label |
|---|---|---|---|---|---|---|
| I. Emotion Recognition | FACED | 250Hz | 32 | 10s | 10,332 | 9-class |
|  | SEED-V | 1000Hz | 62 | 1s | 117,744 | 5-class |
| II. Motor Imagery Classification | PhysioNet-MI | 160Hz | 64 | 4s | 9,837 | 4-class |
|  | SHU-MI | 250Hz | 32 | 4s | 11,988 | 2-class |
| III. Sleep Staging | ISRUC | 200Hz | 6 | 30s | 89,240 | 5-class |
| IV. Seizure Detection | CHB-MIT | 256Hz | 16 | 10s | 326,993 | 2-class |
| V. Imagined Speech Classification | BCIC2020-3 | 256Hz | 64 | 3s | 6,000 | 5-class |
| VI. Mental Disorder Diagnosis | Mumtaz2016 | 256Hz | 19 | 5s | 7,143 | 2-class |
| VII. Vigilance Estimation | SEED-VIG | 200Hz | 17 | 8s | 20,355 | regression |
| VIII. Mental Stress Detection | MentalArithmetic | 500Hz | 20 | 5s | 1,707 | 2-class |
| IX. Event Type Classification | TUEV | 250Hz | 16 | 5s | 112,491 | 6-class |
| X. Abnormal Detection | TUAB | 250Hz | 16 | 10s | 409,455 | 2-class |

**Baselines**   We compare CBraMod with both non-foundation-model and foundation-model baseline on all the downstream BCI tasks, for comprehensive evaluation. We adopt the following methods as non-foundation-model baselines: **EEGNet** (Lawhern et al., 2018), **EEGConformer** (Song et al., 2022), **SPaRCNet** (Jing et al., 2023), **ContraWR** (Yang et al., 2021), **CNN-Transformer** (Peh et al., 2022), **FFCL** (Li et al., 2022), and **ST-Transformer** (Song et al., 2021). We re-implemented above baselines based on the public code provided by BIOT (Yang et al., 2023) unless their experimental results have already been reported in existing studies. Besides, we use **BIOT** (Yang et al., 2023) and **LaBraM** (Jiang et al., 2024) as the foundation-model baselines. We fine-tune BIOT and LaBraM based on their public code and pre-trained weights, unless their experimental results have already been reported in the original papers. Notably, LaBraM has three different configurations, LaBraM-Base, LaBraM-Large and LaBraM-Huge, but **only the pre-trained weights of LaBraM-Base are opened publicly**. We only fine-tune the LaBraM-Base in our experiments.

**Metrics**   We adopt **Balanced Accuracy**, **AUC-PR** and **AUROC** as evaluation metrics for *binary classification*, where AUROC is set as the monitor score. For *multi-class classification*, we use **Balanced Accuracy**, **Cohen's Kappa** and **Weighted F1** for evaluation, where Cohen's Kappa is set as the monitor score. **Pearson's Correlation**, **R2 Score** and **RMSE** are used as evaluation metrics for *regression*, where R2 score is utilized as the monitor score. We obtain all the results with five different random seeds and report the mean and standard deviation values. Appendix D shows more.

## 3.3 Results

For proving the capability and generalizability of CBraMod, we evaluate CBraMod and baselines on up to 10 downstream BCI tasks using 12 publicly available datasets, listed in Table 1. In all the experiments, **we ensure strict consistency in the splits of training, validation, and test sets for every method**. In this section, we show the experimental results on downstream BCI tasks of emotion recognition and motor imagery classification on 4 datasets. More results are shown in Appendix E.

**Performance Comparison with Baselines**   We compare CBraMod with the baselines as follows:

Table 2: The results of different methods on emotion recognition.

| | FACED, 9-class | | | SEED-V, 5-class | | |
|---|---|---|---|---|---|---|
| Methods | Balanced Accuracy | Cohen's Kappa | Weighted F1 | Balanced Accuracy | Cohen's Kappa | Weighted F1 |
| EEGNet | 0.4090 ± 0.0122 | 0.3342 ± 0.0251 | 0.4124 ± 0.0141 | 0.2961 ± 0.0102 | 0.1006 ± 0.0143 | 0.2749 ± 0.0098 |
| EEGConformer | 0.4559 ± 0.0125 | 0.3858 ± 0.0186 | 0.4514 ± 0.0107 | 0.3537 ± 0.0112 | 0.1772 ± 0.0174 | 0.3487 ± 0.0136 |
| SPaRCNet | 0.4673 ± 0.0155 | 0.3978 ± 0.0289 | 0.4729 ± 0.0133 | 0.2949 ± 0.0078 | 0.1121 ± 0.0139 | 0.2979 ± 0.0083 |
| ContraWR | 0.4887 ± 0.0078 | 0.4231 ± 0.0151 | 0.4884 ± 0.0074 | 0.3546 ± 0.0105 | 0.1905 ± 0.0188 | 0.3544 ± 0.0121 |
| CNN-Transformer | 0.4697 ± 0.0132 | 0.4017 ± 0.0168 | 0.4720 ± 0.0125 | 0.3678 ± 0.0078 | 0.2072 ± 0.0183 | 0.3642 ± 0.0088 |
| FFCL | 0.4673 ± 0.0158 | 0.3987 ± 0.0383 | 0.4699 ± 0.0145 | 0.3641 ± 0.0092 | 0.2078 ± 0.0201 | 0.3645 ± 0.0132 |
| ST-Transformer | 0.4810 ± 0.0079 | 0.4137 ± 0.0133 | 0.4795 ± 0.0096 | 0.3052 ± 0.0072 | 0.1083 ± 0.0121 | 0.2833 ± 0.0105 |
| BIOT | 0.5118 ± 0.0118 | 0.4476 ± 0.0254 | 0.5136 ± 0.0112 | 0.3837 ± 0.0187 | 0.2261 ± 0.0262 | 0.3856 ± 0.0203 |
| LaBraM-Base | 0.5273 ± 0.0107 | 0.4698 ± 0.0188 | 0.5288 ± 0.0102 | 0.3976 ± 0.0138 | 0.2386 ± 0.0209 | 0.3974 ± 0.0111 |
| CBraMod | **0.5509** ± 0.0089 | **0.5041** ± 0.0122 | **0.5618** ± 0.0093 | **0.4091** ± 0.0097 | **0.2569** ± 0.0143 | **0.4101** ± 0.0108 |

*Emotion Recognition.* We use FACED (Chen et al., 2023) and SEED-V[1] (Liu et al., 2021b) as the evaluation datasets of emotion recognition. FACED is a large finer-grained affective computing EEG dataset recording 32-channel EEG signals at 250 Hz sampling rate from 123 subjects, which covers nine emotion categories including amusement, inspiration, joy, tenderness, anger, fear, disgust, sadness, and neutral emotion. The EEG signals are segmented into 10,332 10-second samples and resampled to 200 Hz. We use subject 1 to 80 for training, 81 to 100 for validation, and 101 to 123 for test in FACED. SEED-V is an emotion EEG dataset containing five emotion categories (happy, sad, neutral, disgust, and fear), consisting of EEG signals (62 channels, 1000 Hz) from 16 subjects through three sessions per subject, where one session has fifteen trials. The EEG signals are segmented into 117,744 1-second samples and resampled to 200 Hz. We divide the fifteen trials of each session into three equal parts (5:5:5) as the training, validation and test sets, respectively. The experimental results of emotion recognition are presented in Table 2. CBraMod achieves the state-of-the-art performance on both FACED and SEED-V. Specifically, CBraMod obtains a great performance improvement compared to the best baseline LaBraM (0.5041 v.s. 0.4698 in Cohen's Kappa on FACED and 0.2569 v.s. 0.2386 in Cohen's Kappa on SEED-V).

Table 3: The results of different methods on motor imagery classification.

| | PhysioNet-MI, 4-class | | | SHU-MI, 2-class | | |
|---|---|---|---|---|---|---|
| Methods | Balanced Accuracy | Cohen's Kappa | Weighted F1 | Balanced Accuracy | AUC-PR | AUROC |
| EEGNet | 0.5814 ± 0.0125 | 0.4468 ± 0.0199 | 0.5796 ± 0.0115 | 0.5889 ± 0.0177 | 0.6311 ± 0.0142 | 0.6283 ± 0.0152 |
| EEGConformer | 0.6049 ± 0.0104 | 0.4736 ± 0.0171 | 0.6062 ± 0.0095 | 0.5900 ± 0.0107 | 0.6370 ± 0.0093 | 0.6351 ± 0.0101 |
| SPaRCNet | 0.5932 ± 0.0152 | 0.4564 ± 0.0234 | 0.5937 ± 0.0147 | 0.5978 ± 0.0097 | 0.6510 ± 0.0062 | 0.6431 ± 0.0082 |
| ContraWR | 0.5892 ± 0.0133 | 0.4527 ± 0.0248 | 0.5918 ± 0.0116 | 0.5873 ± 0.0128 | 0.6315 ± 0.0105 | 0.6273 ± 0.0113 |
| CNN-Transformer | 0.6053 ± 0.0118 | 0.4725 ± 0.0223 | 0.6041 ± 0.0105 | 0.5975 ± 0.0169 | 0.6412 ± 0.0076 | 0.6323 ± 0.0082 |
| FFCL | 0.5726 ± 0.0092 | 0.4323 ± 0.0182 | 0.5701 ± 0.0079 | 0.5692 ± 0.0252 | 0.5943 ± 0.0172 | 0.6014 ± 0.0168 |
| ST-Transformer | 0.6035 ± 0.0081 | 0.4712 ± 0.0199 | 0.6053 ± 0.0075 | 0.5992 ± 0.0206 | 0.6394 ± 0.0122 | 0.6431 ± 0.0111 |
| BIOT | 0.6153 ± 0.0154 | 0.4875 ± 0.0272 | 0.6158 ± 0.0197 | 0.6179 ± 0.0183 | 0.6770 ± 0.0119 | 0.6609 ± 0.0127 |
| LaBraM-Base | 0.6173 ± 0.0122 | 0.4912 ± 0.0192 | 0.6177 ± 0.0141 | 0.6166 ± 0.0192 | 0.6761 ± 0.0083 | 0.6604 ± 0.0091 |
| CBraMod | **0.6417** ± 0.0091 | **0.5222** ± 0.0169 | **0.6427** ± 0.0100 | **0.6370** ± 0.0151 | **0.7139** ± 0.0088 | **0.6988** ± 0.0068 |

*Motor Imagery Classification.* We use PhysioNet-MI (Goldberger et al., 2000; Schalk et al., 2004) and SHU-MI (Ma et al., 2022) for evaluation on motor imagery classification. PhysioNet-MI is an EEG motor imagery dataset collected from 109 subjects with 64 channels and 160 Hz sampling rate. It covers four motor imagery classes including left fist, right fist, both fists and both feet. The EEG signals are divided into 9,837 4-second samples and resampled to 200 Hz. Subject 1–70, 71–89, 90–109 are used for training, validation and test, respectively. SHU-MI is an EEG motor imagery dataset with two categories (left hand and right hand). It consists of 32-channel EEG signals with 250 Hz sampling rate, collected from 25 subjects. The EEG signals are segmented into 11,988 samples and resampled to 200 Hz. We use subject 1 to 15 for training, subject 16 to 20 for validation, and 21 to 25 for test. The results are shown in Table 3. CBraMod achieves the best performance on both datasets. Specifically, CBraMod achieves a significant performance gain compared to the best baseline LaBraM (0.5222 v.s. 0.4912 in Cohen's Kappa) on PhysioNet-MI. On SHU-MI, CBraMod obtains a better performance compared to the best baseline BIOT (0.6988 v.s. 0.6609 in AUROC).

---

[1]We used the open-source version of SEED-V with 16 subjects, rather than the private version with 20 subjects used in LaBraM (Jiang et al., 2024).

All the above results indicate that CBraMod can learn powerful EEG representations which are helpful to the downstream BCI tasks of emotion recognition and motor imagery classification.

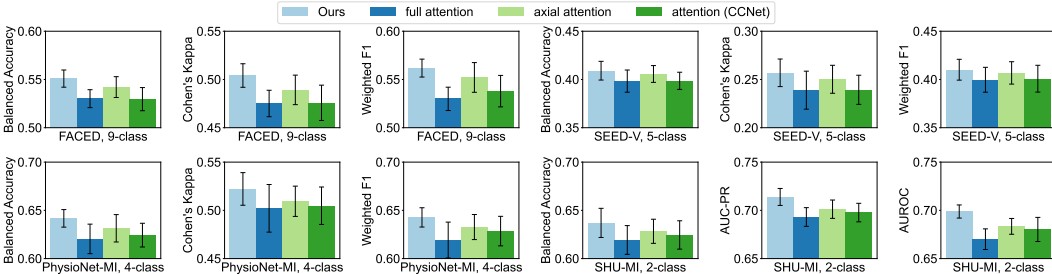

Figure 4: The results of attention mechanism comparison.

**Attention Mechanism Comparison** In order to prove the effectiveness of criss-cross attention mechanism, we compare it with other attention mechanism such as full attention (Vaswani et al., 2017; Dosovitskiy et al., 2020), axial attention (Ho et al., 2019) and the criss-cross attention module in CCNet (Huang et al., 2019). Full attention mechanism equally models the spatial-temporal dependencies among all EEG patches in one transformer block. Axial attention mechanism models the dependencies along specific axial dimensions of EEG patches in one transformer block. It can sequentially model spatial or temporal dependencies among EEG patches in different blocks. In our experiment, the axial attention only models the spatial dependencies in transformer block 1–6 and models the temporal dependencies in transformer block 7–12. The criss-cross attention mechanism in CCNet is based on the affinity operation, using a single attention map to achieve criss-cross modeling. We pre-trained and fine-tuned the models based on the three attention mechanisms with the same settings as CBraMod. The performance comparison on emotion recognition and motor imagery classification is shown in Figure 4. Full attention achieves the worst performance across all datasets. It indicates that full attention equally models the dependencies among all EEG patches, ignoring the unique structural characteristics of EEG signals. Meanwhile, there are more than 570 EEG patches in one EEG sample (19 channels, 30 seconds) in the pre-training dataset, which exceeds the appropriate modeling range for full attention. Axial attention outperforms full attention because it prioritizes directly modeling specific axial dimensions of EEG patches, which can effectively capture the spatial-temporal dependencies among EEG patches. But it performs worse than criss-cross attention in CBraMod. It indicates that the criss-cross attention can model spatial and temporal dependencies among EEG patches in parallel, which are more effective than sequential modeling of axial attention for EEG modeling. The attention in CCNet performs slightly better than full attention but significantly worse that our attention mechanism, indicating that the single-map criss-cross attention designed for image in CCNet may be not suitable for EEG modeling. EEG signals contain heterogeneous spatial and temporal dependencies. Our method employs dual parallel spatial and temporal attentions to capture spatial and temporal dependencies, which is more suitable for EEG modeling.

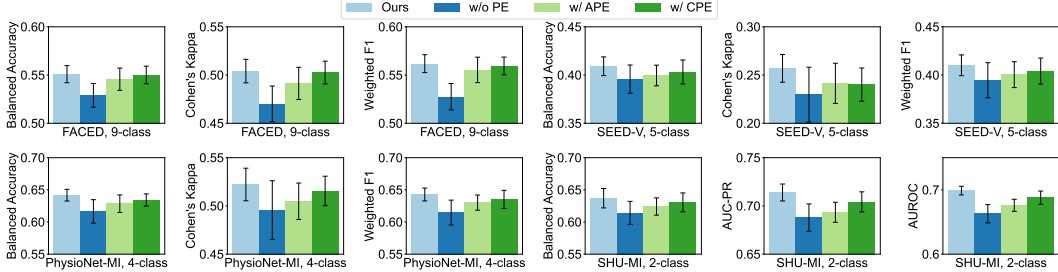

Figure 5: The results of positional encoding comparison.

**Positional Encoding Comparison** We conduct the positional encoding comparison to evaluate the effectiveness of asymmetric conditional positional encoding (ACPE). Specifically, we compare ACPE with following settings: 1) without positional encoding (w/o PE): do not use positional encoding to encode postional information; 2) with absolute positional encoding (w/ APE): replace ACPE with

APE; 3) with conditional positional encoding (w/ CPE): replace ACPE with CPE. The results are shown in Figure 5. CBraMod without PE performs the worst, indicating that positional encoding is important in EEG modeling. APE performs better than w/o PE but worse than CPE and ACPE, indicating that APE is not so effective as CPE in adapting to different EEG formats. CPE, usually used for image patches, performs slightly worse compared to ACPE in CBraMod, indicating that the asymmetric designs are slightly better than symmetric designs in EEG modeling. It may be because that EEG patches exhibit different dependencies in spatial (channel) and temporal dimension, which are different from the dependencies between image patches.

Table 4: The results of ablation study on pre-training.

| | FACED, 9-class | | | SEED-V, 5-class | | |
|---|---|---|---|---|---|---|
| Settings | Balanced Accuracy | Cohen's Kappa | Weighted F1 | Balanced Accuracy | Cohen's Kappa | Weighted F1 |
| clean pre-training | **0.5509** ± 0.0089 | **0.5041** ± 0.0122 | **0.5618** ± 0.0093 | **0.4091** ± 0.0097 | **0.2569** ± 0.0143 | **0.4101** ± 0.0108 |
| dirty pre-training | 0.5319 ± 0.0245 | 0.4768 ± 0.0348 | 0.5397 ± 0.0219 | 0.3914 ± 0.0217 | 0.2224 ± 0.0318 | 0.3894 ± 0.0243 |
| w/o pre-training | 0.5232 ± 0.0216 | 0.4615 ± 0.0289 | 0.5304 ± 0.0187 | 0.3902 ± 0.0241 | 0.2211 ± 0.0384 | 0.3879 ± 0.0225 |

| | PhysioNet-MI, 4-class | | | SHU-MI, 2-class | | |
|---|---|---|---|---|---|---|
| Settings | Balanced Accuracy | Cohen's Kappa | Weighted F1 | Balanced Accuracy | AUC-PR | AUROC |
| clean pre-training | **0.6417** ± 0.0091 | **0.5222** ± 0.0169 | **0.6427** ± 0.0100 | **0.6370** ± 0.0151 | **0.7139** ± 0.0088 | **0.6988** ± 0.0068 |
| dirty pre-training | 0.6245 ± 0.0153 | 0.5063 ± 0.0256 | 0.6223 ± 0.0167 | 0.6301 ± 0.0189 | 0.7026 ± 0.0176 | 0.6895 ± 0.0131 |
| w/o pre-training | 0.6196 ± 0.0143 | 0.4994 ± 0.0289 | 0.6157 ± 0.0145 | 0.6289 ± 0.0179 | 0.7032 ± 0.0145 | 0.6878 ± 0.0166 |

**Ablation Study on Pre-training**     To further evaluate the effectiveness of pre-training, we conduct an ablation study on pre-training. The pre-training ablation experiment is designed as follows: 1) w/o pre-training: directly training CBraMod on downstream datasets; 2) dirty pre-training: pre-training CBraMod on TUEG corpus without bad samples dropping. 3) clean pre-training: pre-training CBraMod on TUEG corpus with bad samples dropping. The results are presented in Table 4. Obviously, clean pre-training achieves the best performance, obtaining significant performance increases compared to dirty pre-training and w/o pre-training. Besides, the performance has a smaller variance compared to dirty pre-training and w/o pre-training. These indicate that our pre-training strategy can help CBraMod learn generic representations from EEG, which can improve the generalizability and stability of CBraMod on downstream datasets. Dirty pre-training performs slightly better than w/o pre-training, indicating that a large amount of dirty data in the original dataset can weaken the effectiveness of pre-training.

# 4    CONCLUSION

In this paper, we propose an EEG foundation model called CBraMod, which can learn generic representations of EEG signals through patch-based masked EEG reconstruction. Specifically, we devise a criss-cross transformer as the backbone of CBraMod to model the spatial and temporal dependencies between EEG patches in parallel, and an asymmetric convolutional positional encoding scheme to encode spatial-temporal positional information of EEG signals with diverse formats. CBraMod is pre-trained on TUEG, a very large corpus of EEG. CBraMod achieves the state-of-the-art performance across up to 10 downstream BCI tasks (12 public datasets), proving its strong capability and generalizability. We hope that the proposed modeling approach and positional encoding scheme can provide meaningful insights for building EEG foundation models, thereby advancing the development of real-world BCI systems.

## ACKNOWLEDGMENTS

This work was supported by STI 2030 Major Projects (2021ZD0200400), Natural Science Foundation of China (No. 61925603) and the Key Program of the Natural Science Foundation of Zhejiang Province, China (No. LZ24F020004). The corresponding authors are Dr. Sha Zhao and Dr. Gang Pan.

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

## A   RELATED WORK

**EEG Decoding.**   EEG is a non-invasive technique to measure brain activity.   Early studies on EEG decoding predominantly employed traditional machine learning methods (Bashashati et al., 2007; McFarland et al., 2006; Lotte et al., 2007; Qi et al., 2019), which usually depend on hand-crafted features that require lots of prior knowledge and could have weak generalizability. With the development of deep learning (LeCun et al., 2015) techniques, an increasing number of researchers are shifting their focus to studying EEG decoding methods based on deep learning (Parvaneh et al., 2019; Craik et al., 2019; Al-Saegh et al., 2021; Sekkal et al., 2022; Yang et al., 2022). For example, some studies utilize convolutional neural network (CNN) to extract temporal and spatial features from EEG for different BCI tasks like motor imagery classification, emotion recognition and seizure detection (Schirrmeister et al., 2017; Sakhavi et al., 2018; Lawhern et al., 2018; Abdelhameed & Bayoumi, 2021; Ding et al., 2022). Long Short-Term Memory (LSTM) is also used for EEG feature extraction and classification on BCI tasks such as motor imagery classification and sleep staging (Wang et al., 2018; Phan et al., 2019). Some researchers propose CNN-LSTM to learn EEG features for motor imagery classification, emotion recognition and sleep staging (Supratak et al., 2017; Zhang et al., 2019a; Dar et al., 2020; Li et al., 2022; Wang et al., 2023a), where CNN is usually used to extract local features and LSTM is utilized to capture global dependencies. Transformer architecture is also utilized to learn spatial-temporal feature for BCI tasks including emotion recognition, sleep staging and person identification (Song et al., 2021; Liu et al., 2021a; Du et al., 2022; Phan et al., 2022). To combine the strengths of CNN and transformer, some works devise a CNN-Transformer network for EEG classification (Song et al., 2022; Peh et al., 2022; Wang et al., 2023b; Zhou et al., 2024b; Wang et al., 2024c). In addition, some studies use graph neural networks learn spatial-temporal features from multi-channel EEG for various BCI tasks (Jia et al., 2020; 2021; Ding et al., 2023). These methods perform well on specific tasks or datasets, but collecting labeled EEG data is costly and labor-intensive. Meantime, the variations in EEG signal formats across different datasets pose challenges for enhancing model performance and generalizability.

**Brain Foundation Model.**   A foundation model (Bommasani et al., 2021) is a deep learning model trained on extensive data, typically using self-supervision at a large scale, and can be adapted, such as through fine-tuning, for various downstream tasks. Foundation models, such as BERT (Devlin et al., 2018), MAE (He et al., 2022), CLIP (Radford et al., 2021), GPT-4 (Achiam et al., 2023), SAM (Kirillov et al., 2023) and SORA (Brooks et al., 2024), have achieved remarkable success in computer vision, natural language, and multimodal, but there is still a vast unexplored potential for foundation models in brain signals. Some studies propose time series pre-trained models (Eldele et al., 2021; Zhang et al., 2022; Woo et al., 2024; Chen et al., 2024; Deng et al., 2024) based on contrastive learning or others for many scenarios (e.g., weather, traffic flow, exchange rates), one of which is brain signals. Some studies learn unsupervised representations from EEG by utilizing self-supervised learning pretext tasks to uncover the structure of clinical EEG signals (Banville et al., 2021). BENDER (Kostas et al., 2021) is proposed to address the problem of limited labeled data on EEG by using a contrastive self-supervised learning task to learn generic EEG representations. MAEEG (Chien et al., 2022) is masked auto-encoder for learning EEG representations by reconstruction-based self-supervised learning. BrainBERT (Wang et al., 2022) is a pre-training model for intracranial recordings, which learns a complex non-linear transformation of neural data by construction of the masked stereo-electroencephalographic (SEEG) spectrogram. Brant (Zhang et al., 2023) is a foundation model for intracranial neural signals which attends long-term dependency and captures spatial correlation across channels. Brant-2 (Yuan et al., 2024) represents an improved version of Brant, tailored for the integration of scalp EEG and intracranial EEG. Brant-X (Zhang et al., 2024) extends this framework to achieve alignment between different physiological signals and EEG. BIOT (Yang et al., 2023) is a generic biosignal learning model which enables joint pre-training and knowledge transfer across different biosignal datasets in the wild. EEG2Rep (Foumani et al., 2024) is a self-prediction approach for self-supervised representation learning from EEG, which predicting masked inputs in the latent representation space. LaBraM (Jiang et al., 2024) is a large brain model, which learns EEG generic representations by predicting the corresponding neural tokens of masked EEG patches. EEGPT (Wang et al., 2024a) is a pretrained transformer model designed for universal EEG feature extraction, based on a mask-based dual self-supervised learning method for efficient feature extraction. However, most of existing EEG foundational model the spatial and temporal dependencies between all EEG patches together, ignoring the unique structural characteristics of

EEG signals. Meanwhile, existing EEG foundation models have limited generalizability, performing well only on a limited range of downstream tasks.

## B    MORE DETAILS FOR EXPERIMENTAL SETTINGS ON PRE-TRAINING

Here, we introduce more details for experimental settings on CBraMod pre-training.

For preprocessing of pre-training dataset, as CBraMod adopts criss-cross EEG modeling which can address EEG signals with long time duration well, we segment all EEG recordings to 30-second EEG samples, which is longer than the time duration of pre-training samples on existing studies such as BIOT (Yang et al., 2023) (10 seconds) and LaBraM (Jiang et al., 2024) (4 or 8 seconds). We choose this time duration for two main reasons: 1) longer segments may help the model learn more long-term dependencies, potentially improving performance on downstream tasks, as noted in BENDER (Kostas et al., 2021); 2) 30 seconds generally covers the length of EEG sample segments for all the downstream tasks in this work. More hyperparameters are listed in Table 5.

Table 5: Hyperparameters for CBraMod pre-training.

|  | Hyperparameters | Settings |
|---|---|---|
| EEG sample | Channels | 19 |
|  | Time points | 6000 |
|  | Patch dimension | 200 |
|  | Sequence length | 6000/200=30 |
|  | Mask ratio | 0.5 |
|  | Mask token | Full zero |
| Patch Encoder (Time-Domain Branch, 3-layer 1D CNN) | Input dimension | {1, 25, 25} |
|  | Output dimension | {25, 25, 25} |
|  | Kernel size | {49, 3, 3} |
|  | Stride | {25, 1, 1} |
|  | Padding | {24, 1, 1} |
| Patch Encoder (Frequency-Domain Branch | FFT function | torch.fft.rfft |
|  | Fully-connected layer | (101, 200) |
| Positional Encoder (1-layer 2D Depthwise CNN) | Input dimension | 200 |
|  | Output dimension | 200 |
|  | Kernel size | (19, 7) |
|  | Stride | (1, 1) |
|  | Padding | (9, 3) |
| Criss-cross transformer | Layers | 12 |
|  | Hidden dimension | 200 |
|  | Heads | 8 |
|  | S-Attention heads | 4 |
|  | T-Attention heads | 4 |
|  | Feed-forward dimension | 800 |
| Pre-training | Epochs | 40 |
|  | Batch size | 128 |
|  | Dropout | 0.1 |
|  | Optimizer | AdamW |
|  | Learning rate | 5e-4 |
|  | Adam $\beta$ | (0.9, 0.999) |
|  | Adam $\epsilon$ | 1e-8 |
|  | Weight decay | 5e-2 |
|  | Scheduler | CosineAnnealingLR |
|  | Cosine cycle epochs | 40 |
|  | Minimal learning rate | 1e-5 |
|  | Clipping gradient norm | 1 |
|  | Weights init | Kaiming normalization |

## C    PRE-TRAINING VISUALIZATION

The pretraining loss curve of CBraMod is presented in Figure 6. From the curve, it is evident that during the 40-epoch pretraining process, the loss function generally follows a downward trend, with minor fluctuations observed between epochs 10 and 14. The overall trend suggests that our model is effectively learning from the pretraining data, leading to the extraction of reliable EEG representations.

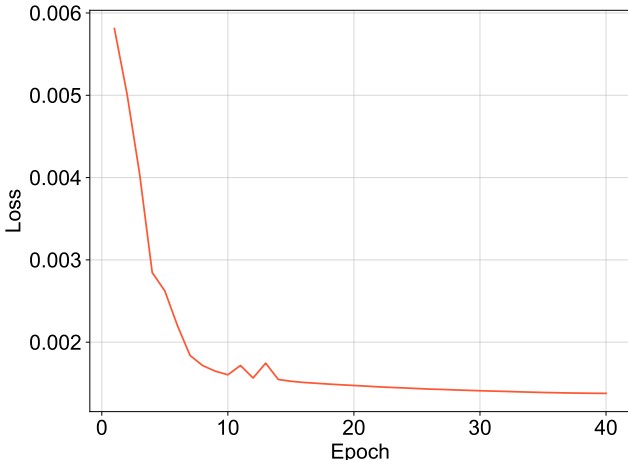

Figure 6: The pre-training loss curve of CBraMod.

## D MORE DETAILS FOR EXPERIMENTAL SETTINGS ON DOWNSTREAM BCI TASKS

We provide more details for experiment settings on downstream BCI tasks.

### D.1 MORE DETAILS FOR DOWNSTREAM TASK

**I. Emotion Recognition** refers to detecting and interpreting emotional states or patterns based on EEG. We choose FACED (Chen et al., 2023) and SEED-V (Liu et al., 2021b) as downstream datasets for this task.

**II. Motor Imagery Classification** is a process that involves decoding or classifying motor imagery tasks based on brain activity patterns captured through EEG. We use PhysioNet-MI (Schalk et al., 2004) and SHU-MI (Ma et al., 2022) to evaluate the performance of CBraMod on this task.

**III. Sleep Staging**, also known as sleep stage classification, categorizes different stages of sleep based on physiological signals (e.g. EEG) recorded during sleep. ISRUC (Khalighi et al., 2016) is a commonly used benchmark for sleep staging.

**IV. Seizure Detection** refers to the process of identifying and recognizing epileptic seizures based on physiological signals, such as EEG. Here, CHB-MIT (Shoeb, 2009) is choosed for this task.

**V. Imagined Speech Classification** involves the identification and interpretation of imagined speech or covert speech intentions based on brain activity patterns such as EEG. BCIC2020-3 (Jeong et al., 2022) is a dataset for this task on 2020 international brain–computer interface competition.

**VI. Mental Disorder Diagnosis** refers to the process of identifying and categorizing mental health states based on EEG. Mumtaz2016 (Mumtaz, 2016) is an EEG dataset collected from patients with major depressive disorder and normal controls.

**VII. Vigilance Estimation** refers to the assessment and measurement of an individual's level of vigilance or sustained attention from EEG. SEED-VIG (Min et al., 2017) is dataset used to assess the level of vigilance.

**VIII. Mental Stress Detection** involves using EEG to identify an individual's level of stress. MentalArithmetic (Zyma et al., 2019) is a dataset containing EEG recordings of subjects before and during the performance of mental arithmetic tasks.

**IX. EEG Events Classification** involves the categorization and identification of specific events or patterns in EEG. We choose TUEV (Obeid & Picone, 2016) for evaluation, which was consistent with BIOT (Yang et al., 2023) and LaBraM (Jiang et al., 2024).

**X. Abnormal Detection** refers to the process of identifying and detecting abnormal patterns or events in EEG. Similarly, TUAB (Obeid & Picone, 2016) is used for evaluation, consistent with BIOT (Yang et al., 2023) and LaBraM (Jiang et al., 2024).

## D.2 BASELINES

Here, we introduce the details of the baselines for performance evaluation.

**EEGNet** (Lawhern et al., 2018) is a compact convolutional neural network based on depthwise and separable convolutions.

**EEGConformer** (Song et al., 2022) is EEG model using CNN to learn low-level local features and self-attention to extract the global correlation within the local temporal features.

**SPaRCNet** (Jing et al., 2023) is a deep neural network based on 1D CNN with dense residual connections.

**ContraWR** (Yang et al., 2021) is a CNN based model, which first transforms the biosignals into multi-channel spectrogram and then uses 2D-CNN based ResNet (He et al., 2016) to extract features from spectrogram.

**CNN-Transformer** (Peh et al., 2022) is model utilizing CNN to extract local features and using transformer to capture global dependencies.

**FFCL** (Li et al., 2022) is neural network combining CNN and LSTM in parallel, where the CNN extracts spatial features and the LSTM extracts temporal features.

**ST-Transformer** Song et al. (2021) is a transformer based network which relies on the attention mechanism to learn the spatial and temporal features of EEG signals.

**BIOT** (Yang et al., 2023) is an EEG foundation model which learns EEG generic representations based on linear transformer and supervised-unsupervised-combined pre-training. It is worth noting that the pre-trained BIOT can only accept EEG signals with a maximum of 18 channels as input. Therefore, for EEG signals with more than 18 channels, we will use multiple BIOT models to process different sets of channels.

**LaBraM** (Jiang et al., 2024) is a large brain model, which learns EEG generic representations by predicting the corresponding neural tokens of masked EEG patches based on full-attention transformer.

## D.3 METRICS

In this section, we introduce the details of the metrics used in the paper. Consistent with LaBraM (Jiang et al., 2024), we adopt the following metrics:

**Balanced Accuracy** is a performance metric that considers the accuracy of each class in imbalanced datasets, which is defined as the average of recall obtained on each class. We use it for both binary classification and multi-class classification.

**AUC-PR** is a performance metric calculating the area under the precision recall (PR) curve for binary classification task.

**AUROC** is a widely used statistic calculating the area under the receiver operating characteristic (ROC) curve. We use it for binary classification.

**Coken's Kappa** is a statistical measure used to assess the agreement between two raters or classifiers in categorical classification tasks, which is usually used for imbalanced multi-class classification task.

**Weighted F1** is a weighted average of individual F1-scores from each class, with each score weighted by the number of samples in the corresponding class, which provides a reliable measure in multi-class classification tasks.

**Pearson's correlation** is a statistical measure that quantifies the linear relationship between two continuous variables, which can be used to quantify the performance of a regression model.

**R2 score**, also known as the coefficient of determination, is a statistical measure commonly used to evaluate the goodness of fit of a regression model.

**RMSE** (Root Mean Square Error) is a commonly used performance metric in regression tasks to measure the average magnitude of the errors made by a regression model, which calculates the square root of the average of the squared differences between the predicted values and the true values.

### D.4    FINE-TUNING

Table 6: Hyperparameters for CBraMod fine-tuning.

| Hyperparameters | Settings |
|---|---|
| Epochs | 50 |
| Batch size | 64 |
| Dropout | 0.1 |
| Optimizer | AdamW |
| Learning rate | 1e-4 |
| Adam $\beta$ | (0.9, 0.999) |
| Adam $\epsilon$ | 1e-8 |
| Weight decay | 5e-2 |
| Scheduler | CosineAnnealingLR |
| Cosine cycle epochs | 50 |
| Minimal learning rate | 1e-6 |
| Clipping gradient norm | 1 |
| Label smoothing (multi-class classification) | 0.1 |

We load the pre-trained weights of CBraMod and replace the reconstruction head with a task-specific head which is composed of multi-layer perceptrons. Here the learned EEG representations are flattened and fed into the task-specific head for downstream tasks. Then we fine-tune CBraMod in downstream datasets. **We employ binary cross-entropy (BCE) loss for binary classification, cross-entropy loss for multi-class classification, and mean-square-error loss function for regression.** More hyperparameters for CBraMod fine-tuning on downstream datasets are shown in Table 6.

## E    MORE RESULTS ON OTHER DOWNSTEAM BCI TASKS

In this section, we report more results of CBraMod and baselines on other downstream BCI tasks.

### E.1    SLEEP STAGING

Table 7: The results of different methods on sleep staging (ISRUC, 5-class).

| Methods | Params | Balanced Accuracy | Cohen's Kappa | Weighted F1 |
|---|---|---|---|---|
| EEGNet | 0.003M | $0.7154 \pm 0.0121$ | $0.7040 \pm 0.0173$ | $0.7513 \pm 0.0124$ |
| EEGConformer | 0.55M | $0.7400 \pm 0.0133$ | $0.7143 \pm 0.0162$ | $0.7634 \pm 0.0151$ |
| SPaRCNet | 0.79M | $0.7487 \pm 0.0075$ | $0.7097 \pm 0.0132$ | $0.7624 \pm 0.0092$ |
| ContraWR | 1.6M | $0.7402 \pm 0.0126$ | $0.7178 \pm 0.0156$ | $0.7610 \pm 0.0137$ |
| CNN-Transformer | 3.2M | $0.7363 \pm 0.0087$ | $0.7129 \pm 0.0121$ | $0.7719 \pm 0.0105$ |
| FFCL | 2.4M | $0.7277 \pm 0.0182$ | $0.7016 \pm 0.0291$ | $0.7614 \pm 0.0197$ |
| ST-Transformer | 3.5M | $0.7381 \pm 0.0205$ | $0.7013 \pm 0.0352$ | $0.7681 \pm 0.0175$ |
| DeepSleepNet | 21M | $0.7419 \pm 0.0144$ | $0.7036 \pm 0.0241$ | $0.7643 \pm 0.0122$ |
| USleep | 1.1M | $0.7586 \pm 0.0116$ | $0.7209 \pm 0.0143$ | $0.7805 \pm 0.0105$ |
| BIOT | 3.2M | $0.7527 \pm 0.0121$ | $0.7192 \pm 0.0231$ | $0.7790 \pm 0.0146$ |
| LaBraM-Base | 5.8M | $0.7633 \pm 0.0102$ | $0.7231 \pm 0.0182$ | $0.7810 \pm 0.0133$ |
| CBraMod | 4.0M | $\mathbf{0.7865} \pm 0.0110$ | $\mathbf{0.7442} \pm 0.0152$ | $\mathbf{0.8011} \pm 0.0099$ |

ISRUC (Khalighi et al., 2016) is a sleep dataset, which comprises 100 all-night PSG recordings of 100 adults. In our experiments, we used PSG recordings in Sub-group 1 to evaluate our method. The

EEG signals of ISRUC are collected with 6 channels (F3-A2, C3-A2, O1-A2, F4-A1, C4-A1, O2-A1) and 200 Hz sampling rate. All EEG signals are segmented into 89,240 30-second samples, which are classified into five different sleep stages (Wake, N1, N2, N3, REM) by sleep experts according to the American Academy of Sleep Medicine (AASM) sleep standard (Iber et al., 2007). In our experiment, we set subject 1 to 80 as training set, subject 81 to 90 as validation set and subject 91 to 100 as test set. Notably, according to AASM standard (Iber et al., 2007), experts usually identify the current stage based on the transition patterns, combining the sleep stages of other samples in a sleep sequence together. The transition patterns of sleep stages between epochs play a critical role in automatic sleep staging. Existing works on sleep staging (Phan & Mikkelsen, 2022) usually regard sleep staging as a sequence-to-sequence classification task and set 20 as the sequence length, where one sleep sequence has 20 30-second samples. Therefore, in experiment on sleep staging, we set each compared model as sample encoder, and use a one-layer transformer as sequence encoder. Specifically, we included two widely recognized baseline in the field of sleep staging, DeepSleepNet (Supratak et al., 2017) and USleep (Perslev et al., 2021), to compare the performance of our method with that of classic algorithms in the field. As shown in Table 7, CBraMod outperforms all baselines. Specifically, CBraMod obtain significantly better performance compared to existing best method LaBraM (0.7442 v.s. 0.7231 in Cohen's Kappa).

## E.2 SEIZURE DETECTION

Table 8: The results of different methods on seizure detection (CHB-MIT, 2-class).

| Methods | Params | Balanced Accuracy | AUC-PR | AUROC |
|---|---|---|---|---|
| EEGNet | 0.003M | $0.5658 \pm 0.0106$ | $0.1914 \pm 0.0182$ | $0.8048 \pm 0.0136$ |
| EEGConformer | 0.55M | $0.5976 \pm 0.0141$ | $0.2209 \pm 0.0215$ | $0.8226 \pm 0.0170$ |
| SPaRCNet | 0.79M | $0.5876 \pm 0.0191$ | $0.1247 \pm 0.0119$ | $0.8143 \pm 0.0148$ |
| ContraWR | 1.6M | $0.6344 \pm 0.0002$ | $0.2264 \pm 0.0174$ | $0.8097 \pm 0.0114$ |
| CNN-Transformer | 3.2M | $0.6389 \pm 0.0067$ | $0.2479 \pm 0.0227$ | $0.8662 \pm 0.0082$ |
| FFCL | 2.4M | $0.6262 \pm 0.0104$ | $0.2049 \pm 0.0346$ | $0.8271 \pm 0.0051$ |
| ST-Transformer | 3.5M | $0.5915 \pm 0.0195$ | $0.1422 \pm 0.0094$ | $0.8237 \pm 0.0491$ |
| BIOT | 3.2M | $0.7068 \pm 0.0457$ | $0.3277 \pm 0.0460$ | $0.8761 \pm 0.0284$ |
| LaBraM-Base | 5.8M | $0.7075 \pm 0.0358$ | $0.3287 \pm 0.0402$ | $0.8679 \pm 0.0199$ |
| CBraMod | 4.0M | $\mathbf{0.7398} \pm 0.0284$ | $\mathbf{0.3689} \pm 0.0382$ | $\mathbf{0.8892} \pm 0.0154$ |

CHB-MIT (Goldberger et al., 2000; Shoeb, 2009) is a database, collected at the Children's Hospital Boston, consisting of EEG recordings from 23 pediatric subjects with intractable seizures. Subjects were monitored for up to several days following withdrawal of anti-seizure medication in order to characterize their seizures and assess their candidacy for surgical intervention. All EEG signals are collected based on the international 10-20 system of EEG electrode positions and are sampled at 256 Hz, with binary classes (seizure or not). In our experiment, we preprocess the EEG signals of CHB-MIT based on the strategy consistent with BIOT (Yang et al., 2023). We use the common 16 bipolar montage channels for CHB-MIT dataset. All EEG signals are resampled to 200 Hz and divided 326,993 10-second samples. We use subject 1 to 19 for training, subject 20, 21 for validation, and subject 22,23 for test. The results of seizure detection are shown in Table 8. CBraMod achieves the state-of-the-art performance.

## E.3 IMAGINED SPEECH CLASSIFICATION

BCIC2020-3 (Jeong et al., 2022) is a dataset for imagined speech classification on 2020 international brain–computer interface competition. During the data collection experiment, 15 subjects were seated in a comfortable chair in front of a 24-inch LCD monitor screen. The subjects were instructed to imagine the silent pronunciation of the given word as if they were performing real speech, without moving any articulators nor making the sound. EEG signals of five-class imagined speech words/phrases ("hello", "help me", "stop", "thank you" and "yes") were recorded at 64 channels and 256 Hz sampling rate. Training, validation and test set are provided by the dataset. Specifically, 60 trials per class are released for training purpose, 10 trials per class are released for validation purpose and 10 trials per class are released for test purpose. The sum of trials is $80 \times 5 \times 15 = 6000$.

Table 9: The results of different methods on imagined speech classification (BCIC2020-3, 5-class).

| Methods | Params | Balanced Accuracy | Cohen's Kappa | Weighted F1 |
|---|---|---|---|---|
| EEGNet | 0.003M | $0.4413 \pm 0.0096$ | $0.3016 \pm 0.0123$ | $0.4413 \pm 0.0102$ |
| EEGConformer | 0.55M | $0.4506 \pm 0.0133$ | $0.3133 \pm 0.0183$ | $0.4488 \pm 0.0154$ |
| SPaRCNet | 0.79M | $0.4426 \pm 0.0156$ | $0.3033 \pm 0.0233$ | $0.4420 \pm 0.0108$ |
| ContraWR | 1.6M | $0.4257 \pm 0.0162$ | $0.3078 \pm 0.0218$ | $0.4407 \pm 0.0182$ |
| CNN-Transformer | 3.2M | $0.4533 \pm 0.0092$ | $0.3166 \pm 0.0118$ | $0.4506 \pm 0.0127$ |
| FFCL | 2.4M | $0.4678 \pm 0.0197$ | $0.3301 \pm 0.0359$ | $0.4689 \pm 0.0205$ |
| ST-Transformer | 3.5M | $0.4126 \pm 0.0122$ | $0.2941 \pm 0.0159$ | $0.4247 \pm 0.0138$ |
| BIOT | 3.2M | $0.4920 \pm 0.0086$ | $0.3650 \pm 0.0176$ | $0.4917 \pm 0.0079$ |
| LaBraM-Base | 5.8M | $0.5060 \pm 0.0155$ | $0.3800 \pm 0.0242$ | $0.5054 \pm 0.0205$ |
| CBraMod | 4.0M | $\textbf{0.5373} \pm 0.0108$ | $\textbf{0.4216} \pm 0.0163$ | $\textbf{0.5383} \pm 0.0096$ |

One trial is a 3-second EEG signal, which is regard as one sample. Thus we get 6,000 64-channel 3-second EEG samples, which are resampled to 200 Hz. The experimental results of imagined speech classification are shown in Table 9. CBraMod achieves a great performance increases compared to the best baseline LaBraM (0.4216 v.s. 0.3800 in Cohen's Kappa).

E.4 MENTAL DISORDER DIAGNOSIS

Table 10: The results of different methods on mental disorder diagnosis (Mumtaz2016, 2-class).

| Methods | Params | Balanced Accuracy | AUC-PR | AUROC |
|---|---|---|---|---|
| EEGNet | 0.003M | $0.9232 \pm 0.0104$ | $0.9626 \pm 0.0095$ | $0.9639 \pm 0.0093$ |
| EEGConformer | 0.55M | $0.9308 \pm 0.0117$ | $0.9684 \pm 0.0105$ | $0.9702 \pm 0.0101$ |
| SPaRCNet | 0.79M | $0.9316 \pm 0.0095$ | $0.9754 \pm 0.0065$ | $0.9781 \pm 0.0083$ |
| ContraWR | 1.6M | $0.9195 \pm 0.0115$ | $0.9589 \pm 0.0102$ | $0.9621 \pm 0.0092$ |
| CNN-Transformer | 3.2M | $0.9305 \pm 0.0068$ | $0.9757 \pm 0.0074$ | $0.9742 \pm 0.0059$ |
| FFCL | 2.4M | $0.9314 \pm 0.0038$ | $0.9717 \pm 0.0021$ | $0.9753 \pm 0.0033$ |
| ST-Transformer | 3.5M | $0.9135 \pm 0.0103$ | $0.9578 \pm 0.0086$ | $0.9594 \pm 0.0059$ |
| BIOT | 3.2M | $0.9358 \pm 0.0052$ | $0.9736 \pm 0.0034$ | $0.9758 \pm 0.0042$ |
| LaBraM-Base | 5.8M | $0.9409 \pm 0.0079$ | $0.9798 \pm 0.0093$ | $0.9782 \pm 0.0057$ |
| CBraMod | 4.0M | $\textbf{0.9560} \pm 0.0056$ | $\textbf{0.9923} \pm 0.0032$ | $\textbf{0.9921} \pm 0.0025$ |

Mumtaz2016 (Mumtaz, 2016) is an EEG dataset collected from 34 patients with major depressive disorder (MDD) and 30 normal controls (NCs). All EEG signals recorded from 19 electrodes placed according to the international 10-20 system. The sampling rate is 256 Hz. The data collection experiment comprises three sessions including eyes-open session, eyes-closs session and task session. In our experiment, we only use eyes-open and eyes-close sessions. 24 MDD patients and 19 NCs are used for training, 5 MDD patients and 4 NCs are used for validation, and 5 MDD patients and 5 NCs are used for test. EEG signals are band-pass filtered (0.3Hz–75Hz) to remove low and high frequency noise, notch filtered (50Hz) to remove power line noise and resampled to 200Hz. Then we segment all EEG signals into 7,143 5-second samples. As shown in Table 10, CBraMod achieve the state-of-the-art performance. Specifically, CBraMod performs 1.4% better compared to the best baseline LaBraM (0.9560 v.s. 0.9409 in balanced accuracy, 0.9923 v.s. 0.9798 in AUC-PR and 0.9921 v.s. 0.9782 in AUROC).

E.5 VIGILANCE ESTIMATION

SEED-VIG (Min et al., 2017) is dataset oriented at exploring the vigilance estimation problem. In the data collection experiment, the researchers built a virtual driving system, in which an enormous screen is placed in front of a real car. Subjects can play a driving game in the car, as if they are driving in the real-world environment. The SEED-VIG dataset is collected when the subjects drive in the system. The vigilance level is labeled with the PERCLOS indicator by the SMI eye-tracking glasses. All EEG signals are recorded from 23 subjects at 17 channels and 200 Hz sampling rate, and

Table 11: The results of different methods on vigilance estimation (SEED-VIG, regression).

| Methods | Params | Pearson's Correlation | R2 Score | RMSE↓ |
|---|---|---|---|---|
| EEGNet | 0.003M | $0.5701 \pm 0.0167$ | $0.2366 \pm 0.0084$ | $0.2828 \pm 0.0074$ |
| EEGConformer | 0.55M | $0.5750 \pm 0.0139$ | $0.2344 \pm 0.0091$ | $0.2850 \pm 0.0083$ |
| SPaRCNet | 0.79M | $0.5715 \pm 0.0163$ | $0.2433 \pm 0.0055$ | $0.2798 \pm 0.0043$ |
| ContraWR | 1.6M | $0.5854 \pm 0.0142$ | $0.2453 \pm 0.0062$ | $0.2782 \pm 0.0056$ |
| CNN-Transformer | 3.2M | $0.5714 \pm 0.0172$ | $0.2371 \pm 0.0052$ | $0.2805 \pm 0.0039$ |
| FFCL | 2.4M | $0.5647 \pm 0.0097$ | $0.2301 \pm 0.0035$ | $0.2914 \pm 0.0052$ |
| ST-Transformer | 3.5M | $0.5752 \pm 0.0127$ | $0.2366 \pm 0.0071$ | $0.2838 \pm 0.0036$ |
| BIOT | 3.2M | $0.5996 \pm 0.0182$ | $0.2543 \pm 0.0073$ | $0.2742 \pm 0.0029$ |
| LaBraM-Base | 5.8M | $0.5931 \pm 0.0098$ | $0.2432 \pm 0.0085$ | $0.2762 \pm 0.0048$ |
| CBraMod | 4.0M | $\mathbf{0.6459} \pm 0.0098$ | $\mathbf{0.3365} \pm 0.0068$ | $\mathbf{0.2587} \pm 0.0039$ |

are segmented into 20,355 8-second samples. In our experiment, we use subject 1 to 15 for training, subject 16 to 19 for validation and subject 20 to 23 for test. The experiment results of vigilance estimation are presented in Table 11. Obviously, CBraMod obtains a great performance increases compared to the best baseline BIOT (0.5996 v.s. 0.6459 in Pearson's correlation, 0.3365 v.s. 0.2543 in R2 score and 0.2587 v.s. 0.2742 in RMSE).

### E.6 MENTAL STRESS DETECTION

Table 12: The results of different methods on mental stress detection (MentalArithmetic, 2-class).

| Methods | Params | Balanced Accuracy | AUC-PR | AUROC |
|---|---|---|---|---|
| EEGNet | 0.003M | $0.6770 \pm 0.0116$ | $0.5763 \pm 0.0102$ | $0.7321 \pm 0.0108$ |
| EEGConformer | 0.55M | $0.6805 \pm 0.0123$ | $0.5829 \pm 0.0134$ | $0.7424 \pm 0.0128$ |
| SPaRCNet | 0.79M | $0.6879 \pm 0.0107$ | $0.5825 \pm 0.0193$ | $0.7418 \pm 0.0132$ |
| ContraWR | 1.6M | $0.6631 \pm 0.0097$ | $0.5787 \pm 0.0164$ | $0.7332 \pm 0.0082$ |
| CNN-Transformer | 3.2M | $0.6779 \pm 0.0268$ | $0.5777 \pm 0.0285$ | $0.7258 \pm 0.0336$ |
| FFCL | 2.4M | $0.6798 \pm 0.0142$ | $0.5786 \pm 0.0266$ | $0.7330 \pm 0.0198$ |
| ST-Transformer | 3.5M | $0.6631 \pm 0.0173$ | $0.5672 \pm 0.0259$ | $0.7132 \pm 0.0174$ |
| BIOT | 3.2M | $0.6875 \pm 0.0186$ | $0.6004 \pm 0.0195$ | $0.7536 \pm 0.0144$ |
| LaBraM-Base | 5.8M | $0.6909 \pm 0.0125$ | $0.5999 \pm 0.0155$ | $0.7721 \pm 0.0093$ |
| CBraMod | 4.0M | $\mathbf{0.7256} \pm 0.0132$ | $\mathbf{0.6267} \pm 0.0099$ | $\mathbf{0.7905} \pm 0.0073$ |

MentalArithmetic (Goldberger et al., 2000; Zyma et al., 2019) is a dataset containing EEG recordings of 36 subjects before and during the performance of mental arithmetic tasks. EEG recordings before mental arithmetic tasks are classified into the label of "without mental stress" and ones during mental arithmetic tasks are classified into the label of "with mental stress". All EEG signals are recorded from 20 electrodes placed according to the international 10-20 system at 500 Hz sampling rate. Band-pass filtering (0.5Hz–45Hz) are used to remove low and high frequency noise. In our experiment, we resample EEG signals to 200 Hz and segment them into 1,707 5-second samples. Subject 1 to 28 are set to training set, subject 29 to 32 are set to validation set and subject 33 to 36 are set to test set. As shown in Table 12, CBraMod achieves the state-of-the-art performance. Specifically, CBraMod performs 2.5% better compared to the best baselines LaBraM (0.7256 v.s. 0.6909 in balanced accuracy, 0.6267 v.s. 0.5999 in AUC-PR and 0.7905 v.s. 0.7721 in AUROC).

### E.7 EVENT TYPE CLASSIFICATION

TUEV (Obeid & Picone, 2016) is EEG corpus that contains annotations of EEG segments as one of six classes: (1) spike and sharp wave (SPSW), (2) generalized periodic epileptiform discharges (GPED), (3) periodic lateralized epileptiform discharges (PLED), (4) eye movement (EYEM), (5) artifact (ARTF) and (6) background (BCKG), which is usually used by existing studies (Yang et al., 2023; Jiang et al., 2024). The EEG signals are recorded at 23 channels and 250 Hz sampling rate. For fair comparison with reported results by BIOT (Yang et al., 2023) and LaBraM (Jiang et al., 2024),

Table 13: The results of different methods on event type classification (TUEV, 6-class).

| Methods | Params | Balanced Accuracy | Cohen's Kappa | Weighted F1 |
|---|---|---|---|---|
| EEGNet | 0.003M | $0.3876 \pm 0.0143$ | $0.3577 \pm 0.0155$ | $0.6539 \pm 0.0120$ |
| EEGConformer | 0.55M | $0.4074 \pm 0.0164$ | $0.3967 \pm 0.0195$ | $0.6983 \pm 0.0152$ |
| SPaRCNet | 0.79M | $0.4161 \pm 0.0262$ | $0.4233 \pm 0.0181$ | $0.7024 \pm 0.0104$ |
| ContraWR | 1.6M | $0.4384 \pm 0.0349$ | $0.3912 \pm 0.0237$ | $0.6893 \pm 0.0136$ |
| CNN-Transformer | 3.2M | $0.4087 \pm 0.0161$ | $0.3815 \pm 0.0134$ | $0.6854 \pm 0.0293$ |
| FFCL | 2.4M | $0.3979 \pm 0.0104$ | $0.3732 \pm 0.0188$ | $0.6783 \pm 0.0120$ |
| ST-Transformer | 3.5M | $0.3984 \pm 0.0228$ | $0.3765 \pm 0.0306$ | $0.6823 \pm 0.0190$ |
| BIOT | 3.2M | $0.5281 \pm 0.0225$ | $0.5273 \pm 0.0249$ | $0.7492 \pm 0.0082$ |
| LaBraM-Base | 5.8M | $0.6409 \pm 0.0065$ | $0.6637 \pm 0.0093$ | $0.8312 \pm 0.0052$ |
| LaBraM-Large | 46M | $0.6581 \pm 0.0156$ | $0.6622 \pm 0.0136$ | $0.8315 \pm 0.0040$ |
| LaBraM-Huge | 369M | $0.6616 \pm 0.0170$ | $0.6745 \pm 0.0195$ | $0.8329 \pm 0.0086$ |
| CBraMod (excluding TUEV) | 4.0M | $0.6659 \pm 0.0124$ | $0.6744 \pm 0.0121$ | $0.8331 \pm 0.0071$ |
| CBraMod | 4.0M | $\mathbf{0.6671} \pm 0.0107$ | $\mathbf{0.6772} \pm 0.0096$ | $\mathbf{0.8342} \pm 0.0064$ |

we adopt a similar preprocessing strategy as theirs. Specifically, we use the common 16 bipolar montage channels in the international 10-20 system for TUEV. A band-pass filtered (0.3 Hz–75 Hz) was applied to remove the low-frequency and high-frequency noise. A notch filter (60 Hz) was used to remove the power line noise. All EEG signals are resampled to 200 Hz and divided into 112,491 5-second samples. The original dataset provide the training and test splits. We further divide the training subjects into training and validation set by 80%:20%, consistent with BIOT. Given that the TUEV dataset is a subset of the pretraining dataset TUEG, we introduced an additional experimental setup, CBraMod (excluding TUEV), to eliminate potential effects of data leakage. In this setup, we excluded TUEV from the pretraining process, re-pretrained a new instance of CBraMod, and subsequently evaluated it on TUEV. The experimental results of event type classification are shown in Table 13. LaBraM-Huge achieves good performance with such a large number of parameters (369 M), but CBraMod still performs slightly better compared to it (0.6671 v.s. 0.6616 in balanced accuracy, 0.6772 v.s. 0.6745 in Cohen's Kappa and 0.8342 v.s. 0.8329 in weighted F1). CBraMod (excluding TUEV) exhibits slight fluctuations compared to CBraMod, yet it remains competitive. This indicates that our method successfully learns a generic EEG representation through pre-training, thereby enhancing the performance on datasets unseen during pre-training.

## E.8 ABNORMAL DETECTION

Table 14: The results of different methods on abnormal detection (TUAB, 2-class).

| Methods | Params | Balanced Accuracy | AUC-PR | AUROC |
|---|---|---|---|---|
| EEGNet | 0.003M | $0.7642 \pm 0.0036$ | $0.8299 \pm 0.0043$ | $0.8412 \pm 0.0031$ |
| EEGConformer | 0.55M | $0.7758 \pm 0.0049$ | $0.8427 \pm 0.0054$ | $0.8445 \pm 0.0038$ |
| SPaRCNet | 0.79M | $0.7896 \pm 0.0018$ | $0.8414 \pm 0.0018$ | $0.8676 \pm 0.0012$ |
| ContraWR | 1.6M | $0.7746 \pm 0.0041$ | $0.8421 \pm 0.0104$ | $0.8456 \pm 0.0074$ |
| CNN-Transformer | 3.2M | $0.7777 \pm 0.0022$ | $0.8433 \pm 0.0039$ | $0.8461 \pm 0.0013$ |
| FFCL | 2.4M | $0.7848 \pm 0.0038$ | $0.8448 \pm 0.0065$ | $0.8569 \pm 0.0051$ |
| ST-Transformer | 3.5M | $0.7966 \pm 0.0023$ | $0.8521 \pm 0.0026$ | $0.8707 \pm 0.0019$ |
| BIOT | 3.2M | $0.7959 \pm 0.0057$ | $0.8792 \pm 0.0023$ | $0.8815 \pm 0.0043$ |
| LaBraM-Base | 5.8M | $0.8140 \pm 0.0019$ | $0.8965 \pm 0.0016$ | $0.9022 \pm 0.0009$ |
| LaBraM-Large | 46M | $0.8226 \pm 0.0015$ | $0.9130 \pm 0.0005$ | $0.9127 \pm 0.0005$ |
| LaBraM-Huge | 369M | $0.8258 \pm 0.0011$ | $0.9204 \pm 0.0011$ | $0.9162 \pm 0.0016$ |
| CBraMod (excluding TUAB) | 4.0M | $0.8249 \pm 0.0025$ | $0.9221 \pm 0.0015$ | $0.9156 \pm 0.0017$ |
| CBraMod | 4.0M | $\mathbf{0.8289} \pm 0.0022$ | $\mathbf{0.9258} \pm 0.0008$ | $\mathbf{0.9227} \pm 0.0011$ |

We use TUAB (Obeid & Picone, 2016) for evaluation on abnormal detection consistent with BIOT (Yang et al., 2023) and LaBraM (Jiang et al., 2024). TUAB is an EEG corpus that have been annotated as normal or abnormal. Similar to TUEV, the EEG signals of TUAB are recorded at 23 channels and 250 Hz sampling rate. For fair comparison, we also adopt a similar preprocess-

ing strategy as BIOT and LaBraM. The common 16 bipolar montage channels in the international 10-20 system are used for TUAB. We utilize a band-pass filtered (0.3 Hz–75 Hz) to remove the low-frequency and high-frequency noise, and use a notch filter (60 Hz) to remove the power line noise. Then all EEG signals are resampled to 200 Hz and divided into 409,455 10-second samples, which are used for binary classification to predict normal/abnormal. The original dataset provide the training and test splits. Be same as BIOT, we further divide the training subjects into training and validation set by 80%:20%. Considering that the TUAB dataset is a subset of the pretraining dataset TUEG, we implemented an additional experimental setup, CBraMod (excluding TUAB), to mitigate potential data leakage effects. In this setup, TUAB was excluded from the pre-training process, a new instance of CBraMod was re-pretrained, and its performance was subsequently evaluated on TUAB. As shown in Table 14, CBraMod achieves the state-of-the-art performance, obtaining slightly better results compared to LaBraM-Huge. CBraMod (excluding TUAB) exhibits slight fluctuations compared to CBraMod but remains competitive, demonstrating its capability to learn generic EEG representations that enhance performance on datasets not encountered during pre-training.

## E.9 MOTOR IMAGERY CLASSIFICATION

Table 15: The results of different methods on motor imagery classification (BCIC-IV-2a, 4-class).

| Methods | Params | Balanced Accuracy | AUC-PR | AUROC |
|---|---|---|---|---|
| EEGNet | 0.003M | $0.4482 \pm 0.0094$ | $0.2693 \pm 0.0121$ | $0.4226 \pm 0.0108$ |
| EEGConformer | 0.55M | $0.4696 \pm 0.0106$ | $0.2924 \pm 0.0141$ | $0.4533 \pm 0.0128$ |
| SPaRCNet | 0.79M | $0.4635 \pm 0.0117$ | $0.2847 \pm 0.0147$ | $0.4432 \pm 0.0126$ |
| ContraWR | 1.6M | $0.4678 \pm 0.0125$ | $0.2905 \pm 0.0160$ | $0.4413 \pm 0.0142$ |
| CNN-Transformer | 3.2M | $0.4600 \pm 0.0108$ | $0.2800 \pm 0.0148$ | $0.4460 \pm 0.0114$ |
| FFCL | 2.4M | $0.4470 \pm 0.0143$ | $0.2627 \pm 0.0176$ | $0.4238 \pm 0.0139$ |
| ST-Transformer | 3.5M | $0.4575 \pm 0.0145$ | $0.2733 \pm 0.0198$ | $0.4471 \pm 0.0142$ |
| BIOT | 3.2M | $0.4748 \pm 0.0093$ | $0.2997 \pm 0.0139$ | $0.4607 \pm 0.0125$ |
| LaBraM-Base | 5.8M | $0.4869 \pm 0.0085$ | $0.3159 \pm 0.0154$ | $0.4758 \pm 0.0103$ |
| CBraMod | 4.0M | $\mathbf{0.5138} \pm 0.0066$ | $\mathbf{0.3518} \pm 0.0094$ | $\mathbf{0.4984} \pm 0.0085$ |

To further validate the performance of CBraMod on the motor imagery tasks, we conducted additional experiments on the BCIC-IV-2a (Brunner et al., 2008) dataset[2]. The BCI Competition IV Dataset 2a, provided by Graz University of Technology, comprises EEG recordings from 9 subjects performing 4 motor imagery tasks: imagining movements of the left hand, right hand, both feet, and tongue. Data were collected over two sessions on separate days using 22 Ag/AgCl electrodes at a sampling rate of 250 Hz. Each session consisted of 288 EEG trials, with 72 trials per task. We used [2, 6] seconds of each trial. A band-pass filtered (0.3 Hz–40 Hz) was applied to remove the low-frequency and high-frequency noise. We resample the EEG signals to 200 Hz and obtain 5,088 4-second samples. We use a strict subject-indenpendent train/validation/test strategy as other MI datasets in our paper. Subject 1-5, 6-7, 8-9 are used for training, validation, and test, respectively. The results are shown in Table 15. CBraMod continues to outperform existing methods on this dataset, further reinforcing the generalizability of our method.

All the results across such wide range of downstream BCI tasks indicate that CBraMod can learn generic EEG representations, which contributes to its strong capability and generalizability.

## F SCALING DATA SIZE AND MODEL SIZE

Given the positioning of this work as a foundational model, we conduct experiments to explore whether scaling laws hold at both the data scale and model size levels. Specifically, in the data scale experiments, we examine how the performance of CBraMod varies with different pretraining data sizes, ranging from 1 hour to 9000 hours. For the model size experiments, we design multiple variants

---

[2]As the reviewer suggested, we have added an additional comparison experiment using the BCIC-IV-2a dataset. For the sake of narrative clarity, we did not include this dataset in the count of the 12 publicly available downstream datasets presented in the main text. Including this dataset, we have actually evaluated a total of 13 downstream datasets.

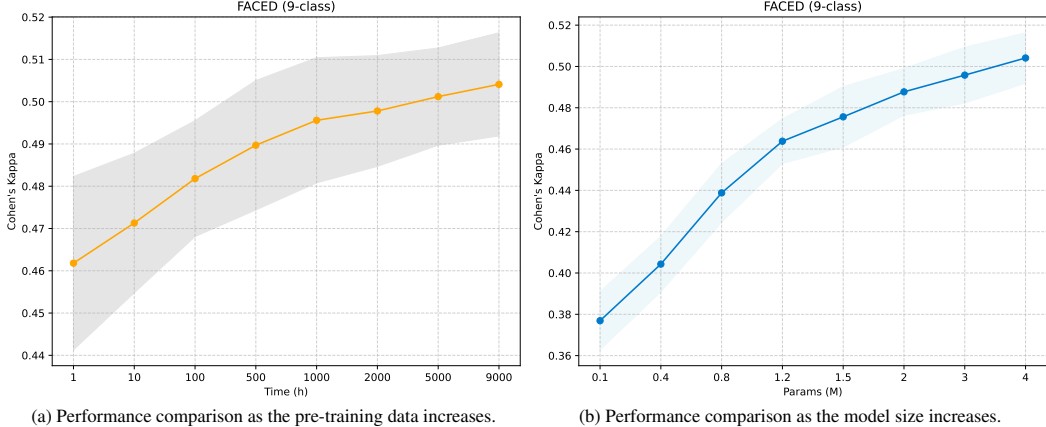

(a) Performance comparison as the pre-training data increases.

(b) Performance comparison as the model size increases.

Figure 7: Scaling Data Size and Model Size.

of CBraMod with varying model sizes (0.1M to 4M parameters) and evaluate their performance on downstream tasks. The results of these experiments are presented in Table 7. As shown in Table 7(a), the performance of CBraMod improves as the scale of pretraining data increases, although the rate of improvement slows beyond 1000 hours. Similarly, Table 7(b) demonstrates that larger model sizes lead to better performance in downstream tasks. Nevertheless, the 9000-hour pretraining data and 4M model size explored in this study do not represent the upper bounds of scaling laws. In the domain of large language models, both data scale and model size have already surpassed the billion-level threshold. While computational constraints have limited our exploration to this range, prior studies (Jiang et al., 2024) on scaling laws suggest that larger model sizes could further improve performance in EEG downstream tasks.

# G  ARCHITECTURE COMPARISON ON OUR PRE-TRAINING DATASETS

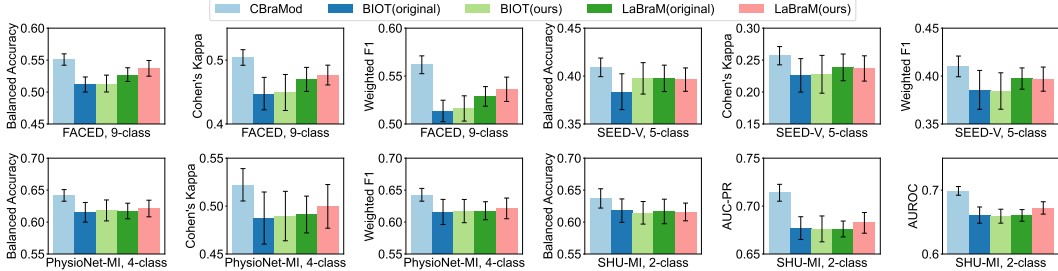

Figure 8: Performance comparison with architectures of existing EEG foundation models on our pre-training datasets.

In this paper, we use different model architecture and pre-training dataset compared to BIOT (Yang et al., 2023) and LaBraM (Jiang et al., 2024). To further evaluate whether the model architecture or the pre-training dataset contributes more to performance improvement, we conduct a comparison experiment. Specifically, we pre-trained the architecture of BIOT and LaBraM on our pre-training dataset with the same settings as CBraMod, and compared the performance with BIOT (original), LaBraM (original) and CBraMod on downstream datasets. The experimental results are shown in Figure 8.

BIOT (ours) and BIOT (original) exhibit very similar performance. LaBraM (ours) generally achieves slightly better performance than LaBraM (original), particularly on the FACED dataset. But CBraMod performs significantly better than LaBraM (ours) on all datasets. All results empirically prove that the model architecture contributes more to the performance improvement compared to the larger size of pretraining dataset.

## H  COMPARISON ON SEGMENT LENGTH

Table 16: Performance comparison on segment length of pre-training data. **Bold** indicates the best. Underline indicates the second best.

| | FACED, 9-class | | | SEED-V, 5-class | | |
|---|---|---|---|---|---|---|
| Methods | Balanced Accuracy | Cohen's Kappa | Weighted F1 | Balanced Accuracy | Cohen's Kappa | Weighted F1 |
| BIOT | $0.5118 \pm 0.0118$ | $0.4476 \pm 0.0254$ | $0.5136 \pm 0.0112$ | $0.3837 \pm 0.0187$ | $0.2261 \pm 0.0262$ | $0.3856 \pm 0.0203$ |
| LaBraM | $0.5273 \pm 0.0107$ | $0.4698 \pm 0.0188$ | $0.5288 \pm 0.0102$ | $0.3976 \pm 0.0138$ | $0.2386 \pm 0.0209$ | $0.3974 \pm 0.0111$ |
| CBraMod (4s) | $0.5448 \pm 0.0112$ | $0.4938 \pm 0.0132$ | $\underline{0.5541} \pm 0.0125$ | $\underline{0.4086} \pm 0.0136$ | $0.2539 \pm 0.0171$ | $0.4097 \pm 0.0126$ |
| CBraMod (8s) | $\underline{0.5453} \pm 0.0092$ | $\underline{0.4950} \pm 0.0119$ | $0.5534 \pm 0.0108$ | $0.4083 \pm 0.0144$ | $\mathbf{0.2573} \pm 0.0186$ | $\mathbf{0.4112} \pm 0.0131$ |
| CBraMod (30s) | $\mathbf{0.5509} \pm 0.0089$ | $\mathbf{0.5041} \pm 0.0122$ | $\mathbf{0.5618} \pm 0.0093$ | $\mathbf{0.4091} \pm 0.0097$ | $\underline{0.2569} \pm 0.0143$ | $\underline{0.4101} \pm 0.0108$ |

| | PhysioNet-MI, 4-class | | | SHU-MI, 2-class | | |
|---|---|---|---|---|---|---|
| Methods | Balanced Accuracy | Cohen's Kappa | Weighted F1 | Balanced Accuracy | AUC-PR | AUROC |
| BIOT | $0.6153 \pm 0.0154$ | $0.4875 \pm 0.0272$ | $0.6158 \pm 0.0197$ | $0.6179 \pm 0.0183$ | $0.6770 \pm 0.0119$ | $0.6609 \pm 0.0127$ |
| LaBraM | $0.6173 \pm 0.0122$ | $0.4912 \pm 0.0192$ | $0.6177 \pm 0.0141$ | $0.6166 \pm 0.0192$ | $0.6761 \pm 0.0083$ | $0.6604 \pm 0.0091$ |
| CBraMod (4s) | $0.6360 \pm 0.0113$ | $0.5149 \pm 0.0185$ | $0.6374 \pm 0.0128$ | $\underline{0.6340} \pm 0.0196$ | $0.7089 \pm 0.0110$ | $\underline{0.6951} \pm 0.0105$ |
| CBraMod (8s) | $\underline{0.6392} \pm 0.0104$ | $\underline{0.5189} \pm 0.0198$ | $\underline{0.6398} \pm 0.0096$ | $0.6338 \pm 0.0182$ | $\underline{0.7098} \pm 0.0105$ | $0.6946 \pm 0.0089$ |
| CBraMod (30s) | $\mathbf{0.6417} \pm 0.0091$ | $\mathbf{0.5222} \pm 0.0169$ | $\mathbf{0.6427} \pm 0.0100$ | $\mathbf{0.6370} \pm 0.0151$ | $\mathbf{0.7139} \pm 0.0088$ | $\mathbf{0.6988} \pm 0.0068$ |

We conduct an experiment to further evaluate the impact of segment length. Specifically, we pre-trained CBraMod in 4s or 8s segment length on our pre-training dataset and compared the results with BIOT, LaBraM and CBraMod (30s). On most datasets, CBraMod (4s or 8s) performs slightly worse than CBraMod (30s), but significantly better than BIOT and LaBraM. It indicates that the segment length has an impact on performance improvement, but the effect is relatively minor.

## I  ABLATION STUDY ON TIME-DOMAIN AND FREQUENCY-DOMAIN SIGNALS

Table 17: The results of ablation study on time-domain and frequency-domain signals.

| | FACED, 9-class | | | SEED-V, 5-class | | |
|---|---|---|---|---|---|---|
| Settings | Balanced Accuracy | Cohen's Kappa | Weighted F1 | Balanced Accuracy | Cohen's Kappa | Weighted F1 |
| Combining | $\mathbf{0.5509} \pm 0.0089$ | $\mathbf{0.5041} \pm 0.0122$ | $\mathbf{0.5618} \pm 0.0093$ | $\mathbf{0.4091} \pm 0.0097$ | $\mathbf{0.2569} \pm 0.0143$ | $\mathbf{0.4101} \pm 0.0108$ |
| Time-domain | $0.5435 \pm 0.0078$ | $0.4956 \pm 0.0134$ | $0.5531 \pm 0.0085$ | $0.4078 \pm 0.0134$ | $0.2548 \pm 0.0204$ | $0.4077 \pm 0.0123$ |
| Frequency-domain | $0.5205 \pm 0.0157$ | $0.4652 \pm 0.0241$ | $0.5218 \pm 0.0143$ | $0.3987 \pm 0.0184$ | $0.2356 \pm 0.0315$ | $0.3975 \pm 0.0201$ |

| | PhysioNet-MI, 4-class | | | SHU-MI, 2-class | | |
|---|---|---|---|---|---|---|
| Settings | Balanced Accuracy | Cohen's Kappa | Weighted F1 | Balanced Accuracy | AUC-PR | AUROC |
| Combining | $\mathbf{0.6417} \pm 0.0091$ | $\mathbf{0.5222} \pm 0.0169$ | $\mathbf{0.6427} \pm 0.0100$ | $\mathbf{0.6370} \pm 0.0151$ | $\mathbf{0.7139} \pm 0.0088$ | $\mathbf{0.6988} \pm 0.0068$ |
| Time-domain | $0.6324 \pm 0.0097$ | $0.5108 \pm 0.0175$ | $0.6333 \pm 0.0086$ | $0.6302 \pm 0.0145$ | $0.7015 \pm 0.0104$ | $0.6879 \pm 0.0094$ |
| Frequency-domain | $0.6158 \pm 0.0076$ | $0.4898 \pm 0.0138$ | $0.6149 \pm 0.0094$ | $0.6181 \pm 0.0167$ | $0.6817 \pm 0.0125$ | $0.6659 \pm 0.0113$ |

CBraMod combines both time-domain and frequency-domain signals to learn generic representation. Here, we conduct an ablation study to verify the effectiveness of time-domain and frequency-domain signals. The results are shown in Table 17. It is obvious that combining time-domain and frequency-domain signals achieves a better performance compared to only using time-domain features or frequency-domain signals, indicating that both time-domain and frequency-domain signals are important for learning EEG representations. Notably, only using time-domain signals performs significantly better compared to frequency-domain signals, and slightly worse compared to combining time-domain and frequency-domain signals. The reason could be that time-domain signals contain frequency-domain information, which can be implicitly captured by neural networks. However, frequency-domain signals lack time-domain information due to the FFT process, leading to a decrease in performance.

## J  ABLATION STUDY ON FINE-TUNING

In this section, we conduct an ablation study on fine-tuning to explore the impact of fine-tuning. The experiment is designed as follows: 1) CBraMod: adjusting all parameters of CBraMod during training on downstream datasets; 2) CBraMod (fixed): fixing the pre-trained parameters of CBraMod and only adjusting the parameters of the classifier during training on downstream datasets. 3) BIOT (fixed): fixing the pre-trained parameters of BIOT and only adjusting the parameters of the classifier

Table 18: The results of ablation study on fine-tuning. **Bold** indicates the best. Underline indicates the second best.

| | FACED, 9-class | | | SEED-V, 5-class | | |
|---|---|---|---|---|---|---|
| Settings | Balanced Accuracy | Cohen's Kappa | Weighted F1 | Balanced Accuracy | Cohen's Kappa | Weighted F1 |
| CBraMod | **0.5509** ± 0.0089 | **0.5041** ± 0.0122 | **0.5618** ± 0.0093 | **0.4091** ± 0.0097 | **0.2569** ± 0.0143 | **0.4101** ± 0.0108 |
| CBraMod (Fixed) | 0.3146 ± 0.0346 | 0.2579 ± 0.0542 | 0.3077 ± 0.0298 | 0.2536 ± 0.0257 | 0.0842 ± 0.0384 | 0.2568 ± 0.0275 |
| BIOT (Fixed) | 0.2775 ± 0.0318 | 0.1839 ± 0.0512 | 0.2599 ± 0.0271 | 0.2461 ± 0.0287 | 0.0798 ± 0.0361 | 0.2489 ± 0.0257 |
| LaBraM (Fixed) | 0.3004 ± 0.0458 | 0.2377 ± 0.0617 | 0.2943 ± 0.0375 | 0.2521 ± 0.0267 | 0.0854 ± 0.0342 | 0.2543 ± 0.0265 |
| | PhysioNet-MI, 4-class | | | SHU-MI, 2-class | | |
| Settings | Balanced Accuracy | Cohen's Kappa | Weighted F1 | Balanced Accuracy | AUC-PR | AUROC |
| CBraMod | **0.6417** ± 0.0091 | **0.5222** ± 0.0169 | **0.6427** ± 0.0100 | **0.6370** ± 0.0151 | **0.7139** ± 0.0088 | **0.6988** ± 0.0068 |
| CBraMod (Fixed) | 0.3845 ± 0.0345 | 0.2983 ± 0.0498 | 0.3946 ± 0.0378 | 0.5217 ± 0.0247 | 0.5304 ± 0.0253 | 0.5238 ± 0.0317 |
| BIOT (Fixed) | 0.3698 ± 0.0371 | 0.2703 ± 0.0472 | 0.3723 ± 0.0364 | 0.5123 ± 0.0206 | 0.5215 ± 0.0197 | 0.5146 ± 0.0264 |
| LaBraM (Fixed) | 0.3715 ± 0.0432 | 0.2814 ± 0.0586 | 0.3796 ± 0.0472 | 0.5233 ± 0.0272 | 0.5329 ± 0.0284 | 0.5218 ± 0.0275 |

during training on downstream datasets. 4) LaBraM (fixed): fixing the pre-trained parameters of LaBraM and only adjusting the parameters of the classifier during training on downstream datasets. The results are shown in Table 18. Obviously, fixing the pre-trained parameters during training on downstream datasets will lead to a very large performance decline. It indicates that CBraMod cannot currently serve as a fixed-parameter feature extractor like CLIP (Radford et al., 2021) and SAM (Kirillov et al., 2023), and fine-tuning is still necessary. Notably, CBraMod (fixed) outperforms BIOT (fixed) and LaBraM (fixed), indicating that CBraMod has better generalizability on unseen datasets compared existing methods on the fixing setting.

## K    LOW-RESOURCE COMPARISON WITH EXISTING METHODS

Table 19: Performance comparison on low-resource settings with 30% of fine-tuning data. **Bold** indicates the best. Underline indicates the second best.

| | FACED, 9-class | | | SEED-V, 5-class | | |
|---|---|---|---|---|---|---|
| Methods | Balanced Accuracy | Cohen's Kappa | Weighted F1 | Balanced Accuracy | Cohen's Kappa | Weighted F1 |
| CBraMod (full data) | **0.5509** ± 0.0089 | **0.5041** ± 0.0122 | **0.5618** ± 0.0093 | **0.4091** ± 0.0097 | **0.2569** ± 0.0143 | **0.4101** ± 0.0108 |
| CBraMod (30%) | 0.4035 ± 0.0233 | 0.3239 ± 0.0265 | 0.4056 ± 0.0256 | 0.3877 ± 0.0236 | 0.2291 ± 0.0246 | 0.3886 ± 0.0255 |
| BIOT (30%) | 0.3428 ± 0.0329 | 0.2573 ± 0.0346 | 0.3501 ± 0.0341 | 0.3505 ± 0.0375 | 0.1775 ± 0.0425 | 0.3492 ± 0.0416 |
| LaBraM (30%) | 0.3513 ± 0.0315 | 0.2672 ± 0.0371 | 0.3548 ± 0.0325 | 0.3686 ± 0.0305 | 0.2044 ± 0.0384 | 0.3700 ± 0.0321 |
| | PhysioNet-MI, 4-class | | | SHU-MI, 2-class | | |
| Methods | Balanced Accuracy | Cohen's Kappa | Weighted F1 | Balanced Accuracy | AUC-PR | AUROC |
| CBraMod (full data) | **0.6417** ± 0.0091 | **0.5222** ± 0.0169 | **0.6427** ± 0.0100 | **0.6370** ± 0.0151 | **0.7139** ± 0.0088 | **0.6988** ± 0.0068 |
| CBraMod (30%) | 0.5613 ± 0.0162 | 0.4150 ± 0.0267 | 0.5621 ± 0.0184 | 0.6231 ± 0.0198 | 0.6901 ± 0.0134 | 0.6754 ± 0.0105 |
| BIOT (30%) | 0.5189 ± 0.0312 | 0.3477 ± 0.0371 | 0.5201 ± 0.0308 | 0.5419 ± 0.0345 | 0.6260 ± 0.0273 | 0.6021 ± 0.0301 |
| LaBraM (30%) | 0.5269 ± 0.0237 | 0.3598 ± 0.0342 | 0.5288 ± 0.0225 | 0.5636 ± 0.0289 | 0.6519 ± 0.0227 | 0.6467 ± 0.0274 |

In practical applications, obtaining labeled data for brain-computer interface (BCI) systems or clinical studies often demands substantial time and financial resources. This limitation highlights the critical need for developing and investigating EEG foundational models, which can reduce reliance on extensive labeled datasets. To illustrate the practical utility of our approach, we compare CBraMod with existing foundational models across multiple datasets with limited labeled data availability. Specifically, we fine-tune CBraMod, BIOT, and LaBraM using 30% of labeled data. The results are presented in Table 19. The results clearly demonstrate that CBraMod consistently outperforms existing foundational models in low-resource settings across these datasets. This observation underscores CBraMod's superior capability to effectively capture and leverage generic EEG representations, even when the availability of labeled data is limited. Importantly, when we compare the low-resource results with the results of full-data fine-tuning, we observe that the performance of CBraMod in low-resource scenarios is only marginally lower than the full-data performance when trained on SEED-V and SHU-MI. This finding further highlights the practical utility of CBraMod, showcasing its ability to sustain strong performance even with limited labeled data. This empirically demonstrates that our model can effectively mitigate the challenges posed by limited downstream task training data to some extent, offering significant advantages for real-world applications where labeled data is often scarce.

## L  COMPARISON ON SPLIT RATIO OF CRISS-CROSS ATTENTION

Table 20: Performance comparison on split ratio of criss-cross attention.

| | FACED, 9-class | | | SEED-V, 5-class | | |
|---|---|---|---|---|---|---|
| Settings | Balanced Accuracy | Cohen's Kappa | Weighted F1 | Balanced Accuracy | Cohen's Kappa | Weighted F1 |
| CBraMod (2:6) | 0.5405 ± 0.0096 | 0.4921 ± 0.0134 | 0.5526 ± 0.0106 | 0.4046 ± 0.0105 | 0.2438 ± 0.0155 | 0.4062 ± 0.0101 |
| CBraMod (4:4) | **0.5509** ± 0.0089 | **0.5041** ± 0.0122 | **0.5618** ± 0.0093 | **0.4091** ± 0.0097 | **0.2569** ± 0.0143 | **0.4101** ± 0.0108 |
| CBraMod (6:2) | 0.5413 ± 0.0124 | 0.4910 ± 0.0145 | 0.5531 ± 0.0128 | 0.4033 ± 0.0093 | 0.2410 ± 0.0166 | 0.4051 ± 0.0119 |
| | **PhysioNet-MI, 4-class** | | | **SHU-MI, 2-class** | | |
| Settings | Balanced Accuracy | Cohen's Kappa | Weighted F1 | Balanced Accuracy | AUC-PR | AUROC |
| CBraMod (2:6) | 0.6253 ± 0.0105 | 0.4995 ± 0.0176 | 0.6271 ± 0.0129 | 0.6205 ± 0.0170 | 0.6904 ± 0.0090 | 0.6751 ± 0.0088 |
| CBraMod (4:4) | **0.6417** ± 0.0091 | **0.5222** ± 0.0169 | **0.6427** ± 0.0100 | **0.6370** ± 0.0151 | **0.7139** ± 0.0088 | **0.6988** ± 0.0068 |
| CBraMod (6:2) | 0.6268 ± 0.0102 | 0.5013 ± 0.0188 | 0.6291 ± 0.0104 | 0.6210 ± 0.0139 | 0.6897 ± 0.0115 | 0.6764 ± 0.0072 |

Our criss-cross transformer models spatial and temporal dependencies separately through two parallel attention mechanisms: spatial and temporal attention. It splits the 8 heads of the intermediate layer embeddings into two equal parts (spatial heads : temporal heads = 4:4), which are then processed by the spatial and temporal attention mechanisms, respectively, ensuring the model assigns equal emphasis to spatial and temporal dependencies. In this section, we adjust the split ratio of heads and observe how the decoding performance on downstream tasks varies when the model prioritizes spatial dependencies or temporal dependencies. The experimental results are shown in Table 20. Evidently, the model's performance shows a noticeable decline when it prioritizes either spatial (6:2) or temporal (2:6) dependencies over treating them equally. This indicates that spatial and temporal dependencies are equally important in EEG representation learning, providing empirical evidence for the validity of our criss-cross attention mechanism.

## M  MASK RATIO ANALYSIS

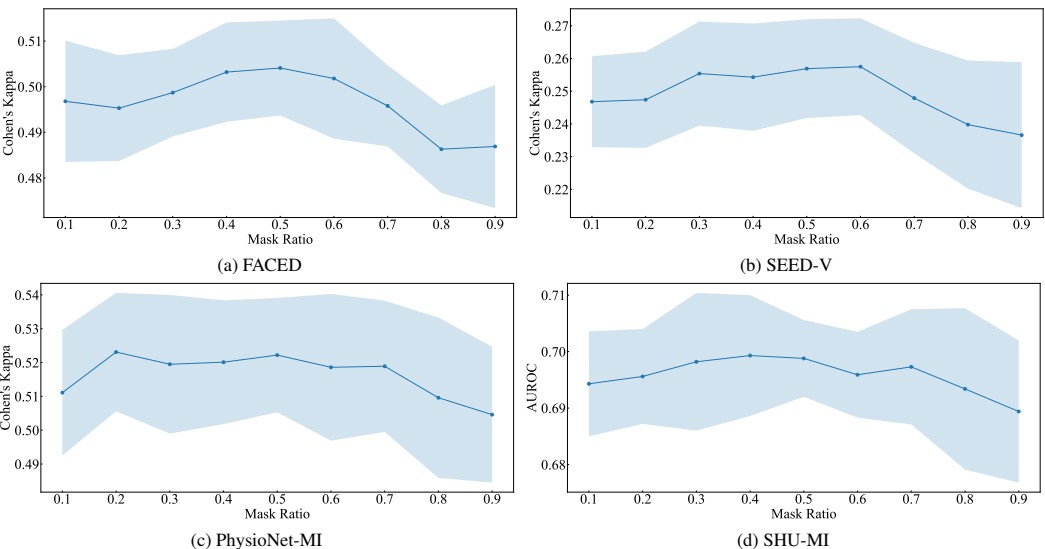

(a) FACED

(b) SEED-V

(c) PhysioNet-MI

(d) SHU-MI

Figure 9: Performance comparison on mask ratio.

In this section, we conduct comparison experiment on mask ratio to explore its impact on performance of CBraMod on downstream datasets. The results are as shown in Figure 9. CBraMod usually achieves a better performance when the range of mask ratio is 0.3 to 0.7. On FACED, CBraMod performs the best when mask ratio is set to 0.5, shown in Figure 9(a). As shown in Figure 9(b), CBraMod achieves the best performance on SEED-V as mask ratio is 0.6 and 0.5 mask ratio follows 0.6. In Figure 9(c) we can see that 0.2 mask ratio is the best on PhysioNet-MI, and 0.5 mask ratio perform the second best. Finally, on SHU-MI, 0.4 is the best mask ratio and performance of 0.5 mask ratio is close to 0.4

mask ratio, shown in Figure 9(d). In summary, 0.5 mask ratio is a very appropriate choice to achieve good performance on multiple downstream datasets.

# N  MASK TOKEN COMPARISON

Table 21: The results of mask token comparison.

| | FACED, 9-class | | | SEED-V, 5-class | | |
|---|---|---|---|---|---|---|
| Methods | Balanced Accuracy | Cohen's Kappa | Weighted F1 | Balanced Accuracy | Cohen's Kappa | Weighted F1 |
| Full-zero token | $0.5509 \pm 0.0089$ | $\mathbf{0.5041} \pm 0.0122$ | $0.5618 \pm 0.0093$ | $\mathbf{0.4091} \pm 0.0097$ | $\mathbf{0.2569} \pm 0.0143$ | $\mathbf{0.4101} \pm 0.0108$ |
| Learnable token | $\mathbf{0.5527} \pm 0.0096$ | $0.5033 \pm 0.0131$ | $\mathbf{0.5627} \pm 0.0094$ | $0.4078 \pm 0.0105$ | $0.2517 \pm 0.0148$ | $0.4089 \pm 0.0123$ |

| | PhysioNet-MI, 4-class | | | SHU-MI, 2-class | | |
|---|---|---|---|---|---|---|
| Methods | Balanced Accuracy | Cohen's Kappa | Weighted F1 | Balanced Accuracy | AUC-PR | AUROC |
| Full-zero token | $\mathbf{0.6417} \pm 0.0091$ | $\mathbf{0.5222} \pm 0.0169$ | $0.6427 \pm 0.0100$ | $\mathbf{0.6370} \pm 0.0151$ | $0.7139 \pm 0.0088$ | $0.6988 \pm 0.0068$ |
| Learnable token | $0.6399 \pm 0.0112$ | $0.5211 \pm 0.0215$ | $\mathbf{0.6433} \pm 0.0175$ | $0.6349 \pm 0.0142$ | $\mathbf{0.7156} \pm 0.0101$ | $\mathbf{0.7012} \pm 0.0096$ |

In this section, we compare the performance of full-zero mask token and learnable mask token of explore the effectiveness of mask token type. The experiment is designed as follows: 1) Full-zero token: utilizing a full-zero vector with the same dimension as patch embeddings to mask the EEG patches; 2) Learnable token: using a vector whose parameters are learnable to mask the EEG patches. The results of mask token comparison are presented in Table 21. There is no significant performance difference between full-zero and learnable mask token, indicating that their capabilities are similar.

# O  PARAMETERS AND FLOPS COMPARISON

Table 22: Parameters and FLOPs comparison on CHB-MIT (16 channels, 10 seconds).

| Methods | Params | FLOPs |
|---|---|---|
| EEGNet | 0.003M | 8.9M |
| Conformer | 0.55M | 29.6M |
| SPaRCNet | 0.79M | 65.7M |
| ContraWR | 1.6M | 66.4M |
| CNN-Transformer | 3.2M | 79.1M |
| FFCL | 2.4M | 209.9M |
| ST-Transformer | 3.5M | 42.1M |
| BIOT | 3.2M | 483.3M |
| LaBraM-Base | 5.8M | 483.0M |
| LaBraM-Large | 46M | 3.06G |
| LaBraM-Huge | 369M | 22.8G |
| CBraMod (full attention) | 4.1M | 469.4M |
| CBraMod (axial attention) | 4.0M | 334.7M |
| CBraMod (attention of CCNet) | 4.0M | 354.6M |
| CBraMod (criss-cross attention) | 4.0M | 318.9M |

In this section, taking the CHB-MIT dataset as an example, we compare the parameter counts and FLOPs of CBraMod with existing methods and CBraMod with other attention mechanism. The numbers of parameters in baselines are provided by LaBraM (Jiang et al., 2024), and others are calculated by Thop[3]. The results are shown in Table 22. The parameters and FLOPs of foundation models are more than the non-foundation-model baselines because the foundation model usually need to be large for learning generic representations. LaBraM-Large and LaBraM-Huge achieve a good performance on some downstream datasets, but their parameters and FLOPs are significantly more than CBraMod. CBraMod has fewer FLOPs compared to the foundation models with similar parameter counts, BIOT and LaBraM-Base. Moreover, the CBraMod based on criss-cross attention has the fewest FLOPs compared to other attention mechanism. It indicates that CBraMod achieves lower computational complexity by our criss-cross EEG modeling.

---

[3]https://github.com/Lyken17/pytorch-OpCounter

# P  INTERPRETABILITY ANALYSIS

## P.1  TOPOGRAPHY VISUALIZATION

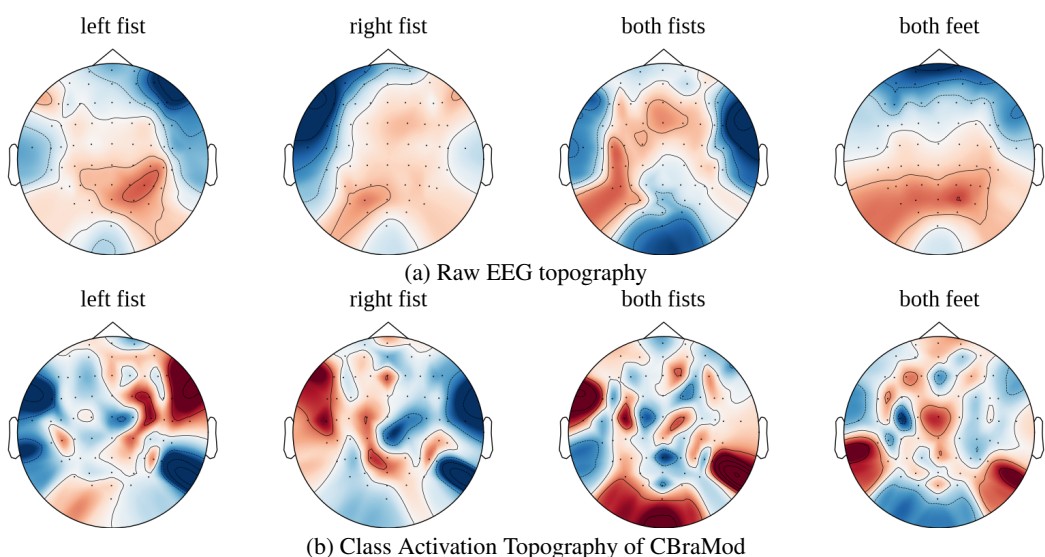

(a) Raw EEG topography

(b) Class Activation Topography of CBraMod

Figure 10: Topography visualization on motor imagery classification (PhysioNet-MI).

In this section, we provide a topography visualization on motor imagery classification. The setup of visualization analysis are as follows: **Raw EEG topography**: We computed the energy intensity of the raw EEG signals for each channel and visualized the results using a topographic map. **Class Activation Topography of CBraMod**: We utilized Grad-CAM (Gradient-weighted Class Activation Mapping) (Selvaraju et al., 2017) to compute the contribution of each channel in the learned representations of CBraMod to the classification outcomes, visualizing the results as a Class Activation Topography. The visualization results are shown in Figure 10. Electrodes related to left and right fist movements exhibit symmetric patterns, while bilateral electrodes are linked to both fists and both feet movements, with distinct channels corresponding to each of the four classes. Compared to the raw EEG signals, the representations learned by CBraMod exhibit more pronounced differences in the importance assigned to various channels for different classes.

## P.2  REPRESENTATION VISUALIZATIONS ON DOWNSTREAM DATASETS

In this section, we provide a representation visualization analysis on downstream datasets using UMAP (McInnes et al., 2018) dimensionality reduction. The experimental setup is as follows: 1) **Raw EEG sample of a downstream dataset**: we directly visualize the raw EEG sample from a downstream dataset. 2) **CBraMod (w/o fine-tuning) on a downstream dataset**: we visualize the representations of the pre-trained CBraMod without fine-tuning on the test set of a downstream dataset. 3) **CBraMod (w/ fine-tuning) on a downstream dataset**: we fine-tune the pre-trained CBraMod on the training set of a downstream dataset and visualize its representations on the test set. The visualization results are as shown in Figure 11. It is evident that the representation distributions in Figure 11(b) and (e) exhibit better clustering effects compared to the distribution of the raw EEG samples. It indicates that pre-training enables CBraMod to learn generic EEG representations, allowing it to capture label-related features of downstream datasets to some extent, even without fine-tuning. Additionally, the representation distributions in Figure 11(c) and (f) also exhibit better clustering effects compared to the representations distribution without fine-tuning. It reflects the role of fine-tuning on the downstream data.

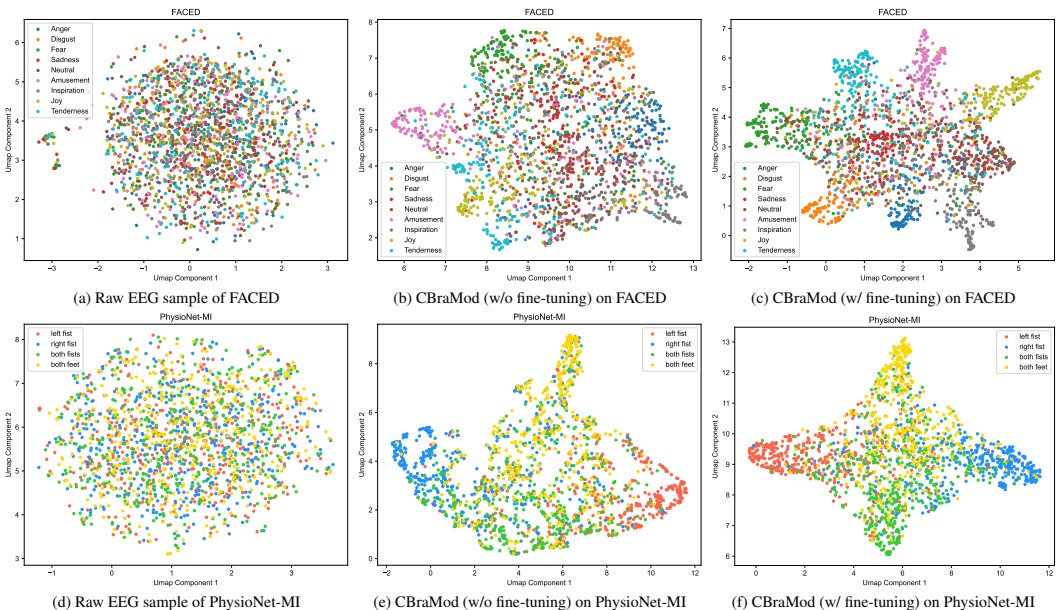

Figure 11: Representation visualizations on downstream datasets.

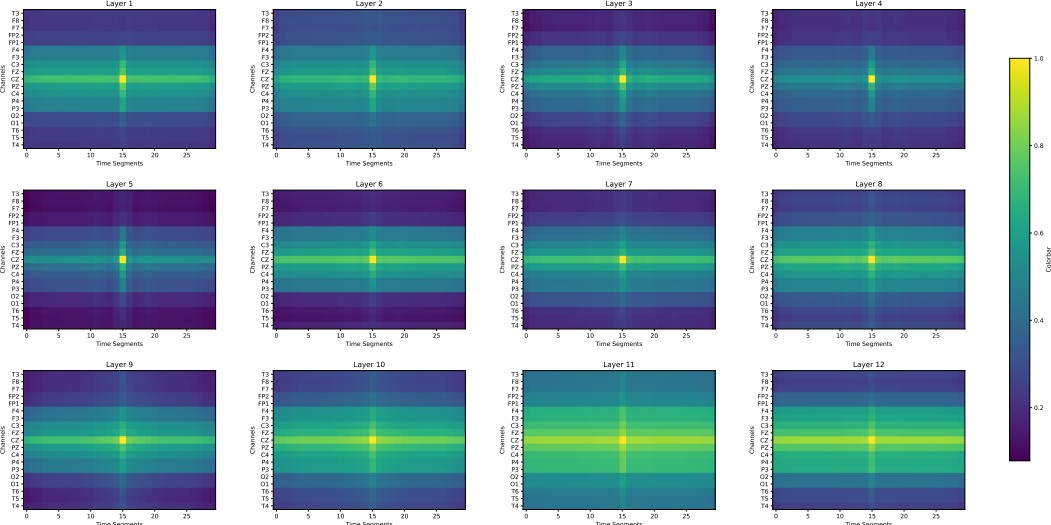

Figure 12: Visualization of patch relationships from each criss-cross transformer layer.

## P.3 VISUALIZATION OF PATCH RELATIONSHIPS FROM EACH CRISS-CROSS TRANSFORMER LAYER

In this section, we provide a visualization analysis for criss-cross EEG modeling. Specifically, we feed EEG samples (19 channels × 30 seconds) of TUEG into the pre-trained CBraMod to obtain the output of each criss-cross transformer layer, $E_m \in \mathbb{R}^{C \times n \times d}$, where $m \in [1, 2, ..., 12]$ is the layer number, $C = 19$ is the number of channels, $n = 30$ is the number of time segments and $d = 200$ is the dimension of patch embedding. Thus, $E_m$ consists of $19 \times 30$ EEG patches. Without loss of generality, we select the 16-th patch of the Cz channel as the central patch, then calculate the correlation coefficient between the embedding of this patch and the embeddings of all other patches. The results are visualized in the form of a heatmap, as shown in the Figure 12. In the Figure 12, the channels are arranged in a sequence corresponding to the anterior brain, Cz, and posterior brain regions. It is evident that the correlation coefficients between the other patches and

the central patch exhibit a criss-cross shape. It indicates that the proposed method has successfully learned the criss-cross spatial-temporal dependencies pattern between patches in the EEG signal. Furthermore, in the deeper transformer layers, we observe that the associations between all patches and the central patch are stronger. This suggests that the criss-cross transformer is also capable of learning non-criss-cross dependencies through its multi-layered structure. All of these visualizations provide interpretability for the criss-cross EEG modeling approach we propose.

## Q    A LEAVE-ONE-SUBJECT-OUT COMPARISON WITH EEG-SIMPLECONV ON BCIC-IV-2A

Table 23: The results of the LOSO comparison with EEG-SimpleConv (BCIC-IV-2a, 4-class).

| Methods | Balanced Accuracy | AUC-PR | AUROC |
|---|---|---|---|
| EEG-SimpleConv (w/o Key Ingredients) | $0.5650 \pm 0.0989$ | $0.4201 \pm 0.1319$ | $0.5484 \pm 0.1047$ |
| CBraMod (w/o Key Ingredients) | $\mathbf{0.5968} \pm 0.0816$ | $\mathbf{0.4558} \pm 0.1088$ | $\mathbf{0.5889} \pm 0.0875$ |
| EEG-SimpleConv (w/ Key Ingredients) | $0.7221 \pm 0.0768$ | $0.5765 \pm 0.1069$ | $0.6977 \pm 0.0918$ |
| CBraMod (w/ Key Ingredients) | $\mathbf{0.7405} \pm 0.0635$ | $\mathbf{0.5997} \pm 0.0833$ | $\mathbf{0.7195} \pm 0.0682$ |

Specifically, we included the performance comparison with the strong baseline, EEG-SimpleConv (El Ouahidi et al., 2024), under the leave-one-subject-out (LOSO) protocol on BCIC-IV-2a. In EEG-SimpleConv paper, the reported performance of $72.1 \pm 7.3$ accuracy under the leave-one-subject-out (LOSO) protocol relies heavily on several advanced preprocessing and training techniques referred to as "Key Ingredients," including Euclidean Alignment (EA), session statistics, Mixup, and subject-wise regularization. When these techniques are not employed, EEG-SimpleConv achieves significantly lower performance: $56.4 \pm 9.0$ accuracy, reported in the original paper.

To provide a more rigorous and fair evaluation, we conducted experiments under the same LOSO protocol used by EEG-SimpleConv, both with and without incorporating the "Key Ingredients." The results are summarized as shown in Table 23. These results demonstrate that: 1) Our reproduction of EEG-SimpleConv matches the original paper's findings. 2) CBraMod consistently outperforms EEG-SimpleConv under both settings, achieving higher balanced accuracy, Cohen's Kappa, and weighted F1 scores while also demonstrating lower inter-subject variance, highlighting its robustness and effectiveness.

## R    COMPARISON ON CONVERGENCE SPEED WITH SUPERVISED MODEL

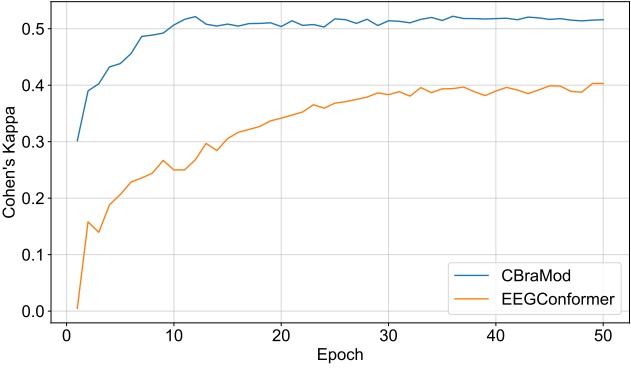

Figure 13: Performance curves comparison for downstream task fine-tuning in FACED.

In this section, we compared the performance curves of our method with existing supervised learning models during downstream task training (using EEGConformer on the FACED dataset as an example). Specifically, we fine-tuned the pretrained CBraMod on the FACED training set, testing its performance on the validation set at the end of each epoch. We then plotted the performance variation curve and compared it with the performance curve of EEGConformer. The results are shown in Figure 13.

Obviously, CBraMod can achieve a decent result within the first epoch and converge within 10 epochs. However, EEGConformer achieves a Cohen's Kappa result close to 0 in the first epoch, and it only reaches convergence after approximately 30 epochs. These results demonstrate that our pretrained model achieves faster convergence on downstream tasks, further proving that our method is capable of learning generic EEG representations that can effectively adapt to downstream tasks.

## S DISCUSSION

### S.1 IMPLICATION

**Novel Insights for EEG Foundation Model Construction**     The proposed criss-cross EEG modeling strategy and asymmetric conditional positional encoding (ACPE) scheme offer profound insights into the construction of EEG foundation models. Traditional EEG modeling approaches often overlook the unique structural characteristics of EEG signals, which exhibit heterogeneous spatial and temporal dependencies. By devising a criss-cross transformer that models these dependencies in parallel, our approach captures the intricate relationships within EEG data more effectively. This strategy not only enhances the model's ability to generalize across diverse EEG formats but also provides a more nuanced understanding of the underlying brain dynamics. The ACPE scheme further augments this capability by dynamically encoding positional information, making the model adaptable to varying channel configurations and reference contexts. This flexibility is crucial for EEG foundation models, as it allows them to be applied across a wide range of clinical and BCI applications. The success of our approach underscores the importance of tailored modeling strategies for EEG data, setting a new baseline for future research in this domain.

**Real-World BCI System Development**     The exploration of EEG foundation models, as exemplified by our work, holds significant promise for the development of practical BCI systems, particularly in the context of universal BCI systems. By leveraging large-scale pre-training on large EEG corpora, our model can learn generic representations that are robust and adaptable to various downstream tasks. This capability is essential for building BCI systems that can be deployed across different user populations and clinical settings, thereby enhancing their accessibility and utility. Moreover, the strong generalization ability of our model reduces the dependency on task-specific labeled data, which is often scarce and expensive to obtain. This is particularly beneficial in real-world applications where data collection can be challenging and resource-intensive. The ability to fine-tune the model on limited data while maintaining high performance is a critical step towards the realization of universal BCI systems that can be widely adopted in clinical practice.

In summary, the novel modeling strategies and the demonstrated generalizability of our EEG foundation model provide valuable insights and practical benefits for the development of advanced BCI systems. These advancements not only push the boundaries of current EEG decoding techniques but also pave the way for more inclusive and effective brain-computer interfaces in the future.

### S.2 LIMITATION

Our work still has some limitations. Firstly, TUEG corpus is a very large dataset with a high amount of dirty data. We employed a rather crude approach to filter out clean data for pre-training, which helped address the issue of having a high amount of dirty data. However, it also resulted in a significant reduction in the amount of available pre-training data. Secondly, our method has achieved success on multiple downstream datasets and has lower computational complexity compared to other EEG Foundation models, but it still has a higher parameter count and computational complexity compared to non-foundation models. The high parameter count and computational complexity result in a higher usage threshold for EEG foundation models and makes it difficult to deploy them on devices with lower computational power. Next, due to limited computational resources, we have not yet analyzed the potential scaling laws for EEG pre-training in a larger scale (e.g. billion level). Finally, large models have achieved tremendous success in fields such as vision and language. However, there is still a lack of sufficient exploration on how to utilize these pre-trained large models in other fields for understanding EEG signals and various other brain signals.

## S.3 FUTURE WORK

Giving the above limitations, our future works are as follows: 1) A larger and cleaner EEG corpus is essential to train a better EEG foundation model. We will collect more EEG corpus and explore more automated and effective data preprocessing methods for learning more powerful representations. 2) We will explore ways to develop an effective and efficient EEG foundation model. For example, we can directly train a smaller EEG foundation model or utilize techniques such as knowledge distillation to obtain a smaller EEG foundation model from a pre-trained large EEG foundation model. 3) We will explore using larger pretraining datasets and model size, to further enhance the performance of the EEG foundation model and analyze the potential scaling laws for EEG pre-training. 4) We will explore establishing connections between large models in other fields (e.g. vision and language) and brain signals, utilizing the knowledge from these models to decode brain signals. It may involve techniques such as transfer learning.

