# OpenReview forum: "CBraMod: A Criss-Cross Brain Foundation Model for EEG Decoding"
_ICLR.cc/2025/Conference — ICLR 2025 Poster_

### Official Review · Reviewer_iYYg · 2024-10-31

**Soundness:** 3
**Presentation:** 3
**Contribution:** 3
**Rating:** 8
**Confidence:** 5

**Summary:**

* This paper proposes a new LLM model for EEG decoding, primarily trained on the Temple University EEG (TUEG) dataset and tested across different EEG decoding datasets.
* The authors introduce a new way to encode EEG data with various channel configurations, using asymmetric conditional positional dynamic encoding inspired by Chu et al. (2021) with different kernel sizes. They also present a new criss-cross transformer layer with temporal and spatial attention, improving upon the work of Huang et al. (2020) and adapting it effectively to the EEG context.
* The encoder's contribution lies in combining the BioT encoder for inputting frequency information via SFFT and the Labram temporal encoder utilizing Conv2D and GroupNorm.
* The pre-training strategy involves masked reconstruction with random masking of EEG patches instead of masking temporal and spatial components.
* The method is showcased on 12 datasets, spanning a variety of EEG decoding tasks.

**Strengths:**

* Although the general idea of using language models for EEG data is not new, the neural network architecture presents some novelty by incorporating and combining multiple ideas from previous works, such as Patch from Visual Transformer, Criss-Cross Transformer, and Asymmetric Conditional Positional with different kernel sizes.
* This work presents a large number of experimental results on 12 different datasets and tasks.
* Some of the experimental results are quite encouraging and show incremental results over the Labram and BioT works.
* The use of large language models in the EEG field is relatively under-explored, especially considering the revolution we are experiencing in parallel fields such as audio, NLP, and vision. Bridging the bridge connecting other communities to the EEG community is very refreshing!

**Weaknesses:**

**Originality**

While the particular combination of methods is novel, it primarily combines existing ideas from other deep learning subfields. This limits the overall novelty of the work, especially given the lack of sufficient ablations to discern which of the numerous components is truly important. To be very clear, there is no problem using components from other fields; in my opinion, this is highly encouraged, but the study needs to go in-depth to understand what leads to compatibility and the "bridge construction."

**Clarity**

In my opinion, the paper's writing could be improved. The text needs refinement to better guide the reader through the findings rather than just describing tables. There is no discussion about other papers; while the engineering contributions are clear, the scientific perspective is significantly lacking.

**Major Concerns**

* The choice of datasets for motor imagery, emotion, and sleep stage tasks seems arbitrary. For instance, in the case of motor imagery, the most widely used dataset is BNCI 2014 version 004, and the largest is from Stieger et al. (2021) to the best of my knowledge. However, the authors chose Physionet-MI.
*  The results in Physionet-MI are significantly below the state-of-the-art. A quick search reveals accuracies of 88.6% ± 9.0 using EEGSym, 86.36% with Zoumpourlis et al. (2023), and 73.60% with TIDNet on unseen subjects, where we have 64.17%. While I understand that the exact reproduction of results in EEG decoding is impossible, the reported results here are more than 20% below the state-of-the-art. This concern extends to sleep stage models, where datasets like SleepEDF+, MASS, or SHHS are standard, and accuracies above 80% are expected. The same issue applies to emotion classification; for example, Zhang et al. (2024) achieved better results on the SEED dataset testing with different train-test splits.

* Furthermore, when critically analyzing the model's usefulness based on the metrics—and considering literature from the field, such as O. Alkob et al. (2018) and Combrisson and Jerbi (2015)—the metric values fall short of demonstrating real usefulness for BCI applications.

* The choice of baselines is inadequate and possibly inherits these shortcomings from the BioT and Labram studies. Although the authors cite well-established works in the literature, there is no comparison with neural networks like EEGNet, ShallowNet, EEGConformer, EEGSym, ATCNet, FBCNet, and many others that effectively capture temporal and spatial information in motor imagery tasks. By using neural networks with only frequency-based encoders (e.g., BioT), the spatial and temporal information critical for motor imagery is not learned. Employing weak baselines for classification tasks makes the gains appear larger than they actually are. This issue also applies to the sleep stage task, where models like USleep, DeepSleepNet, SleepTransformer, XSleepNet, SeqSleepNet, ChambonNet, and others should be considered as baselines.

* Using the TEUV and TUAB datasets for abnormal detection and event-type classification while employing TEUG for pre-training leads to data leakage. This is indicated on the Temple University dataset's own webpage: "The TUH EEG Events Corpus (TUEV: v2.0.1): This corpus is a subset of TUEG."

**References**

O. Alkoby, A. Abu-Rmileh, O. Shriki and D. Todder, "Can we predict who will respond to neurofeedback? A review of the inefficacy problem and existing predictors for successful EEG neurofeedback learning", Neuroscience, vol. 378, pp. 155-164, May 2018.

Combrisson E, Jerbi K. Exceeding chance level by chance: The caveat of theoretical chance levels in brain signal classification and statistical assessment of decoding accuracy. Journal of neuroscience methods. 2015 Jul 30;250:126-36.

Stieger, J. R., Engel, S. A., & He, B. (2021). Continuous sensorimotor rhythm based brain computer interface learning in a large population. Scientific Data, 8(1), 98.

Zhang, Z., Zhong, S. H., & Liu, Y. (2024). TorchEEGEMO: A deep learning toolbox towards EEG-based emotion recognition. Expert Systems with Applications, 249, 123550.

Pérez-Velasco, S., Santamaría-Vázquez, E., Martínez-Cagigal, V., Marcos-Martínez, D., & Hornero, R. (2022). EEGSym: Overcoming inter-subject variability in motor imagery based BCIs with deep learning. IEEE Transactions on Neural Systems and Rehabilitation Engineering, 30, 1766-1775.

Zoumpourlis, G., & Patras, I. (2024). Motor imagery decoding using ensemble curriculum learning and collaborative training. In 2024 12th International Winter Conference on Brain-Computer Interface (BCI) (pp. 1-8). IEEE.

Kostas, D., & Rudzicz, F. (2020). Thinker invariance: enabling deep neural networks for BCI across more people. Journal of Neural Engineering, 17(5), 056008.

**Questions:**

All the questions are pointed out in the Major Concerns section.

---

> ### Author Response · Authors · 2024-11-20
> **Response to Reviewer iYYg (1/n)**
>
> We thank the reviewer for the helpful analysis and constructive feedback.
>
> **W1**: While the particular combination of methods is novel, it primarily combines existing ideas from other deep learning subfields. This limits the overall novelty of the work, especially given the lack of sufficient ablations to discern which of the numerous components is truly important. To be very clear, there is no problem using components from other fields; in my opinion, this is highly encouraged, but the study needs to go in-depth to understand what leads to compatibility and the "bridge construction."
>
> **R**: Thank you for your valuable comment. Our core contributions, Criss-Cross Attention and Asymmetric Conditional Positional Encoding (ACPE), are specifically designed to leverage the unique structure of EEG data and address limitations positional encoding of traditional EEG foundation models.
>
> 1. **Criss-Cross Attention**:
>    - Existing foundation models mostly adopt the full EEG modeling strategy, flattening EEG patches to a sequence and using general transformer to model the dependencies among all EEG patches together, similar to ViT[1] in the field of computer vision. It ignores the unique structural characteristics of EEG signals. EEG signals have different structural characteristics compared to image. It contain heterogeneous spatial and temporal dependencies, but images contain only spatial dependencies. And the dependencies among EEG patches within the same channel or time interval could be stronger than those among patches from different channels and time intervals, but such a prior assumption may not be valid in image.
>    - The criss-cross attention mechanism was developed to better capture and utilize the structural relationships inherent in EEG data. It models heterogeneous spatial and temporal dependencies separately through two parallel attention mechanisms, spatial and temporal attentions. It is not just a general attention mechanism but one that is tailored to the specific characteristics of EEG signals.
>
> 2. **Asymmetric Conditional Positional Encoding**:
>    - Current EEG foundation models use absolute position encoding based on electrode numbers as channel embeddings, as seen in methods like LaBraM[2]. However, EEG channels are not simply equivalent to electrode positions, as they are also influenced by factors such as reference schemes (e.g., ear reference, average reference, REST, and bipolar montages). Therefore, we believe that this traditional position encoding approach limits the adaptability of EEG foundation models to different downstream tasks.
>    - To address this challenge, we proposed an alternative solution with our **ACPE (Asymmetric Conditional Positional Encoding)** inspired by CPE[3]. This method leverages a convolutional network to dynamically learn the spatial relationships between each patch and other patches, allowing the model to better adapt to varying channel configurations in downstream tasks. By doing so, we aim to enhance the flexibility and applicability of EEG foundation models across diverse EEG data formats.
>
> In our revised manuscript, we have enhanced the discussion of these methods, particularly with respect to their motivation and design. Furthermore, in the original version, we already provided compelling evidence of the effectiveness of these innovations through experiments in the "Attention Mechanism Comparison" and "Positional Encoding Comparison" sections.
>
> To further address concerns regarding the interpretability, we have added new interpretability analysis in the revised version **(Line 1638-1750, Page 31-33)**, including **(1) Topography Visualization**, **(2) Representation Visualizations on Downstream Datasets** and **(3) Visualization of Patch Relationships from each Criss-cross Transformer Layer**.
>    - **(1) Topography Visualization** demonstrates that CBraMod effectively captures symmetric patterns between the left and right hemispheres from motor imagery data.
>    - **(2) Representation Visualizations on Downstream Datasets** demonstrates that our pre-trained model can learn representations with clustering effects tailored to downstream tasks.
>    - **(3) Visualization of Patch Relationships from each Criss-cross Transformer Layer** confirms that our criss-cross attention effectively models criss-cross dependencies.
>
> We hope these clarifications, along with the new interpretability analysis, provide a clearer understanding of the novelty and significance of our work. Thank you once again for your feedback.
>
> [1] An image is worth 16x16 words: Transformers for image recognition at scale. arXiv preprint arXiv:2010.11929, 2020.
>
> [2] Large Brain Model for Learning Generic Representations with Tremendous EEG Data in BCI//The Twelfth International Conference on Learning Representations.
>
> [3] Conditional Positional Encodings for Vision Transformers//The Eleventh International Conference on Learning Representations.

---

> > ### Author Response · Authors · 2024-11-20
> > **Response to Reviewer iYYg (2/n)**
> >
> > **W2**: In my opinion, the paper's writing could be improved. The text needs refinement to better guide the reader through the findings rather than just describing tables. There is no discussion about other papers; while the engineering contributions are clear, the scientific perspective is significantly lacking.
> >
> > **R**: Thank you for your insightful feedback. We agree that the presentation of the paper could benefit from clearer guidance through the findings and a more comprehensive scientific discussion. In response to your comment, we have made several improvements in the revised version:
> >
> > 1. **Refined Discussion of Motivation**: As **Response to W1** illustrated, the **Criss-Cross Attention** separates spatial and temporal dependencies in EEG signals, leveraging their unique structural characteristics to enhance representation learning compared to general transformers. The **ACPE** dynamically learns spatial relationships to adapt to diverse EEG channel configurations, overcoming the limitations of traditional absolute position encoding. We have enhanced the description of the motivations behind our proposed approach, providing a clearer narrative about why our method is important and how it addresses existing challenges in EEG analysis, particularly in the context of brain-computer interfaces (BCIs). **(Line 81-93, Page 2)**
> >
> > 2. **Additional Visualizations and Ablation Studies**: To improve the clarity of our findings, we have included more visual results **(Line 1638-1749, Page 31-33)** and detailed ablation studies **(Line 1494-1549, Page 28-29)**. Specifically, we have added new interpretability analyses, including topography visualization, representation clustering on downstream datasets, and patch relationship visualization, to demonstrate CBraMod's ability to capture symmetric patterns, task-specific representations, and criss-cross dependencies effectively. Meanwhile, we conducted additional experiments to analyze the split-head ratio in the criss-cross attention mechanism, confirming that balanced spatial and temporal attention (4:4) yields optimal performance, highlighting their equal importance.
> > 3. **Scientific Perspective and Relevance to the BCI Community**: We have expanded the discussion on the scientific value of our work, focusing on both its immediate implications and its long-term potential for EEG and BCI field in the section "Discussion" of the revised version **(Line 1754-1781, Page 33)**. Specifically, our proposed criss-cross modeling strategy and ACPE scheme address the unique structural characteristics of EEG data, offering novel insights for constructing adaptable and generalizable EEG foundation models. These advancements enhance the development of universal BCI systems by reducing reliance on labeled data and improving applicability across diverse populations and clinical settings.
> >
> > By addressing these points, we aim to guide the reader more effectively through our findings and ensure a deeper scientific understanding of the impact and novelty of our approach. Thank you once again for your valuable suggestions.

---

> ### Author Response · Authors · 2024-11-20
> **Response to Reviewer iYYg (3/n)**
>
> **W3**: The choice of datasets for motor imagery, emotion, and sleep stage tasks seems arbitrary. For instance, in the case of motor imagery, the most widely used dataset is BNCI 2014 version 004, and the largest is from Stieger et al. (2021) to the best of my knowledge. However, the authors chose Physionet-MI.
>
> **R**: Thank you for your comment. We appreciate your feedback on our dataset selection. Here are our responses:
>
> 1. **Comprehensive Dataset Selection**: Our evaluation involves up to 10 downstream tasks across 12 publicly available datasets, which, to the best of our knowledge, is unprecedented. Covering every well-known dataset for each downstream task is not feasible, and the datasets we selected are highly recognized within their respective fields. From the perspective of the authors, the choice of datasets should not be considered arbitrary simply because they do not align with the reviewer’s preferred datasets.
>
> 2. **Additional Experiment for Validation**: To further address the reviewer's concern, we conducted additional experiments on the BCIC-IV-2a dataset, as suggested by the reviewer. BCIC-IV-2a consists of 9 subject with two sessions. We use a strict subject-indenpendent train/validation/test strategy as other MI datasets in our paper. Subject 1-5, 6-7, 8-9 are used for training, validation, and test, respectively. The data preprocessing procedure is identical for all the methods. The results are shown in the follows:
>    ||BCIC-IV-2a, 4-class|||
>     |-|-|-|-|
>     |Methods|Balanced Acc.|Coken’s Kappa|Weighted F1|
>     |EEGNet|0.4482 $\pm$ 0.0094|0.2693 $\pm$ 0.0121|0.4226 $\pm$ 0.0108|
>     |EEGConformer|0.4696 $\pm$ 0.0106|0.2924 $\pm$ 0.0141|0.4533 $\pm$ 0.0128|
>     |SPaRCNet|0.4635 $\pm$ 0.0117|0.2847 $\pm$ 0.0147|0.4432 $\pm$ 0.0126|
>     |ContraWR|0.4678 $\pm$ 0.0125|0.2905 $\pm$ 0.0160|0.4413 $\pm$ 0.0142|
>     |CNN-Transformer|0.4600 $\pm$ 0.0108|0.2800 $\pm$ 0.0148|0.4460 $\pm$ 0.0114|
>     |FFCL|0.4470 $\pm$ 0.0143|0.2627 $\pm$ 0.0176|0.4238 $\pm$ 0.0139|
>     |ST-Transformer|0.4575 $\pm$ 0.0145|0.2733 $\pm$ 0.0198|0.4471 $\pm$ 0.0142|
>     |BIOT|0.4748 $\pm$ 0.0093|0.2997 $\pm$ 0.0139|0.4607 $\pm$ 0.0125|
>     |LaBraM|0.4869 $\pm$ 0.0085|0.3159 $\pm$ 0.0154|0.4758 $\pm$ 0.0103|
>     |CBraMod|**0.5138** $\pm$ 0.0066|**0.3518** $\pm$ 0.0094|**0.4984** $\pm$ 0.0085|
>
>    It demonstrates that our approach continues to outperform existing methods on this dataset, further reinforcing the generalizability of our method. In the revised version, we added this experiment **(Line 1329-1355, Page 25-26)**.
>
> We hope this clarifies our approach and demonstrates the robustness of our methodology. Thank you again for your valuable insights.

---

> > ### Author Response · Authors · 2024-11-20
> > **Response to Reviewer iYYg (4/n)**
> >
> > **W4**: The results in Physionet-MI are significantly below the state-of-the-art. A quick search reveals accuracies of 88.6% ± 9.0 using EEGSym, 86.36% with Zoumpourlis et al. (2023), and 73.60% with TIDNet on unseen subjects, where we have 64.17%. While I understand that the exact reproduction of results in EEG decoding is impossible, the reported results here are more than 20% below the state-of-the-art. This concern extends to sleep stage models, where datasets like SleepEDF+, MASS, or SHHS are standard, and accuracies above 80% are expected. The same issue applies to emotion classification; for example, Zhang et al. (2024) achieved better results on the SEED dataset testing with different train-test splits.
> >
> >
> > **R**: Thank you for your thoughtful feedback and for raising these points. We appreciate the opportunity to clarify the $\textcolor{red}{\textbf{distinctions between our work and the cited studies}}$.
> >
> > 1. **Motor Imagery**:
> >    - The Physionet-MI dataset we use includes four classes: $\textcolor{red}{\textbf{left fist, right fist, both fists, and both feet}}$. The studies you cited, EEGSym[2] (88.6% ± 9.0) and Zoumpourlis et al.[3] (86.36%), only consider **two classes (left fist and right fist)**, while TIDNet[4] (73.60%) evaluates **three classes (left fist, right fist, and eyes-open baseline)**. In contrast, our evaluation encompasses **all the four MI classes (left fist, right fist, both fists, and both feet)**, which introduces significantly greater complexity into the classification task.
> >    - Additionally, our study adopts a strict **subject-independent train/validation/test split** to rigorously evaluate generalization. The cited works may use different data splits or strategies, which makes direct comparison between the results challenging.
> >
> > 2. **Sleep Stage Classification**:
> >    - While the datasets mentioned (SleepEDF+, MASS, and SHHS) are commonly used in the field, our evaluations focus on the widely recognized ISRUC dataset. Each dataset has distinct characteristics, such as class distributions and preprocessing protocols, which make cross-dataset comparisons inherently difficult.
> >    - Moreover, many studies in this domain primarily report **accuracy**, which can be misleading given the **severe class imbalance** typical of sleep stage classification. In contrast, we prioritize metrics such as **balanced accuracy, Cohen’s kappa, and weighted F1-score**, providing a more nuanced and comprehensive evaluation. Therefore, the accuracy benchmarks cited cannot be directly compared with our results.
> >
> > 3. **Emotion Classification**:
> >    - The study by Zhang et al. (2024)[1] evaluates performance on the **SEED and SEED-IV** datasets, while our experiments are conducted on **SEED-V**, which is an entirely separate dataset with different subject pools and characteristics. Direct comparisons across these datasets are not scientifically valid.
> >    - Additionally, differences in evaluation metrics further complicate direct comparisons, as methodologies and reported metrics vary across studies.
> >
> > In conclusion, the concerns raised highlight important considerations but overlook key distinctions in **datasets, preprocessing methods, class configurations, data splits, and evaluation metrics**. These factors must be taken into account to ensure $\textcolor{red}{\textbf{fair and meaningful comparisons.}}$ We hope this explanation provides clarity and context for our reported results.
> >
> >
> >
> > [1] Zhang, Z., Zhong, S. H., & Liu, Y. (2024). TorchEEGEMO: A deep learning toolbox towards EEG-based emotion recognition. Expert Systems with Applications, 249, 123550.
> >
> > [2] Pérez-Velasco, S., Santamaría-Vázquez, E., Martínez-Cagigal, V., Marcos-Martínez, D., & Hornero, R. (2022). EEGSym: Overcoming inter-subject variability in motor imagery based BCIs with deep learning. IEEE Transactions on Neural Systems and Rehabilitation Engineering, 30, 1766-1775.
> >
> > [3] Zoumpourlis, G., & Patras, I. (2024). Motor imagery decoding using ensemble curriculum learning and collaborative training. In 2024 12th International Winter Conference on Brain-Computer Interface (BCI) (pp. 1-8). IEEE.
> >
> > [4] Kostas, D., & Rudzicz, F. (2020). Thinker invariance: enabling deep neural networks for BCI across more people. Journal of Neural Engineering, 17(5), 056008.

---

> ### Author Response · Authors · 2024-11-20
> **Response to Reviewer iYYg (5/n)**
>
> **W5**: Furthermore, when critically analyzing the model's usefulness based on the metrics—and considering literature from the field, such as O. Alkob et al. (2018) [1] and Combrisson and Jerbi (2015)—the metric values fall short of demonstrating real usefulness for BCI applications.
>
>
> **R**: Thank you for your detailed and thoughtful feedback. We acknowledge the importance of contextualizing our work in light of existing literature, particularly the studies by O. Alkob et al. (2018) [1] and Combrisson and Jerbi (2015) [2]. Below, we address your concerns regarding the metrics and their implications for BCI applications:
>
> 1. **Learning Generic Representation**: As highlighted by O. Alkob et al. (2018) [1], the efficacy of neurofeedback treatments and broader BCI systems can vary significantly among individuals, emphasizing the importance of developing models that generalize well across subjects. Our work focuses on **EEG representation learning**, aiming to create a **foundation model** that captures generalized EEG features. This approach is distinct from building application-specific BCI systems but is highly relevant to improving downstream task performance. A robust foundation model has significant potential to enhance **real-world BCI systems**, aligning well with the goal of enabling adaptive and personalized BCI solutions.
>
> 2. **Addressing Real-World Challenges**: We fully recognize that real-world BCI systems face unique challenges, including **signal noise, individual variability**, and the need for real-time adaptability. These challenges were also noted by O. Alkob et al. (2018) [1] in their discussion of neurofeedback inefficacy. In terms of data preprocessing, we only applied the most basic bandpass filtering and notch filtering, minimizing preprocessing steps in the hope that our model can learn useful representations even in noisy environments. To address individual differences, we employed a strict **subject-independent train/validation/test split** across most of the downstream datasets to rigorously evaluate generalization. These approaches are designed to ensure that our work holds practical value for real-world BCI systems.
>
> 3. **Bridging EEG Representation Learning and Practical Applications**: While our primary focus is on EEG representation learning, we believe that this is a **critical step towards enabling effective BCI systems**. Our work explores the way for a **One-For-All solution**, holding the promise of addressing multiple tasks with a unified model, positioning them as a transformative tool for diverse EEG applications. **This goal aligns closely with the overarching theme of ICLR**, which emphasizes the importance of advancing learning representations to tackle real-world challenges effectively.
>
> 4. **Metric Interpretation and Comparability**: As noted by Combrisson and Jerbi (2015) [2], evaluating classification performance in small datasets or under subject-independent settings requires careful consideration of **chance levels and statistical significance**. In our study, we utilized metrics such as **balanced accuracy, Cohen’s kappa, and weighted F1-score** for multi-class classification, which provide a **more comprehensive evaluation** than raw accuracy, particularly under conditions of class imbalance and inter-subject variability. While raw accuracy may not meet traditional expectations in certain tasks, the inclusion of more nuanced metrics ensures a fair and reliable assessment of our model’s capabilities.
>
> In summary, while our current results may not directly demonstrate the final utility of our model in real-world BCI applications, they underscore the potential of **representation learning** to enhance downstream BCI tasks and address the variability highlighted in neurofeedback and BCI research. We appreciate the reviewer’s insights.
>
>
> [1] O. Alkoby, A. Abu-Rmileh, O. Shriki and D. Todder, "Can we predict who will respond to neurofeedback? A review of the inefficacy problem and existing predictors for successful EEG neurofeedback learning", Neuroscience, vol. 378, pp. 155-164, May 2018.
>
> [2] Combrisson E, Jerbi K. Exceeding chance level by chance: The caveat of theoretical chance levels in brain signal classification and statistical assessment of decoding accuracy. Journal of neuroscience methods. 2015 Jul 30;250:126-36.

---

> > ### Author Response · Authors · 2024-11-20
> > **Response to Reviewer iYYg (6/n)**
> >
> > **W6**: The choice of baselines is inadequate and possibly inherits these shortcomings from the BioT and Labram studies. Although the authors cite well-established works in the literature, there is no comparison with neural networks like EEGNet, ShallowNet, EEGConformer, EEGSym, ATCNet, FBCNet, and many others that effectively capture temporal and spatial information in motor imagery tasks. By using neural networks with only frequency-based encoders (e.g., BioT), the spatial and temporal information critical for motor imagery is not learned. Employing weak baselines for classification tasks makes the gains appear larger than they actually are. This issue also applies to the sleep stage task, where models like USleep, DeepSleepNet, SleepTransformer, XSleepNet, SeqSleepNet, ChambonNet, and others should be considered as baselines.
> >
> > **R**: Thank you for your insightful comments regarding the choice of baselines. We appreciate your suggestions and would like to clarify our baseline selection and the steps we’ve taken to address your concerns:
> >
> > 1. **Rationale for Baseline Selection**: We selected the same baselines as those used in existing state-of-the-art methods, **BIOT (NeurIPS 2023) [1] and LaBraM (ICLR 2024 Spotlight) [2]**,  in the EEG foundation model field. These baselines are widely recognized and have demonstrated strong performance in a variety of EEG-related tasks. Given their prominence and relevance to our work, we believed these baselines provided an appropriate and rigorous comparison for evaluating the effectiveness of our proposed approach.
> >
> > 2. **Additional Comparisons with Suggested Models**: To further address your concern and eliminate any doubts, we have conducted additional experiments incorporating some of the neural network-based models you suggested for downstream tasks. Specifically, we compare our model with EEGNet[3] and EEGConformer[4] for motor imagery on PhysioNet-MI. EEGNet is a widely recognized but relatively older baseline model on EEG decoding. EEGConformer is a strong supervised-learning baseline model, developed by the same first author as ST-Transformer[5], which we have already used as a baseline in the orignial version of our paper. The results from these additional comparisons are as follows:
> >    ||PhysioNet-MI, 4-class|||
> >     |-|-|-|-|
> >     |Methods|Balanced Acc.|Coken’s Kappa|Weighted F1|
> >     |EEGNet|0.5814 $\pm$ 0.0125|0.4468 $\pm$ 0.0199|0.5796 $\pm$ 0.0115|
> >     |EEGConformer|0.6049 $\pm$ 0.0104|0.4736 $\pm$ 0.0171|0.6062 $\pm$ 0.0095|
> >     |CBraMod|**0.6417** $\pm$ 0.0091|**0.5222** $\pm$ 0.0169|**0.6427** $\pm$ 0.0100|
> >
> >    In the revised version of our paper, we added EEGNet and EEGConformer as baselines for all downstream datasets evaluation.
> >
> >    Meanwhile, we also compare our model with DeepSleepNet[6] and USleep[7] for sleep staging on ISRUC, which are both widely recognized baseline model on sleep staging.
> >    The results from these additional comparisons are as follows:
> >    ||ISRUC, 5-class|||
> >     |-|-|-|-|
> >     |Methods|Balanced Acc.|Coken’s Kappa|Weighted F1|
> >     |DeepSleepNet|0.7419 $\pm$ 0.0144|0.7036 $\pm$ 0.0241|0.7643 $\pm$ 0.0122|
> >     |USleep|0.7586 $\pm$ 0.0116|0.7209 $\pm$ 0.0143|0.7805 $\pm$ 0.0105|
> >     |CBraMod|**0.7865** $\pm$ 0.0110| **0.7442** $\pm$ 0.0152| **0.8011** $\pm$ 0.0099|
> >
> >    All the results demonstrate the superior performance of our method over these baselines. The results are added in the revised version of our paper **(Line 1092, Page 21)**.
> >
> > We hope these additions and clarifications address your concerns. Thank you again for your thoughtful feedback, and we are confident that the expanded set of comparisons will provide a more comprehensive evaluation of our approach.
> >
> > [1] Yang C, Westover M B, Sun J. BIOT: biosignal transformer for cross-data learning in the wild[C]//Proceedings of the 37th International Conference on Neural Information Processing Systems. 2023: 78240-78260.
> >
> > [2] Jiang W, Zhao L, Lu B. Large Brain Model for Learning Generic Representations with Tremendous EEG Data in BCI[C]//The Twelfth International Conference on Learning Representations.
> >
> > [3] Lawhern V J, Solon A J, Waytowich N R, et al. EEGNet: a compact convolutional neural network for EEG-based brain–computer interfaces[J]. Journal of neural engineering, 2018, 15(5): 056013.
> >
> > [4] Song Y, Zheng Q, Liu B, et al. EEG conformer: Convolutional transformer for EEG decoding and visualization[J]. IEEE Transactions on Neural Systems and Rehabilitation Engineering, 2022, 31: 710-719.
> >
> > [5] Song Y, Jia X, Yang L, et al. Transformer-based spatial-temporal feature learning for EEG decoding[J]. arXiv preprint arXiv:2106.11170, 2021.
> >
> > [6] DeepSleepNet: A model for automatic sleep stage scoring based on raw single-channel EEG[J]. IEEE transactions on neural systems and rehabilitation engineering, 2017, 25(11): 1998-2008.
> >
> > [7] U-Sleep: resilient high-frequency sleep staging[J]. NPJ digital medicine, 2021, 4(1): 72.

---

> ### Author Response · Authors · 2024-11-20
> **Response to Reviewer iYYg (7/n)**
>
> **W7**: Using the TEUV and TUAB datasets for abnormal detection and event-type classification while employing TEUG for pre-training leads to data leakage. This is indicated on the Temple University dataset's own webpage: "The TUH EEG Events Corpus (TUEV: v2.0.1): This corpus is a subset of TUEG."
>
> **R**: Thank you for raising the important concern regarding data leakage between the TUEV, TUAB, and TUEG datasets. We would like to clarify the following points:
>
> 1. **Purpose of Using TEUV and TUAB for Evaluation**: The primary reason for using the TEUV and TUAB datasets for evaluation is to conduct a more comprehensive comparison with the state-of-the-art methods like BIOT and LaBraM. Both of these methods have discussed scenarios where the evaluation datasets overlap with the pre-training data. In our case, the pre-training process is completely unsupervised and does not involve any label information, which we believe effectively prevents full data leakage.
>
>
> 2. **Additional Experiments to Address Data Leakage Concern**: We fully understand your concern about potential data leakage. To further eliminate this doubt, we conducted an additional experiment in which we removed the TUAB and TUEV subsets from the TEUG dataset before pre-training CBraMod. After this step, we used the pre-trained model to validate its performance on the TUAB and TUEV subsets. This experiment ensures that no data overlap exists between the pre-training and evaluation phases and provides further confidence in the integrity of our results.
>    ||TUEV, 6-class|||
>     |-|-|-|-|
>     |Methods|Balanced Acc.|Coken’s Kappa|Weighted F1|
>     |CBraMod (excluding TUEV)|0.6659 $\pm$ 0.0124|0.6744 $\pm$ 0.0121|0.8331 $\pm$ 0.0071|
>     |CBraMod|0.6671 $\pm$ 0.0107|0.6772 $\pm$ 0.0096|0.8342 $\pm$ 0.0064|
>
>    ||TUAB, 2-class|||
>     |-|-|-|-|
>     |Methods|Balanced Acc.|AUC-PR|AUROC|
>     |CBraMod (excluding TUAB)|0.8249 $\pm$ 0.0025|0.9221 $\pm$ 0.0015|0.9156 $\pm$ 0.0017|
>     |CBraMod|0.8289 $\pm$ 0.0022|0.9258 $\pm$ 0.0008|0.9227 $\pm$ 0.0011|
>
>    The experimental results show that, after thoroughly eliminating data leakage, the performance of our model only experiences fluctuations, but remains competitive. This indicates that our method successfully learns a generic EEG representation through pre-training, thereby enhancing the performance on datasets unseen during pre-training. The experiments are added in the revised version of our paper **(Line 1269, Page 24), (Line 1310, Page 25)**.
>
>
> We hope these clarifications and the additional experiment help address your concerns regarding data leakage. Thank you for your careful review, and we look forward to your feedback.

---

> > ### Comment · Reviewer_iYYg · 2024-11-25
> >
> > Dear authors,
> >
> > I sincerely appreciate all the effort you have made during this period. I believe you have managed to clarify some doubts and points I had about the results.
> >
> > I also apologize for the incorrect indication of studies with the wrong number of classes for Physionet. Considering studies that used the same number of classes, for example, EEG-SimpleConv, their results are at least 10% to 15% lower, especially compared to a much simpler approach, 64.6% (EEG-SimpleConv) vs 51.38% (CBraMod).
> >
> > This comparison is extremely evidenced by the new experiment on the BCNI competition 2a dataset, the results are below what we understand as the state of the art in the literature dedicated to EEG Decoding.
> >
> >
> > |   	Method  	| Cross-Subject |
> > |:-----------------:|:-------------:|
> > | Multi-branch 3D |      	52.2 |
> > | CCNN       	|      	55.3 |
> > | CMO-CNN     	|      	63.3 |
> > | DFNN     	|      	64.4 |
> > | TIDNet      	|      	65.4 |
> > | EEGConformer [4]  | 44.78 ± 0.73  |
> > | ShallowNet [2]	| 47.79 ± 0.84  |
> > | EEGNet [1]    	| 52.74 ± 1.34  |
> > | EEGTCNet [6]  	| 54.51 ± 2.46  |
> > | TS-SEFFNet [3]	| 55.21 ± 0.36  |
> > | ATCNet [5]    	| 57.81 ± 2.00  |
> > | BaseNet + CAT 	| 57.99 ± 1.10  |
> > | EEGNet    	| 64.0 ± 11.6   |
> > | EEG-TCNet   	| 65.1 ± 10.9   |
> > | EEGNet      	| 66.2 ± 9.9	|
> > | EEG-Inception   | 66.3 ± 8.7	|
> > | EEG-ITNet    	| 69.4 ± 8.9	|
> > | EEG-SimpleConv	| 72.1 ± 7.3	|
> > | CBraMod (yours)   |    51.38 ± 0.66 |
> >
> > TABLE: Combined table from BaseNet paper and EEG-SimpleConv paper
> >
> > What really worries me about these results is that I did not perform a systematic search for state-of-the-art works, and I was able to find several that beat what you are present here. My search results were extremely favorable for you, given that these methods do not employ domain adaptation techniques, which can greatly increase the results a lot, especially in cross-subject evaluation.
> >
> > IMHO, some points answered by the authors are not convincing, such as the failure to use a more solid theoretical formulation to estimate the real utility of the BCI system per subject. Even using cross-subject training, a simple separation of the test set by the subject would allow such analysis as a whole.
> >
> > The rationale for choosing the baselines was also not convincing, and such points were also pointed out during the review of the articles that you use to justify BioT and Labram. Especially Labram, which presents severe reproducibility problems. A set of models dedicated to each modality is more appropriate.
> >
> > Given this context, I maintain my rating.

---

> ### Author Response · Authors · 2024-11-26
> **Response to Reviewer iYYg (8/n)**
>
> We appreciate your continued engagement and detailed feedback. However, we would like to address some misunderstandings and provide additional context regarding the concerns raised:
>
> 1. **Comparison with EEG-SimpleConv**:
>    We carefully reviewed the EEG-SimpleConv[1] paper and its associated code and found that the reported performance of **72.1 ± 7.3 accuracy** under the leave-one-subject-out (LOSO) protocol relies heavily on several advanced preprocessing and training techniques referred to as "Key Ingredients," including **Euclidean Alignment (EA)**, **session statistics**, **Mixup**, and **subject-wise regularization**. When these techniques are not employed, EEG-SimpleConv achieves significantly lower performance: **56.4 ± 9.0 accuracy** (as reported in the original paper).
>
>    In our work, we deliberately adopt minimal preprocessing to ensure a fair and standardized evaluation, which makes direct comparison to methods employing extensive preprocessing techniques, such as EEG-SimpleConv with Key Ingredients, inappropriate.
>
> 2. **Fair Comparison with LOSO Protocol**:
>    To provide a more rigorous and fair evaluation, we conducted experiments under the same LOSO protocol used by EEG-SimpleConv, both with and without incorporating the "Key Ingredients." The results are summarized as follows:
>
>    ||BCIC-IV-2a, LOSO|||
>    |-|-|-|-|
>    |Methods|Balanced Acc.|Cohen’s Kappa|Weighted F1|
>    |EEG-SimpleConv (*w/o Key Ingredients*)|0.5650 ± 0.0989|0.4201 ± 0.1319|0.5484 ± 0.1047|
>    |CBraMod (*w/o Key Ingredients*)|**0.5968 ± 0.0816**|**0.4558 ± 0.1088**|**0.5889 ± 0.0875**|
>    |EEG-SimpleConv (*w/ Key Ingredients*)|0.7221 ± 0.0768|0.5765 ± 0.1069|0.6977 ± 0.0918|
>    |CBraMod (*w/ Key Ingredients*)|**0.7405 ± 0.0635**|**0.5997 ± 0.0833**|**0.7195 ± 0.0682**|
>
>    These results demonstrate that:
>    - Our reproduction of EEG-SimpleConv matches the original paper’s findings.
>    - CBraMod consistently outperforms EEG-SimpleConv under both settings, achieving higher balanced accuracy, Cohen’s Kappa, and weighted F1 scores while also demonstrating lower inter-subject variance, highlighting its robustness and effectiveness.
>
> 3. **Motivation of EEG foundation models**: The significance of our EEG foundation models lies in their ability to learn generalized representations from large-scale unlabeled data, enabling robust performance **across a wide range of downstream tasks**, not only motor imagery or emotion recognition. In fact, **learning generalizable representations is one of the key pursuits emphasized by the ICLR conference.** Unlike traditional methods that rely on task-specific feature engineering, prior knowledge, or extensive preprocessing techniques, these models provide a unified and stable framework, streamlining EEG analysis and reducing dependency on handcrafted solutions.
>
> We hope this clarification adequately addresses your concerns and provides a clearer context for the experimental design and results presented in our work. We remain open to further discussion and thank you again for your thoughtful feedback. Regardless of whether this paper is accepted, we sincerely appreciate the reviewer's detailed and valuable feedback, particularly the insights on the motor imagery task. We hope our responses can help bridge the gap between the reviewer's and the authors' academic perspectives, working together to advance the field of EEG decoding.
>
> [1] El Ouahidi Y, Gripon V, Pasdeloup B, et al. A Strong and Simple Deep Learning Baseline for BCI Motor Imagery decoding[J]. IEEE Transactions on Neural Systems and Rehabilitation Engineering, 2024.

---

> > ### Comment · Reviewer_iYYg · 2024-11-26
> >
> > Dear authors,
> >
> > After this last message, I was convinced of the usefulness of the model, and I increased the score to 6.
> >
> > I ask that this final discussion be included in the article to help the community as a whole. I politely ask that you also make the full code available in its entirety for reproduction in open review with the baselines. If this is done, I can increase the score further, understanding that the model meets the reproducibility requirements.

---

> > > ### Author Response · Authors · 2024-11-27
> > > **Response to Reviewer iYYg (9/n)**
> > >
> > > Dear Reviewer iYYg,
> > >
> > > Thank you very much for your thoughtful consideration and for raising the score to 6. We deeply appreciate your constructive feedback and your acknowledgment of our model’s usefulness.
> > >
> > > **Incorporating Discussions into the Revised Paper:** We are pleased to inform you that the discussion about performance comparison with EEG-SimpleConv has been incorporated into the revised version of our paper. This addition can be found at the end of the appendix **(Lines 1811-1835, Pages 34)**, aiming to better support the community in understanding our contributions.
> > >
> > > **Enhancing Reproducibility:** In our original supplementary materials, we had already provided the pretraining code. To further address your expectations, we have newly included the pretrained weights and fine-tuning code for the BCIC-IV-2a downstream dataset in the updated supplementary materials. Our model is designed to be plug-and-play, enabling straightforward application to various tasks by adapting the classifier for specific downstream datasets. Due to time constraints, the newly added code might lack detailed annotations, and we kindly ask for your understanding in this regard. We remain committed to ensuring reproducibility and hope these materials meet your expectations. In the future, we will release the refined code on GitHub and provide ongoing maintenance to better serve the community.
> > >
> > > Thank you once again for your support and invaluable feedback.
> > >
> > > Best regards,
> > >
> > > The authors

---

> > > > ### Comment · Reviewer_iYYg · 2024-11-27
> > > >
> > > > I increased the score to 8.
> > > >
> > > > Good luck with your research. I hope to see the GitHub soon!

---

> > > > > ### Author Response · Authors · 2024-11-27
> > > > > **We are deeply grateful for your thoughtful evaluation and for increasing the score from 3 to 8!**
> > > > >
> > > > > Dear Reviewer iYYg,
> > > > >
> > > > > **We are deeply grateful for your thoughtful evaluation and for increasing the score from 3 to 8!** Your rigorous scientific approach and open-minded academic perspective are truly inspiring and set an admirable example for us.
> > > > >
> > > > > Engaging with your comments and suggestions throughout this review process has been an immensely rewarding experience for us! **We have gained valuable insights and made significant improvements to our work thanks to your guidance!** Your positive assessment has been a tremendous source of encouragement for our team, reaffirming the importance of our contributions to the field.
> > > > >
> > > > > We will ensure the quality and accessibility of our work. **We will dedicate our best efforts to finalize and release the GitHub repository as soon as possible, aiming to provide meaningful support to the community and facilitate future advancements.**
> > > > >
> > > > > Once again, we sincerely thank you for your invaluable feedback and kind words.
> > > > >
> > > > > Best regards,
> > > > >
> > > > > The Authors

---

### Official Review · Reviewer_m8H6 · 2024-11-02

**Soundness:** 2
**Presentation:** 3
**Contribution:** 2
**Rating:** 5
**Confidence:** 4

**Summary:**

In response to the limitations of existing EEG models, which primarily focus on whole-brain modeling while neglecting spatiotemporal dependencies, as well as the challenges faced by current EEG backbone models in handling EEG data in various formats, this study proposes an EEG foundation model based on a criss-cross transformer backbone. This work involves pretraining on a substantial dataset of EEG signals and fine-tuning on multiple downstream tasks.

**Strengths:**

This work addresses the challenges currently faced in EEG models by proposing an effective solution for spatiotemporal modeling through enhancements in model architecture. The model has undergone pretraining on a substantial dataset, and the experimental results are comprehensive, providing support for several contributions.

**Weaknesses:**

Due to the pretraining paradigm of CBraMod, it has not effectively addressed the challenges posed by the diversity of EEG data formats. This raises concerns regarding its suitability as a foundation model for EEG. The specific issues are outlined as follows.
1. This work acknowledges that current EEG foundation models still face challenges in handling EEG data of varying formats. However, the solution proposed by CBraMod during the pretraining phase, which involves the selection of 19 common EEG channels, appears to be a simplistic approach that does not effectively address the aforementioned issues.
2. To learn generic representations from both time-domain and frequency-domain EEG signals, this study conducted pretraining on the TUEG dataset, which primarily focuses on medical data related to conditions such as epilepsy. However, as a foundation model, CBraMod exhibits significant discrepancies between the pretraining data and the data used for downstream tasks. This raises questions regarding its theoretical validity.
3. The ablation studies related to pretraining, being a crucial experiment, should be included in the main text.

**Questions:**

1. Given that this work is positioned as a foundation model, it is pertinent to inquire whether scaling laws exist at both the data level and the model size level. Have the authors conducted relevant experiments to explore these scaling laws?
2. It would be valuable to examine the loss function curves during the pretraining phase.
3. Could the authors elucidate their perspective on the motivation or value of foundation models within the EEG domain? Additionally, how does CBraMod balance the relationship between model size, computational cost, and performance?
4. During the pretraining phase, 19 common electrodes were utilized. How does the model handle discrepancies in the number of electrodes when applied to downstream tasks?

---

> ### Author Response · Authors · 2024-11-20
> **Response to Reviewer m8H6 (1/n)**
>
> We are grateful for the reviewer's attentive analysis and helpful feedback.
>
> **W1**: This work acknowledges that current EEG foundation models still face challenges in handling EEG data of varying formats. However, the solution proposed by CBraMod during the pretraining phase, which involves the selection of 19 common EEG channels, appears to be a simplistic approach that does not effectively address the aforementioned issues.
>
> **R**: Thank you for your insightful comment regarding the challenge of handling EEG data with varying formats. We acknowledge that this remains a significant issue for EEG foundation models.
>
> 1. **Limitation of existing EEG foundation model on adaptation**: Current EEG foundation models, such as those employing Transformers, are indeed effective in handling the varying length of EEG signals due to their strong adaptive capabilities. However, one limitation of the Transformer architecture is its inability to learn the spatial relationships between EEG channels, which is crucial for handling variations in the EEG channel format across different downstream tasks. Typically, existing models address this limitation by using absolute position encoding based on electrode numbers as channel embeddings, as seen in methods like LaBraM. However, EEG channels are not simply equivalent to electrode positions, as they are also influenced by factors such as reference schemes (e.g., ear reference, average reference, REST, and bipolar montages). Therefore, we believe that this traditional position encoding approach limits the adaptability of EEG foundation models to different downstream tasks.
>
> 2. **Our Proposed Method**: To address this challenge, we proposed an alternative solution with our **ACPE (Asymmetric Conditional Positional Encoding)**. This method leverages a convolutional network to dynamically learn the spatial relationships between each patch and other patches, allowing the model to better adapt to varying channel configurations in downstream tasks. By doing so, we aim to enhance the flexibility and applicability of EEG foundation models across diverse EEG data formats.
>
> In our revised manuscript, we have enhanced the discussion of these methods, particularly with respect to their motivation and design **(Line 81-93, Page 2)**. We believe our approach offers a novel and valuable contribution to the design of positional encoding in EEG foundation models, and we hope it provides useful insights for overcoming the challenges of EEG data variability.
>
> Thank you again for your valuable feedback.
>
> ---
>
> **W2**: To learn generic representations from both time-domain and frequency-domain EEG signals, this study conducted pretraining on the TUEG dataset, which primarily focuses on medical data related to conditions such as epilepsy. However, as a foundation model, CBraMod exhibits significant discrepancies between the pretraining data and the data used for downstream tasks. This raises questions regarding its theoretical validity.
>
> **R**: Thank you for your comment. We address your concerns regarding the use of the TUEG dataset for pretraining with the following points:
>
> 1. **Large, Diverse, and Long-duration Dataset**: Although TUEG primarily focuses on medical data related to conditions such as epilepsy, it includes a large and diverse participant pool, along with long-duration EEG recordings. This diversity allows the model to learn generalizable features that are not specific to any one condition, while the long-duration recordings provide rich temporal context for capturing complex dependencies within EEG signals.
>
> 2. **Broad Applicability Across Tasks**: By pretraining on this extensive dataset, CBraMod learns generic EEG representations that improve performance on downstream tasks. Despite the medical focus of the pretraining data, the learned features are not overly specialized to epilepsy-related signals. Instead, they capture essential EEG characteristics that can be adapted to a wide range of tasks. The experimental results show that CBraMod achieves the state-of-the-art performance across up to 10 downstream BCI tasks (12 public datasets), empirically proving its strong capability and generalizability.
>
> We hope these points clarify the rationale behind our choice of pretraining data. Thank you again for your valuable feedback.
>
> ---
>
> **W3**: The ablation studies related to pretraining, being a crucial experiment, should be included in the main text.
>
> **R**: Thank you for your suggestion. We acknowledge that the ablation studies related to pretraining are a crucial part of our experiments. In the original version of the paper, these ablation studies were included in the appendix. As the reviewer suggested, we have moved this content to the main text to highlight its importance and ensure clearer visibility in the revised version **(Line 507-527, Page 10)**. We believe this will provide a more comprehensive understanding of the impact of pretraining on our model's performance.

---

> > ### Author Response · Authors · 2024-11-20
> > **Response to Reviewer m8H6 (2/n)**
> >
> > **Q1**: Given that this work is positioned as a foundation model, it is pertinent to inquire whether scaling laws exist at both the data level and the model size level. Have the authors conducted relevant experiments to explore these scaling laws?
> >
> > **R**: Thank you for your insightful question. In response to your inquiry regarding scaling laws at both the data level and model size level, we have made the following updates:
> >
> > 1. **Exploration of Data Size Scaling Law**: In the revised version of the paper, we have added experiments to explore how the performance of our model changes with varying data sizes **(Line 1360-1391, Page 26)**. The performance improves as the scale of pretraining data increases, although the rate of improvement slows beyond 1000 hours.
> > It investigates the impact of scaling the dataset on the model’s ability to learn generalizable representations, which is crucial for understanding how data size affects the performance of foundation models.
> >
> > 2. **Exploration of Model Size Scaling Law**: Additionally, we have explored the scaling of model size and its impact on the performance of CBraMod **(Line 1360-1391, Page 26)**. Larger model sizes lead to better performance in downstream tasks. It examines how increasing the number of parameters in the model influences its ability to capture complex patterns in EEG data, providing insights into the optimal model size for foundation models.
> >
> > In summary, due to rebuttal time and computational resource limitations, we were only able to conduct a limited set of experiments. However, we have made every effort to explore scaling laws for both data size and model size within the constraints. In the future, we plan to extend these experiments to larger scales to further understand the scalability of foundation models. Thank you again for your valuable feedback.
> >
> > ---
> >
> > **Q2**: It would be valuable to examine the loss function curves during the pretraining phase.
> >
> > **R**: Thank you for your suggestion. In the revised version of the paper, we have added the loss function curves during the pretraining phase in the appendix **(Line 968-988, Page 18-19)**. This provides additional insights into the training dynamics, allowing readers to observe how the model's loss evolves throughout the pretraining process. We believe this addition enhances the transparency and clarity of our model's training behavior. Thank you again for your valuable feedback.

---

> ### Author Response · Authors · 2024-11-20
> **Response to Reviewer m8H6 (3/n)**
>
> **Q3**: Could the authors elucidate their perspective on the motivation or value of foundation models within the EEG domain? Additionally, how does CBraMod balance the relationship between model size, computational cost, and performance?
>
> **R**: Thank you for your insightful question. Below is a structured response regarding the motivation behind EEG foundation models and the trade-off between model size, computational cost, and performance in CBraMod:
>
> 1. **Motivation and value for EEG Foundation Models**: The primary goal of EEG foundation models is to **learn generalized representations from large-scale unlabeled EEG data with self-supervised learning**, thereby enhancing the performance of diverse downstream tasks **without relying on handcrafted features, prior knowledge specific to individual tasks or large amounts of labeled data**. Unlike traditional methods that rely on task-specific feature engineering or supervised learning requiring large amounts of labeled data, a foundation model provides a unified framework to streamline EEG analysis and enable applications such as brain-computer interfaces (BCIs).
>
> 2. **Long-term Vision**: In the long run, EEG foundation models aim to **revolutionize the existing paradigms** of EEG research and application, similar to how large language models (LLMs) and vision models have done for their respective fields. These models have the potential to inspire more paradigm-agnostic or universal brain-computer interface (BCI) designs. In the revised version of our paper, we detailedly discussed the impilcations of our work **(Line 1754-1781, Page 33)**.
>
> 3. **Model Size, Computational Cost, and Performance Trade-offs**: In our original paper, we provide comparisons of **model size and computational cost** between our approach, supervised algorithms, and other EEG foundation models **(Line 1607-1636, Page 30-31)**. While our method is computationally more expensive than traditional supervised methods, it is still more efficient compared to existing EEG foundation models.
>
> 4. **Performance Benefits**: While it’s challenging for EEG foundation models to be more efficient than traditional supervised algorithms, they offer **better performance**. By exploring the way for a **One-For-All solution**, EEG foundation models hold the promise of addressing multiple tasks with a unified model, positioning them as a transformative tool for diverse EEG applications.
>
> This summarizes the motivation behind EEG foundation models and the trade-offs in terms of model size, computational cost, and performance in CBraMod. Thank you for your insightful question!
>
> ---
>
> **Q4**: During the pretraining phase, 19 common electrodes were utilized. How does the model handle discrepancies in the number of electrodes when applied to downstream tasks?
>
> **R**: Please see the **Response to W1**.

---

> > ### Comment · Reviewer_m8H6 · 2024-11-25
> >
> > Thank you for the authors’ response. However, my two primary concerns remain unaddressed.
> >
> > First, the authors have not provided a clear and effective explanation of how CBraMod addresses the challenge posed by the diverse formats of EEG data. This leads me to believe that the CBraMod model can only process inputs of the same format (i.e., the 19 selected common channels) during the pretraining phase. This limitation contradicts the authors' claim that CBraMod can handle EEG data of varying formats as a foundational model.
> >
> > Second, regarding the motivation for developing a foundational EEG model, the authors' explanation remains overly general and fails to convincingly articulate the advantages of a larger parameter model that requires more computational resources during training compared to lightweight, task-specific models tailored for individual downstream tasks. Moreover, the comparative results presented by the authors do not seem to demonstrate clear advantages of CBraMod. Additionally, CBraMod appears to undergo pretraining followed by fine-tuning for each downstream task, effectively resulting in task-specific models being generated for each individual task. This approach seems inconsistent with the authors’ "one-for-all" claim.
> >
> > These issues have left me somewhat disappointed with the work, leading me to lower my evaluation score. If there are any misunderstandings on my part, I am open to further discussion.

---

> > > ### Author Response · Authors · 2024-11-25
> > > **Response to Reviewer m8H6 (4/n)**
> > >
> > > Thank you for your thoughtful feedback. We believe there may be some misunderstandings regarding our rebuttal, and we would like to clarify further:
> > >
> > > 1. **Diverse EEG Formats**: While CBraMod was pretrained using 19-channel EEG data, this does not restrict its ability to process data with different channel configurations. The proposed **Asymmetric Conditional Positional Encoding (ACPE)** employs a convolutional network to dynamically learn the relative positional relationships among channels, allowing CBraMod to adapt flexibly to diverse channel layouts in downstream tasks on the fine-tuning process. This capability ensures that CBraMod can generalize beyond the specific format used during pretraining, helping mitigate the challenge of heterogeneous EEG formats.
> > >
> > > 2. **Motivation of EEG foundation models**: The primary goal of our EEG foundation models is to learn generalized representations from large-scale unlabeled EEG data with self-supervised learning, thereby enhancing the performance of diverse downstream tasks without relying on handcrafted features, prior knowledge specific to individual tasks or large amounts of labeled data. In fact, learning generalizable representations is one of the key pursuits emphasized by the ICLR conference. Unlike traditional methods that rely on task-specific feature engineering or supervised learning requiring large amounts of labeled data, CBraMod provides a unified framework to streamline EEG analysis and enable applications such as brain-computer interfaces (BCIs). The evaluation on all the 10 downstream tasks of our CBraMod further proves its effectiveness.
> > >
> > > We acknowledge that EEG foundation models are still in their early stages compared to large-scale models in NLP and computer vision, and none of the existing EEG foundation models can truly adopt a "one-for-all" paradigm. However, CBraMod offers valuable insights in the fields of EEG and BCI. This underscores the significant potential for exploration and innovation in designing robust, scalable EEG foundation models in the future.
> > >
> > > We hope this clarification addresses your concerns and would be happy to engage further if there are additional questions.

---

> ### Author Response · Authors · 2024-11-28
> **Response to Reviewer m8H6 (5/n): More Detailed Responses**
>
> Dear Reviewer m8H6,
>
> First and foremost, we would like to sincerely thank you for your careful review and valuable feedback on our work. We have carefully considered each of the concerns you raised and would like to provide more detailed responses to address these concerns.
>
> **Concern 1:** First, the authors have not provided a clear and effective explanation of how CBraMod addresses the challenge posed by the diverse formats of EEG data. This leads me to believe that the CBraMod model can only process inputs of the same format (i.e., the 19 selected common channels) during the pretraining phase. This limitation contradicts the authors' claim that CBraMod can handle EEG data of varying formats as a foundational model.
>
> **Response 1:** Thank you for your valuable comment. We would like to clarify the concerns raised regarding the diversity of EEG formats and CBraMod's ability to handle them.
>
> 1. **CBraMod’s Capability to Handle EEG data with varying numbers of channels and temporal segments:**  The backbone of CBraMod is the criss-cross transformer, which is an improved version of the traditional transformer architecture for EEG signals. The transformer backbone allows CBraMod to **process EEG data with varying numbers of channels and temporal segments**. The criss-cross transformer is designed to handle such diversity at a technical level, making it highly adaptable to different EEG formats. Therefore, CBraMod is not limited to processing data with the 19 selected channels during pretraining.
>
> 2. **Improvement in positional encoding with ACPE:**
> To address the challenges of positional encoding in EEG data, we introduced **Asymmetric Conditional Positional Encoding (ACPE)**. Traditional EEG foundation models rely on **electrode-wise absolute positional encodings**, which often fail to generalize across different channel configurations, particularly when **various referencing schemes (e.g., earlobe, average, REST, or bipolar references) lead to non-equivalent channel-electrode mappings**. ACPE employs a convolutional network to dynamically learn the relative positional relationships among channels, allowing CBraMod to adapt flexibly to diverse channel layouts in downstream tasks on the fine-tuning process. This capability ensures that CBraMod can generalize beyond the specific format used during pretraining, helping mitigate the challenge of heterogeneous EEG formats.
> **In fact, the strong performance of CBraMod across all downstream datasets has already empirically demonstrated its ability to handle diverse EEG data.**
>
> 3. **Large EEG Corpus TUEG:**
> We selected the TUEG dataset because it includes a large and diverse participant pool, along with long-duration EEG recordings. The dataset is sufficiently large to facilitate the learning of generic EEG representations that can be transferred to various downstream tasks. **Importantly, this choice does not imply that CBraMod is limited to training on 19-channel EEG data.** **CBraMod is inherently capable of handling EEG data with varying numbers of channels**, and we plan to extend the model to encompass more diverse pretraining datasets in the future work.
>
> We hope our contributions can provide insights for EEG foundation model research.
>
>
> [1] Yang C, Westover M B, Sun J. BIOT: biosignal transformer for cross-data learning in the wild[C]//Proceedings of the 37th International Conference on Neural Information Processing Systems. 2023: 78240-78260.
>
> [2] Jiang W, Zhao L, Lu B. Large Brain Model for Learning Generic Representations with Tremendous EEG Data in BCI[C]//The Twelfth International Conference on Learning Representations.

---

> > ### Author Response · Authors · 2024-11-28
> > **Response to Reviewer m8H6 (6/n): More Detailed Responses**
> >
> > **Concern 2:** Second, regarding the motivation for developing a foundational EEG model, the authors' explanation remains overly general and fails to convincingly articulate the advantages of a larger parameter model that requires more computational resources during training compared to lightweight, task-specific models tailored for individual downstream tasks.
> >
> > **Response 2:** We appreciate your thoughtful feedback on the motivation behind developing a foundational EEG model. We would like to address the concerns regarding the computational trade-offs and the advantages of our approach.
> >
> > 1. **Computational Cost Trade-offs in Foundation Models:**
> > We respectfully disagree with the premise that the computational cost of foundation models, such as CBraMod, inherently diminishes their value compared to lightweight, task-specific models. CBraMod demonstrates notable improvements in computational efficiency compared to existing EEG foundation models like **BIOT** and **LaBraM**, as illustrated in the paper **(Line 1608-1636, Page 30-31)**. While CBraMod's computational requirements are higher than those of lightweight supervised models, this trade-off is intrinsic to foundation models, offering better generalization and adaptability to **diverse downstream tasks**. **We believe the computational cost should not be viewed as a significant weakness for a foundation model.**
> >
> > 1. **Superior Performance in Low-Resource Settings:**  We also compared CBraMod with supervised models under **low-resource settings**. The results are shown as follows:
> >    ||FACED, 9-class|||
> >    |-|-|-|-|
> >    |Methods|Balanced Acc.|Coken’s Kappa|Weighted F1|
> >    |CBraMod (30%)|**0.4035** $\pm$ 0.0233|**0.3239** $\pm$ 0.0265|**0.4056** $\pm$ 0.0256|
> >    |EEGNet (30%)|0.2222 $\pm$ 0.0251|0.1268 $\pm$ 0.0340|0.2106 $\pm$ 0.0269|
> >    |EEGConformer (30%)|0.2816 $\pm$ 0.0241|0.1918 $\pm$ 0.0366|0.2676 $\pm$ 0.0271|
> >
> >    The results consistently show that CBraMod outperforms lightweight supervised models when only limited labeled data is available. **This advantage underscores the utility of foundation models like CBraMod in scenarios where labeled data is scarce—a common challenge in EEG analysis.**
> >
> > 2. **Faster Convergence on Downstream Tasks:**
> > To further highlight the advantages of CBraMod, we conducted additional comparison on its convergence speed with a lightweight supervised model, **EEGConformer**, on downstream tasks. The results demonstrate that CBraMod achieves significantly faster convergence, indicating its ability to learn **generic EEG representations** effectively, which translates into accelerated fine-tuning for downstream applications. We have added this comparison in the newest revised version of our paper. **(Line 1837-1862, Page 35)**
> >
> >
> >
> > These findings collectively highlight the strengths of CBraMod as a foundational EEG model. By trading slightly higher training complexity for better generalization, faster convergence, and superior performance in low-resource conditions, CBraMod offers clear advantages over traditional task-specific models.
> >
> >
> > ---
> >
> >
> > **Concern 3:** Moreover, the comparative results presented by the authors do not seem to demonstrate clear advantages of CBraMod.
> >
> > **Response 3:** We respectfully disagree with the assertion that the comparative results fail to demonstrate clear advantages of CBraMod. Our experiments consistently show that CBraMod achieves **significant performance improvements across all the downstream tasks** when compared to supervised models. These results highlight the ability of CBraMod to learn generalizable representations that transfer effectively to diverse tasks.
> >
> > Moreover, CBraMod demonstrates **notable superiority over existing EEG foundation models**, such as BIOT and LaBraM, in terms of task performance. These improvements validate the efficacy of our proposed methods and architectural innovations. In light of this evidence, we find it difficult to agree with the suggestion that CBraMod does not exhibit clear advantages.
> >
> > In fact, the reviewer m8H6 acknowledged the strength of our approach in the **Strengths** section, stating, "*The model has undergone pretraining on a substantial dataset, and the experimental results are comprehensive, providing support for several contributions.*"
> > **It is evident that the reviewer's concern regarding the comparative results is in conflict with the advantages the reviewer has acknowledged.
> > Furthermore, other reviewers (k8cX, 1DYx and iYYg) have also acknowledged the superiority of our experimental results.**
> >
> > Given this positive recognition, we believe that the comprehensive results in our paper substantiate the contributions of CBraMod as a foundational EEG model.

---

> > > ### Author Response · Authors · 2024-11-28
> > > **Response to Reviewer m8H6 (7/n): More Detailed Responses**
> > >
> > > **Concern 4:** Additionally, CBraMod appears to undergo pretraining followed by fine-tuning for each downstream task, effectively resulting in task-specific models being generated for each individual task. This approach seems inconsistent with the authors’ "one-for-all" claim.
> > >
> > > **Response 4:** Thank you for your insightful comment. We would like to clarify the perceived misunderstanding regarding our claims about the "one-for-all" model.
> > >
> > > 1. **Clarification on Our Claims:**  We are concerned there may have been a misunderstanding regarding our claims. To clarify, we **have not asserted that CBraMod is a "one-for-all" model**—neither in the paper nor in the rebuttal. In fact, in our **response to Reviewer m8H6 (3/n)**, we explicitly stated:
> > > *"By exploring the way for a One-For-All solution, EEG foundation models hold the promise of addressing multiple tasks with a unified model, positioning them as a transformative tool for diverse EEG applications."*
> > > This statement highlights the long-term motivation behind EEG foundation models as a research direction. **It does not claim that our current work has fully realized such a paradigm.**
> > >
> > > 2. **Current State of Research on EEG Foundation Models:**
> > > In fact, EEG foundation models are still in their infancy compared to foundation model in NLP and computer vision. **None of the existing models, including BIOT, LaBraM, CBraMod, can yet achieve a true "one-for-all" paradigm.** However, CBraMod makes **notable progress by learning generic representations and achieving strong performance across diverse tasks**, offering valuable insights for advancing the fields of EEG and BCI.
> > >
> > > We hope the clarifications provided above address your concerns effectively.
> > >
> > > ---
> > >
> > > In closing, we would like to express our sincere appreciation for your thoughtful feedback, which has greatly improved the quality of our paper. We look forward to addressing any further questions you may have and refining our work further based on your comments.

---

> > > > ### Author Response · Authors · 2024-11-28
> > > > **A kind reminder to Reviewer m8H6**
> > > >
> > > > Dear Reviewer m8H6,
> > > >
> > > > We would like to kindly remind you that we have submitted a more detailed response to your concerns. We have addressed the points you raised and provided further clarifications in our revised submission.
> > > >
> > > > We hope our additional explanations help clarify the issues, and we greatly appreciate your continued feedback.
> > > >
> > > > Thank you once again for your thoughtful review.
> > > >
> > > > Best regards,
> > > >
> > > > The authors

---

> > > > > ### Comment · Reviewer_m8H6 · 2024-11-30
> > > > > **Response to Authors**
> > > > >
> > > > > Thank you for the authors' response. However, I still have some concerns regarding the main issues I previously raised. First, the authors repeatedly emphasized in their reply that CBraMod is capable of handling EEG data with varying numbers of channels during the pretraining phase. If that is the case, why was the pretraining conducted using only the 19 common channels? Furthermore, it would be more convincing if the authors could experimentally demonstrate that CBraMod is not constrained to processing data with only the 19 selected channels during pretraining.

---

> ### Author Response · Authors · 2024-12-01
> **Response to Reviewer m8H6 (8/n)**
>
> Dear Reviewer m8H6,
>
> Thank you very much for your follow-up comments! We appreciate your continued engagement with our work and would like to address the concerns you raised.
>
> In this work, our core contribution lies in the architectural improvements, such as the criss-cross transformer and ACPE, aiming to learn a generalizable EEG representation. The diversity of channel configurations during the pretraining phase is not so important consideration of this work. For pretraining, we selected the TUEG dateset with a fixed set of 19 channels which can provide comprehensive coverage of different brain regions, ensuring sufficient representativeness. This strategy that selecting a fixed number of channels also aligns with the strategies adopted by existing methods such as BENDR [1] and Neuro-GPT [2]. **It is worth noting that, although we selected a fixed set of channels for pretraining,  this does not imply that CBraMod is inherently constrained to processing data with only 19 channels during pretraining.**
>
> To further address your concerns, we conducted additional experiments where CBraMod was pretrained on datasets with varying scales and channel configurations. We then evaluated the pretrained model on entirely new downstream datasets to validate its ability to generalize beyond the specific formats seen during pretraining.
> Given the limited time, we selected a few not-large datasets for pretraining. The pre-training datasets are as follows:
> |Datasets|Channels|Duration|Samples|
> |-|-|-|-|
> |PhysioNet-MI|64|4s|98,37|
> |SHU-MI|32|4s|11,988|
> |Mumtaz2016|19|5s|7,143|
> |SEED-VIG|17|8s|20,355|
>
> We pre-trained CBraMod on these datasets with different channels and fine-tuned CBraMod in FACED, then compared the performance with CBraMod without pre-training. The experimental results are shown as follows:
>
> ||FACED, 9-class|||
> |-|-|-|-|
> |Methods|Balanced Acc.|Coken’s Kappa|Weighted F1|
> |CBraMod (pre-training on the four datasets)|**0.5351** $\pm$ 0.0146|**0.4789** $\pm$ 0.0230|**0.5433** $\pm$ 0.0142|
> |CBraMod (w/o pre-training)|0.5232 $\pm$ 0.0216|0.4615 $\pm$ 0.0289|0.5304 $\pm$ 0.0187|
>
> The results clearly show that CBraMod pretrained on such datasets with different configurations outperforms an untrained (randomly initialized) CBraMod.
> These findings demonstrate that **CBraMod can address varying channel configurations during pre-training and effectively improve the performance of downstream tasks.**
>
> We hope these explanations and additional results address your concerns more comprehensively. Thank you once again for your valuable feedback, which has greatly contributed to improving our work.
>
> Best regards,
>
> The authors.
>
>
>
> [1] Kostas D, Aroca-Ouellette S, Rudzicz F. BENDR: Using transformers and a contrastive self-supervised learning task to learn from massive amounts of EEG data[J]. Frontiers in Human Neuroscience, 2021, 15: 653659.
>
> [2] Cui W, Jeong W, Thölke P, et al. Neuro-GPT: developing a foundation model for EEG[J]. arXiv preprint arXiv:2311.03764, 2023, 107.

---

> > ### Author Response · Authors · 2024-12-02
> > **A new kind reminder to Reviewer m8H6**
> >
> > Dear Reviewer m8H6,
> >
> > Thank you again for your thoughtful and detailed feedback. Your comments have been invaluable in helping us improve and clarify our work.
> >
> > As the discussion phase is nearing its conclusion, we would like to kindly remind you of our response to the concerns you raised regarding CBraMod’s ability to handle varying channel configurations during pre-training. In **Response to Reviewer m8H6 (8/n)**, we conducted a new experiment that demonstrated CBraMod’s capability to process EEG data with different channel configurations during pre-training. These experiments show that **CBraMod is not constrained to specific channel settings in pre-training process**, and the results provide empirical evidence supporting its adaptability and generalization to downstream tasks. We hope these experimental results sufficiently address your concerns.
> >
> > Furthermore, we warmly welcome the reviewer and other researchers to utilize our pre-trained CBraMod in your downstream tasks, as well as to use our publicly released code to pre-train CBraMod on custom pre-training datasets you organize. **We believe that this work will contribute meaningfully to the EEG research community.**
> >
> > We deeply appreciate your engagement with our work and are looking forward to your feedback. Thank you again for your valuable comments.
> >
> > Best regards,
> >
> > The authors

---

> > > ### Comment · Reviewer_m8H6 · 2024-12-03
> > > **Response to Authors**
> > >
> > > Thank you for the authors' response. Although your reply did not fully address my concerns regarding the pretraining setup, I believe that CBraMod still has some valuable aspects. Therefore, after careful consideration, I have decided to raise my score to 5.

---

> ### Author Response · Authors · 2024-12-03
> **Thank you for your constructive feedback!**
>
> Dear Reviewer m8H6,
>
> We sincerely appreciate your thoughtful engagement with our work throughout the review process and for raising your score! **Your feedback has been instrumental in helping us refine our manuscript, and we truly value the time and effort you have dedicated to evaluating our submission!**
>
> We fully understand your concerns regarding the pretraining setup. Although the discussion period was limited, we have still proved that **CBraMod can effectively handle EEG data with varying channel configurations during the pretraining phase with new experiment**. Of course, we recognize that there is still room for further exploration in pretraining data construction, but it is not the key contribution of our current work. Moving forward, we will actively investigate how the choice of pretraining datasets and configurations may impact model performance, ensuring a more comprehensive understanding in future work.
>
> Once again, **thank you for recognizing the contributions of CBraMod and for your constructive feedback, which will guide us as we continue to advance our research!**
>
> Best regards,
>
> The authors

---

### Official Review · Reviewer_1DYx · 2024-11-04

**Soundness:** 4
**Presentation:** 3
**Contribution:** 3
**Rating:** 8
**Confidence:** 4

**Summary:**

The paper introduces CBraMod, a novel EEG foundation model that leverages a criss-cross transformer architecture to capture the spatial and temporal dependencies of EEG signals. It employs an asymmetric conditional positional encoding scheme to adapt to diverse EEG formats and is pre-trained on a large EEG corpus. The model demonstrates state-of-the-art performance across 10 downstream BCI tasks, showcasing its capability and generalizability.

**Strengths:**

The paper proposes a novel transformer-based architecture for EEG feature extraction and uses it as a foundation model. Substantial comparative experiments have been conducted to show its good performance on different EEG prediction tasks. The presentation is clear, and superior results have been achieved. It’s a good exploration of using large-scale models to obtain EEG representations.

**Weaknesses:**

The current version shows a good architecture for feature extraction instead of a real foundation model. It would be beneficial to give a clearer description of the motivation for proposing such a ‘foundation model’ and how this model facilitates the application of EEG analysis and even brain-computer interfaces. Could we use only a few data for finetuning or tuning specific model faster?

**Questions:**

1. What features do the time-domain and frequency-domain branches capture? It would be beneficial to illustrate the specific frequency components learned by the frequency-domain branch within the patch encoder.
2. How does the CNN-based position encoder capture positional information? Does this approach primarily leverage local features, and have you compared it with more traditional positional encoding methods?
3. It would be helpful to provide a comparison of the computation cost of criss-cross attention against other attention mechanisms mentioned in Figure 5.
4. Have you evaluated the impact of data scale for both pretraining and fine-tuning? It would be interesting to know whether the incorporation of a pre-trained foundation model enables effective learning with a reduced amount of data for specific tasks.
5. Many layers are employed, yet the feature maps are not large. Did each layer contribute significantly to the final results?
6. The paper mentions that only 19 channels were used during pre-training, whereas additional channels were introduced in the downstream task. Could you clarify how the balance between channels was managed?

---

> ### Author Response · Authors · 2024-11-20
> **Response to Reviewer 1DYx (1/n)**
>
> We are grateful to the reviewer for the encouraging comments and thoughtful questions.
>
> **W1**: The current version shows a good architecture for feature extraction instead of a real foundation model. It would be beneficial to give a clearer description of the motivation for proposing such a ‘foundation model’ and how this model facilitates the application of EEG analysis and even brain-computer interfaces. Could we use only a few data for finetuning or tuning specific model faster?
>
> **R**: Thank you for your comments and suggestions regarding the motivation for our proposed EEG foundation model and its potential for fine-tuning under limited data conditions. We address your points as follows:
>
> 1. **Motivation for EEG Foundation Models**: The primary goal of EEG foundation models is to **learn generalized representations from large-scale unlabeled EEG data with self-supervised learning**, thereby enhancing the performance of diverse downstream tasks **without relying on handcrafted features, prior knowledge specific to individual tasks or large amounts of labeled data**. Unlike traditional methods that rely on task-specific feature engineering or supervised learning requiring large amounts of labeled data, a foundation model provides a unified framework to streamline EEG analysis and enable applications such as brain-computer interfaces (BCIs).
>
>    We acknowledge that EEG foundation models are still in their early stages compared to large-scale models in NLP and computer vision. This highlights the significant room for exploration and innovation in designing robust, scalable EEG foundation models.
>
> 2. **Architectural Contributions**: To advance this exploration, our architecture incorporates two key contributions:
>    - **Criss-Cross Attention Mechanism**: Designed to effectively leverage the structural properties of raw EEG data, enabling the model to capture complex dependencies across channels and time.
>    - **Asymmetric  Channel Positional Encoding (ACPE)**: A dynamic positional encoding scheme that overcomes the limitations of traditional absolute position encoding tied to electrode numbering, improving the model’s adaptability to diverse referencing schemes and spatial patterns.
>
>    We have refined the revised manuscript to provide a clearer description of these motivations and their implications for EEG analysis. **(Line 81-93, Page 2)**
>
> 3. **Low-Resource Fine-Tuning Experiment**: To address your concern regarding the model’s efficiency in scenarios with limited data, we conducted an additional experiment under low-resource conditions. The results demonstrate that our method significantly outperforms existing approaches when fine-tuned with only a small amount of data. These findings underscore the practical utility of our proposed architecture in real-world applications where data availability is often constrained. **(Line 1494-1524, Page 28-29)**
>
> We hope these clarifications and additional experiments address your concerns and highlight the potential of our model as a foundation for EEG analysis and BCI development. Thank you again for your valuable feedback.
>
> ---
>
> **Q1**: What features do the time-domain and frequency-domain branches capture? It would be beneficial to illustrate the specific frequency components learned by the frequency-domain branch within the patch encoder.
>
> **R**: Thank you for your insightful question regarding the features captured by the time-domain and frequency-domain branches. Below, we provide further clarification:
>
> 1. **Time-Domain Branch**: As described in the original manuscript, the time-domain branch utilizes a convolutional neural network (CNN) to directly process raw signals within each patch. This enables the model to extract temporal features such as signal patterns, trends, and temporal dependencies that are critical for EEG analysis.
>
> 2. **Frequency-Domain Branch**: For the frequency-domain branch, we first apply a Fourier Transform to convert raw signals into the frequency domain. The transformed frequency components are then projected into frequency-domain embeddings through a linear layer. This branch captures spectral characteristics, such as power distribution across frequency bands, which are essential for tasks like abnormal detection and event classification.
>
> 3. **Experimental Validation**: In the appendix section *"Ablation Study on Time-Domain and Frequency-Domain Signals"* **(Line 1440-1463, Page 27-28)**, We provided experimental evidence demonstrating the complementary contributions of these two branches. The results show that Time-Domain and Frequency-Domain Branch both contribute to the performance gains.
>
>
> We hope this response clarifies the distinct roles and contributions of the time-domain and frequency-domain branches in our model. Thank you again for your thoughtful feedback.

---

> > ### Author Response · Authors · 2024-11-20
> > **Response to Reviewer 1DYx (2/n)**
> >
> > **Q2**: How does the CNN-based position encoder capture positional information? Does this approach primarily leverage local features, and have you compared it with more traditional positional encoding methods?
> >
> > **R**: Thank you for raising this question regarding how the CNN-based Asymmetric  Channel Positional Encoder (ACPE) captures positional information and its comparison with traditional positional encoding methods. We provide a detailed explanation below:
> >
> > 1. **Limitations of Traditional Positional Encoding in EEG Models**:  Current EEG foundation models, such as LaBraM, typically use absolute positional encoding based on electrode numbering to represent channel embeddings. However, this approach assumes a fixed correspondence between electrode positions and channel embeddings, which may not hold in practice. EEG channel configuration depends not only on electrode placement but also on referencing schemes (e.g., linked-ear reference, average reference, REST, or bipolar montage). Consequently, such fixed positional encoding schemes may limit the adaptability of EEG foundation models to diverse tasks and setups.
> >
> > 2. **Dynamic Positional Encoding with ACPE**: Inspired by Conditional Positional Encoding methods, we propose ACPE, a CNN-based approach that dynamically encodes the positional relationships between each patch and its surrounding patches. This encoder captures both temporal and spatial positional relationships, allowing the model to adapt better to variations in EEG channel configurations and referencing schemes.
> >
> > 3. **Comparison with Traditional Methods**: In the original manuscript (see *"Positional Encoding Comparison"* section) **(Line 482-504, Page 9-10)**, we compared ACPE with traditional absolute positional encoding methods. Our results demonstrate that ACPE significantly outperforms these methods across a range of downstream tasks, validating its effectiveness and superiority.
> >
> >
> > We hope this explanation clarifies the motivation and advantages of our ACPE approach and its role in improving model adaptability and performance on downstream tasks. Thank you for your thoughtful feedback.
> >
> >
> > [1] Chu X, Tian Z, Zhang B, et al. Conditional Positional Encodings for Vision Transformers[C]//The Eleventh International Conference on Learning Representations.
> >
> > ---
> >
> > **Q3**: It would be helpful to provide a comparison of the computation cost of criss-cross attention against other attention mechanisms mentioned in Figure 5.
> >
> > **R**: Thank you for highlighting the importance of understanding the computational cost of criss-cross attention. In the orignal version of the paper, we have provide a comparison of the computation cost of CBraMod and existing methods in the appendix "**Parameters and FLOPs Comparison**".
> >
> >  In the revised version of the paper, we have included a comparison of computational resources required for criss-cross attention against other attention mechanisms **(Line 1608-1636, Page 30-31)**. This addition provides a clearer view of the efficiency of our approach relative to other methods, demonstrating that criss-cross attention maintains a favorable balance between computational cost and performance gains.
> >
> >
> > We hope this added analysis addresses your concern and offers more insights into the efficiency of criss-cross attention. Thank you again for your valuable feedback.
> >
> > ---
> >
> > **Q4**: Have you evaluated the impact of data scale for both pretraining and fine-tuning? It would be interesting to know whether the incorporation of a pre-trained foundation model enables effective learning with a reduced amount of data for specific tasks.
> >
> > **R**: Thank you for your insightful suggestion regarding the evaluation of data scale. In the revised version of the paper, we have expanded the analysis to include the following two key points:
> >
> > 1. **Impact of Data Scale during Pretraining**: We have evaluated how varying the data scale during the pretraining phase affects the performance of our foundation model **(Line 1360-1391, Page 26)**. Our experiments demonstrate that the model benefits from larger datasets during pretraining, which helps it learn more robust representations, and it also shows promising results when pre-trained on moderate-scale data.
> >
> > 2. **Low-Resource Fine-Tuning for Downstream Tasks**: We further conducted experiments to assess the effectiveness of fine-tuning our pre-trained model with limited data for downstream tasks. The results indicate that the incorporation of a pre-trained foundation model enables effective learning with significantly fewer data, making the model highly data-efficient for specific tasks.
> >    More details can be seen in **Response to W1**.
> >
> > We believe these additions provide further insights into the data efficiency of our model and hope they address your concerns. Thank you again for your valuable feedback.

---

> ### Author Response · Authors · 2024-11-20
> **Response to Reviewer 1DYx (3/n)**
>
> **Q5**: Many layers are employed, yet the feature maps are not large. Did each layer contribute significantly to the final results?
>
> **R**: Thank you for your question regarding the contribution of multiple layers in our model. To address this concern, we have included a performance comparison for varying numbers of Transformer layers with the fixed 200 embedding dimension. The experimental results are as shown as follows:
>
> ||FACED, 9-class|||
> |-|-|-|-|
> |Methods|Balanced Acc.|Coken’s Kappa|Weighted F1|
> |CBraMod (4 layers)|0.5172 $\pm$ 0.0101|0.4601 $\pm$ 0.0158|0.5188 $\pm$ 0.098|
> |CBraMod (8 layers)|0.5341 $\pm$ 0.0097|0.4827 $\pm$ 0.0144|0.5355 $\pm$ 0.0105|
> |CBraMod (12 layers)|**0.5509** $\pm$ 0.0089|**0.5041** $\pm$ 0.0122|**0.5618** $\pm$ 0.0093|
>
> It is evident that increasing the number of model layers leads to a noticeable improvement in performance. We believe that increasing the number of transformer layers could potentially further improve performance. However, due to constraints on time and computational resources, we are currently unable to evaluate whether additional transformer layers would lead to further performance gains.
>
> We hope this additional experiment clarifies the role of the layers in the model's overall performance. Thank you again for your thoughtful feedback.
>
> ---
>
> **Q6**: The paper mentions that only 19 channels were used during pre-training, whereas additional channels were introduced in the downstream task. Could you clarify how the balance between channels was managed?
>
> **R**: Thank you for raising this point regarding the balance between channels during pretraining and downstream tasks. As mentioned in the **Response to Q2**, we leverage the dynamic characteristics of our ACPE (Asymmetric Conditional Positional Encoding) to adapt to the varying number of channels in different downstream tasks. This approach allows our model to effectively handle changes in the number of channels by dynamically learning spatial relationships between the channels, rather than relying on fixed positional encoding schemes. Consequently, the model can seamlessly adjust to the introduction of additional channels in downstream tasks without compromising performance.
>
> We hope this explanation clarifies how the balance between channels is managed. Thank you again for your thoughtful feedback.

---

> > ### Comment · Reviewer_1DYx · 2024-11-27
> > **Response to authors**
> >
> > Thanks for your detailed responses. Most of my concerns have been addressed. Further experiments to interpret the 'generalizable' features learned with the model would benefit the contributions. I'll keep my score.

---

> ### Author Response · Authors · 2024-11-27
> **Thank you for the valuable feedback**
>
> Dear Reviewer 1DYx,
>
> Thank you for your feedback and for maintaining a positive score. We are glad to have addressed most of your concerns.
> In the revised version of our paper, we have added interpretability analysis for the generic EEG representations of our model **(Line 1638-1750, Page 31-33)**.
> We appreciate your suggestion to further explore the interpretation of generalizable features.
>
> Thank you again for your valuable comments.
>
> Best regards,
>
> The authors

---

### Official Review · Reviewer_k8cX · 2024-11-04

**Soundness:** 3
**Presentation:** 3
**Contribution:** 2
**Rating:** 6
**Confidence:** 4

**Summary:**

The paper introduces CBraMod, a novel foundation model designed for EEG signal decoding within Brain-Computer Interface (BCI) applications. CBraMod employs a novel criss-cross transformer architecture to combine modeling of mutual spatial and temporal dependencies inherent to EEG data. Additionally, the model incorporates an Asymmetric Conditional Positional Encoding (ACPE) scheme to learn to adapt to the diverse EEG data formats across various datasets. CBraMod is pre-trained on the extensive Temple University Hospital EEG Corpus (TUH) and evaluated across ten downstream BCI tasks using twelve public datasets. The results demonstrate that CBraMod achieves state-of-the-art performance, highlighting its strong generalizability and effectiveness across diverse EEG decoding tasks.

**Strengths:**

1. The utilization of a criss-cross transformer to independently model spatial and temporal dependencies is a significant advancement. By partitioning attention mechanisms into Spatial-Attention (S-Attention) and TemporalAttention (T-Attention), CBraMod effectively captures the heterogeneous dependencies in EEG signals, which are often overlooked in traditional full EEG modeling strategies.
2. Asymmetric Conditional Positional Encoding (ACPE):  Dynamic Encoding: The ACPE scheme dynamically encodes spatial and temporal
positional information which may enhance the model’s adaptability to various EEG formats.
3. Comprehensive Pre-training. Pre-training CBraMod on the TUH dataset, comprising over 27 000 hours of EEG puts this architecture in line with several other top-performing models.
• Time-Frequency Encoding. Instead of relying on temporal domain convolution the authors decided to explicitly use a tf-decompositon to augment time-domain embeddings

**Weaknesses:**

1. Limited Comparative Analysis with Recent Models. While the paper compares CBraMod with several non-foundation and foundation model baselines, the inclusion of more recent EEG foundation models, e.g. BrainWave (https://arxiv.org/abs/2402.10251) could provide a more comprehensive evaluation of CBraMod’s relative performance.
2. Lack of interpretability. The authors did not provide any kind of interpretation of the obtained models. First of all, it would be very interesting to see what pieces of EEG (source topographies, frequency bands) contribute to the "powerful embeddings"  learnt by the foundation  model. Secondly, for the majority of the downstream tasks there is well defined defined hypothesis regarding the way the decoded information is encoded in the brain. For example motor imagery classification would require information in the 18-14 Hz and 15-25 Hz range on the sources located on the sensory-motor cortex with very distinct topographies. Addition of these would make the paper more convincing.
3. Consistency of embeddings. We would expect that the obtained representation would somehow cluster in a mechanistically meaningful way.  For example, I would expect some clustering of the embeddings  with participant's age, gender, the downstream task, etc.  Demonstrating this consistency would significantly strengthen the presentation.
4. Ablation Studies on Architectural Components. although the paper discusses the contributions of the criss-cross attention mechanism and the ACPE scheme, more granular ablation studies isolating each component’s impact on specific tasks would strengthen the claims regarding their individual effectiveness and justify their inclusion in the architecture.

**Questions:**

1. I think I missed this but how exactly the architecture adapts to each of the dataset? I.e. I an interested in a)  purely technical adaptation due to variation in the number of channels and b) more conceptual one and needed due to the variability of the electro-dynamic properties of the head as a volume-conductor resulting in quite different spatial patterns of EEG activity.
2.  What is the effect of altering the temporal window size (1 s) for the resulting performance? Could it be that different downstream tasks require different values of this important parameter?
3. How important is the time-frequency-based augmentation of the embeddings? What is the performance gain it brings about?
4. Could the authors provide additional details on whether the entire CBraMod model or only specific layers are fine-tuned for downstream tasks?
5. Related to  (4). Could the authors be more explicit regarding the strategies to prevent overfitting during fine-tuning on smaller
downstream datasets (e.g., dropout rates, weight decay specifics beyond pre-training)?

---

> ### Author Response · Authors · 2024-11-20
> **Response to Reviewer k8cX (1/n)**
>
> We thank the reviewer for the positive review and insightful analysis.
>
> ---
> **W1**: Limited Comparative Analysis with Recent Models. While the paper compares CBraMod with several non-foundation and foundation model baselines, the inclusion of more recent EEG foundation models, e.g. BrainWave (https://arxiv.org/abs/2402.10251) could provide a more comprehensive evaluation of CBraMod’s relative performance.
>
> **R**: Thank you for your valuable suggestion regarding the inclusion of recent EEG foundation models such as BrainWave. While we acknowledge the potential relevance of these models to our study, there are two main challenges in incorporating them into our current analysis:
>
> 1. **Availability of Code and Pretrained Weights**: Many of the recent EEG foundation models, have not publicly released their source code (e.g. BrainWave) or pretrained model weights (e.g. EEG2Rep). This significantly limits our ability to directly compare their performance with CBraMod. As a result, we are unable to include them in our experimental evaluation at this time.
>
> 2. **Strength of Current Baselines**: The baseline models included in our study are among the most representative and competitive models currently available in the field of EEG foundation models. These models have demonstrated the state-of-the-art performance on EEG-related tasks, and we believe they provide a robust comparison for assessing the performance of CBraMod.
>
> We hope this clarifies our rationale for the choice of baselines. We are committed to extending our analysis in the future work. Once the code and pretrained weights for newer models like BrainWave become available, we would be eager to incorporate them into our evaluations.
>
> ---
> **W2**: Lack of interpretability. The authors did not provide any kind of interpretation of the obtained models. First of all, it would be very interesting to see what pieces of EEG (source topographies, frequency bands) contribute to the "powerful embeddings" learnt by the foundation model. Secondly, for the majority of the downstream tasks there is well defined defined hypothesis regarding the way the decoded information is encoded in the brain. For example motor imagery classification would require information in the 18-14 Hz and 15-25 Hz range on the sources located on the sensory-motor cortex with very distinct topographies. Addition of these would make the paper more convincing.
>
>
> **R**: Thank you for your insightful comments on the interpretability of our model. We would like to clarify our approach and address your suggestions as follows:
>
>
> 1. **Powerful Representations Learned from Data**: To address the concern regarding the potential benefits of interpretable features like frequency bands, we have conducted new experiments to compare CBraMod on weak processed (only 60hz notch filter and 0.3 hz high-pass filter) with filter-bank-based FBCNet [1] (4 bands: 4~8Hz, 8~14Hz, 15-25Hz, 25-40Hz) on motor imagery tasks (PhysioNet-MI).
>    ||PhysioNet-MI, 4-class|||
>     |-|-|-|-|
>     |Methods|Balanced Acc.|Coken’s Kappa|Weighted F1|
>     |FBCNet| 0.6164 $\pm$ 0.0103|0.4965 $\pm$ 0.0159|0.6212 $\pm$ 0.0113|
>     |CBraMod| 0.6417 $\pm$ 0.0091|0.5222 $\pm$ 0.0169|0.6427 $\pm$ 0.0100|
>
>    The results show that our model on weak processed achieves competitive or superior performance, demonstrating that it effectively learns relevant information even without explicit frequency-based preprocessing. We believe these findings underscore the strength of our model's data-driven design and its applicability across EEG tasks. More details are presented in the revised version of our paper.
>
>
> 2. **Interpretability**: To further address concerns regarding the interpretability, we have added new interpretability analysis in the revised version **(Line 1638-1750, Page 31-33)**, including **(1) Topography Visualization**, **(2) Representation Visualizations on Downstream Datasets** and **(3) Visualization of Patch Relationships from each Criss-cross Transformer Layer**.
>    - **(1) Topography Visualization** demonstrates that CBraMod effectively captures symmetric patterns between the left and right hemispheres from motor imagery data.
>    - **(2) Representation Visualizations on Downstream Datasets** demonstrates that our pre-trained model can learn representations with clustering effects tailored to downstream tasks.
>    - **(3) Visualization of Patch Relationships from each Criss-cross Transformer Layer** confirms that our criss-cross attention effectively models criss-cross dependencies.
>
>
> We hope these clarifications and additional experiments address your concerns about interpretability and model performance. Thank you again for your valuable feedback.
>
> [1] Mane R, Chew E, Chua K, et al. FBCNet: A multi-view convolutional neural network for brain-computer interface[J]. arXiv preprint arXiv:2104.01233, 2021.

---

> > ### Author Response · Authors · 2024-11-20
> > **Response to Reviewer k8cX (2/n)**
> >
> > **W3**: Consistency of embeddings. We would expect that the obtained representation would somehow cluster in a mechanistically meaningful way. For example, I would expect some clustering of the embeddings with participant's age, gender, the downstream task, etc. Demonstrating this consistency would significantly strengthen the presentation.
> >
> > **R**: Thank you for your thoughtful feedback regarding the consistency of embeddings. We fully agree that demonstrating meaningful clustering in the learned representations would strengthen our presentation. To address this:
> >
> > 1. **Representation Visualizations on Downstream Datasets**: As **the response to W2** illustrated, we have visualized the reprenstations of CBraMod in the downstream task test sets. Specifically, we compare the raw EEG sample, representations from the pre-trained CBraMod without fine-tuning, and the fine-tuned CBraMod.
> >
> > 2. **Clustering Effects on Key Attributes**: In particular, our visualizations show that the embeddings derived from the pre-trained model exhibit good clustering by task-relevant attributes (such as task type), demonstrating the enhanced consistency and relevance of the learned representations.
> >
> > We hope this new visualization effectively addresses your concerns about embedding consistency and further illustrates the advantages of the pre-trained CBraMod. Thank you for your valuable insights.
> >
> > ---
> >
> > **W4**: Ablation Studies on Architectural Components. although the paper discusses the contributions of the criss-cross attention mechanism and the ACPE scheme, more granular ablation studies isolating each component’s impact on specific tasks would strengthen the claims regarding their individual effectiveness and justify their inclusion in the architecture.
> >
> > **R**: Thank you for your detailed feedback regarding the ablation studies. We would like to clarify our contribution focus and address your suggestions as follows:
> >
> > 1. **Focus on Key Contributions**: Our primary architectural contributions lie in the criss-cross attention mechanism and the Asymmetric  Channel Positional Encoding (ACPE) scheme. The existing ablation studies in the paper have been designed to illustrate the effectiveness of these components in enhancing model performance, especially on downstream tasks, and we believe they provide a solid justification for their inclusion.
> >
> > 2. **Additional Analysis on Criss-Cross Attention**: To further address your concerns, we have extended our ablation studies with additional experiments that examine the split-head ratio within the criss-cross attention mechanism. The split-head ratio can be regarded as the ratio of spatial attetion and teporal attention, shown as follows:
> >     ||FACED, 9-class|||
> >     |-|-|-|-|
> >     |Methods|Balanced Acc.|Coken’s Kappa|Weighted F1|
> >     |CBraMod (2:6)|0.5405 $\pm$ 0.0096|0.4921 $\pm$ 0.0134|0.5526 $\pm$ 0.0106|
> >     |CBraMod (4:4)|**0.5509** $\pm$ 0.0089|**0.5041** $\pm$ 0.0122|**0.5618** $\pm$ 0.0093|
> >     |CBraMod (6:2)|0.5413 $\pm$ 0.0124|0.4910 $\pm$ 0.0145|0.5531 $\pm$ 0.0128|
> >
> >    The experimental results reveal that spatial attention and temporal attention are equally important.
> >    It provides more granular insights into the impact of criss-cross attention on downstream task performance, strengthening the evidence for its effectiveness. More details are presented in the revised version of our paper. **(Line 1527-1549, Page 29)**
> >
> >
> > We hope these additional experiments address your concerns and reinforce the rationale for our architectural choices. Thank you again for your constructive feedback.

---

> > > ### Author Response · Authors · 2024-11-20
> > > **Response to Reviewer k8cX (3/n)**
> > >
> > > **Q1**: I think I missed this but how exactly the architecture adapts to each of the dataset? I.e. I an interested in a) purely technical adaptation due to variation in the number of channels and b) more conceptual one and needed due to the variability of the electro-dynamic properties of the head as a volume-conductor resulting in quite different spatial patterns of EEG activity.
> > >
> > > **R**: Thank you for your insightful questions regarding the adaptability of our architecture. We address both the technical and conceptual aspects of adaptation as follows:
> > >
> > > 1. **Limitation of existing EEG foundation models on adaptation**: Existing EEG foundation models often leverage Transformers to handle the variable-length nature of EEG signals, providing substantial flexibility. However, while Transformers can process varying input lengths, they are limited in learning spatial relationships, which is essential in processing variation in EEG channels. Existing EEG foundation models, such as LaBraM, employ absolute positional encoding based on electrode numbering as a channel embedding, but this method assumes a fixed relationship between EEG channels and electrode positions. EEG channels are not solely defined by electrode position but are also influenced by the referencing scheme used (e.g., earlobe, average, REST, or bipolar references). As a result, absolute positional encodings tied directly to electrode numbering may limit the model’s adaptability across tasks and datasets that vary in spatial and reference properties.
> > >
> > > 2. **Dynamic Positional Encoding with ACPE**: Motivated by these limitations, we propose ACPE as a more flexible approach to positional encoding. ACPE employs a convolutional network to dynamically encode spatial positional information among patches, allowing the model to capture relative positional information that is robust to differences in reference schemes. This dynamic position learning enhances the model’s adaptability, making it better suited for downstream tasks with varying spatial configurations and reference contexts.
> > >
> > > In our revised manuscript, we have enhanced the discussion of these methods, particularly with respect to their motivation and design **(Line 81-93, Page 2)**. We believe that ACPE offers a meaningful contribution to the design of position encoding in EEG foundation models, as it addresses both the technical and conceptual aspects of EEG signal variability. We hope this explanation clarifies our design motivation and demonstrates the broader applicability of ACPE.
> > >
> > >
> > >
> > > ---
> > >
> > > **Q2**: What is the effect of altering the temporal window size (1 s) for the resulting performance? Could it be that different downstream tasks require different values of this important parameter?
> > >
> > > **R**: Thank you for your question regarding the impact of temporal window size on performance. We would like to clarify the following points:
> > >
> > > 1. **Adaptability to Varying Window Sizes**: Our model is designed to handle any window size greater than or equal to 1 second. This flexibility means that it does not require task-specific adjustments to the window size, enhancing its generalizability across different downstream tasks.
> > >
> > > 2. **Performance Sensitivity to Window Size**: We acknowledge that altering the window size may influence model performance to some extent, as is common with most EEG models. However, this impact is generally consistent across EEG models and is not unique to our approach. Our model achieves competitive performance even with a fixed 1-second window size in SEED-V, demonstrating its capability. **(Line 391-400, Page 8)**
> > >
> > > We hope this clarifies our approach to temporal window size selection. Thank you again for your valuable feedback.
> > >
> > > ---
> > >
> > > **Q3**: How important is the time-frequency-based augmentation of the embeddings? What is the performance gain it brings about?
> > >
> > > **R**: Thank you for your question regarding the importance and performance gain of time-frequency-based augmentation of the embeddings. As shown in our ablation study in Appendix (Ablation Study on Time-Domain and Frequency-Domain Signals) **(Line 1440-1463, Page 27-28)**, the time-frequency combined approach significantly outperforms both time-domain and frequency-domain single signals. Specifically, we found that the performance of the time-frequency combined model was much higher than using just the frequency-domain features, and slightly better than using time-domain features alone.
> > >
> > > This demonstrates the importance of incorporating both temporal and spectral information in EEG data, as EEG signals inherently contain valuable patterns in both domains. Time-frequency augmentation enables CBraMod to learn more comprehensive embeddings, improving its ability to generalize across various tasks and ultimately leading to enhanced performance.

---

> > ### Comment · Reviewer_k8cX · 2024-11-27
> > **I think the paper looks much better now!**
> >
> > Having anticipated the potential of the presented I have given a relatively high score in the beginning, so I will keep it  and thank the authors for the great work they did addressing the comments of the reviewers.

---

> ### Author Response · Authors · 2024-11-20
> **Response to Reviewer k8cX (4/n)**
>
> **Q4**: Could the authors provide additional details on whether the entire CBraMod model or only specific layers are fine-tuned for downstream tasks?
>
> **R**: Thank you for your question regarding the fine-tuning process. Unless otherwise specified, we **fine-tune the entire CBraMod model** during the adaptation to downstream tasks. This approach allows the model to adjust all of its parameters to the specific requirements of each task, ensuring optimal performance. By fine-tuning the entire model, CBraMod is able to leverage the generalizable embeddings learned during pretraining while adapting to the nuances of the target task, leading to better task-specific performance.
>
> ---
>
> **Q5**: Related to (4). Could the authors be more explicit regarding the strategies to prevent overfitting during fine-tuning on smaller downstream datasets (e.g., dropout rates, weight decay specifics beyond pre-training)?
>
> **R**: Thank you for your question regarding strategies to prevent overfitting during fine-tuning. To mitigate overfitting on smaller downstream datasets, we use a dropout rate of 0.1 and a weight decay of 0.05 during the fine-tuning phase. These regularization techniques help prevent the model from overfitting to the limited data available for each specific task. For further details on these hyperparameters and additional settings used during fine-tuning, please refer to the paper **(Line 1055-1069, Page 20)**, where we provide a comprehensive description of the training procedure and hyperparameter choices.
>
> Furthermore, we have added a new experiment on low-resources settings, where we fine-tuned CBraMod only using 30% data of the training set in each downstream dataset.
> The details are presented in the revised version of our paper **(Line 1494-1524, Page 28-29)**. Overall, our model outperforms existing SOTA methods in low-resource scenarios. This empirically demonstrates that our model can effectively mitigate the challenges posed by limited downstream task training data to some extent.

---

> ### Author Response · Authors · 2024-11-27
> **Thank you for the positive feedback**
>
> Dear Reviewer  k8cX,
>
> Thank you for your encouraging feedback and for maintaining a positive score for our work. We sincerely appreciate your thoughtful evaluation and recognition of our efforts in addressing the reviewers' comments. Your support motivates us to continue improving and advancing EEG research.
>
> Best regards,
>
> The authors

---

### Author Response · Authors · 2024-11-20
**Global Response and Revision of the Paper**

We thank the reviewers for their insightful feedback. We appreciate the positive comments on the novelty of CBraMod (1DYx, iYYg), the effectiveness of proposed method (k8cX, m8H6), the comprehensive Pre-training (k8cX, m8H6),  the substantial comparative experiments (1DYx, m8H6, iYYg), the superior results (1DYx, m8H6, iYYg), the good exploration for large-scale models to obtain EEG representations (1DYx, iYYg), the time-frequency Encoding (k8cX), and the clear presentation (1DYx).

We acknowledge some reviewers' concerns regarding the interpretability, the performance of low-resource settings, the choise of datasets, the choise of baseline, the motivation of CBraMod and the value to EEG/BCI field.  **Therefore, we have addressed these concerns in the rebuttal and the revised version of our paper, aiming to alleviate the reviewers' concerns through further comparative experiments, interpretability analysis and discussion.** Due to the extensive computational resources and time required for model pre-training, this is the best effort we can make within the short timeframe provided for the rebuttal.

Here, we will outline the specific revisions made in the revised version of our paper:
1. In response to the suggestions of multiple reviewers, we have enhanced the description of our motivation in the introduction section. **(Line 80-102, Page 2)**
2. As reviewer iYYg suggested, we have added two supervised-learning baselines, EEGNet and EEGConformer, for all the downstream datasets. **(Line 365, Page 7)**
3. As reviewer m8H6 suggested, we have moved the section "Ablation Study on Pre-training" into the main text. **(Line 517, Page 10)**
4. We have added a description in the Conclusion section highlighting the insights our work provides for the BCI field. **(Line 537-539, Page 10)**
5. As reviewer m8H6 suggested, we have added the pre-training loss curve. **(Line 965-988, Page 18-19)**
6. We have added the detailed description for the additional baseline in the appendix. **(Line 1001-1005, Page 19)**
7. As reviewer iYYg suggested, we have added two baselines for sleep staging task, DeepSleepNet[3] and USleep[4]. **(Line 1092, Page 21)**
8. As reviewer iYYg suggested, we have added the experiment setup, CBraMod (excluding TUEV) **(Line 1269, Page 24)** and CBraMod (excluding TUAB) **(Line 1309, Page 25)**, for TUEV and TUAB datasets, respectively.
9. As reviewer iYYg suggested, we have added the experiment on BCIC-IV-2a dataset for motor imagery classification. **(Line 1329-1355, Page 25-26)**
10. As reviewer m8H6 suggested, we have added the section "Scaling Data Size and Model Size" in the appendix. **(Line 1360-1391, Page 26)**
11. In response to the suggestions of multiple reviewers, we have added the section "Low-resource Comparison with Existing Methods" in the appendix. **(Line 1494-1525, Page 28-29)**
12. As reviewer k8cX suggested, we have added a more granular ablation studies on criss-cross splited heads in the appendix. **(Line 1526-1549, Page 29)**
13. As reviewer 1DYx suggested, we have provided a comparison of the computation cost of criss-cross attention against other attention mechanisms in the appendix section "Parameters and FLOPs Comparison". **(Line 1634, Page 31)**
14. In response to the suggestions of multiple reviewers, we have added the new section $\textcolor{red}{\textbf{``Interpretability Analysis''}}$ in the appendix. **(Line 1638-1750, Page 31-33)**
15. In response to the suggestions of multiple reviewers, we have added a section "Implication" to discuss the value of our work on EEG and BCI field.  **(Line 1754-1781, Page 33)**
16. As reviewer iYYg suggested, we have added the performance comparison with EEG-SimpleConv[5] under the leave-one-subject-out (LOSO) protocol on the BCIC-IV-2a datasets.  **(Lines 1811-1835, Pages 34)**
17. In response to the concerns of reviewer m8H6, we have added a section "Comparison on Convergence Speed with Supervised Model" in the appendix to demonstrate that CBraMod achieves significantly faster convergence. **(Line 1837-1862, Page 35)**


We hope that our work contributes valuable insights and inspiration to the EEG and BCI research communities, addressing critical challenges in this field and opening new avenues for future exploration. **We sincerely look forward to the reviewers' feedback and constructive insights to further refine and enhance our study!**

### References

[1] EEGNet: a compact convolutional neural network for EEG-based brain–computer interfaces[J]. JNE, 2018, 15(5): 056013.

[2]  EEG conformer: Convolutional transformer for EEG decoding and visualization[J]. IEEE TNSRE, 2022, 31: 710-719.

[3] DeepSleepNet: A model for automatic sleep stage scoring based on raw single-channel EEG[J]. IEEE TNSRE, 2017, 25(11): 1998-2008.

[4] U-Sleep: resilient high-frequency sleep staging[J]. NPJ digital medicine, 2021, 4(1): 72.

---

> ### Author Response · Authors · 2024-11-25
> **Rebuttal has been submitted and we are eager to hear your further constructive feedback**
>
> Dear Reviewers,
>
> We hope this message finds you well. We greatly appreciate your insightful feedback on our paper. We have made significant revisions to address your concerns, including:
>
>
> - **Motivation of CBraMod**: Clarified the motivation behind our approach and its unique contributions.
> - **Interpretability**: Added new analyses to visualize model representations and dependencies.
> - **Low-Resource Performance**: Demonstrated the model's effectiveness in low-data scenarios.
> - **Dataset Choice**: Conducted additional experiments on new datasets to ensure comprehensive and fair comparison.
> - **Baseline Comparisons**: Included well-established models like EEGNet and EEGConformer.
> - **Value to EEG/BCI Field**: Expanded on the potential impact and long-term value of our work.
>
> We kindly request your prompt review of our rebuttal to finalize the decision process. Your timely response is crucial.
>
> Thank you for your time and consideration.
>
> Best regards,
>
> The authors.

---

### Comment · Reviewer_k8cX · 2024-12-03
**E4 - depression classification**

Dear Authors,

Thanks for the very detailed, intensive and breathtaking rebuttal. I have two questions.

1. Section E.4  where you classify depressed vs. normal subjects. What will happen to the performance of all (or some of) the models when you pre-filter the data in the 0.3-30 Hz  range rather than 0.3-75 Hz as you (and perhaps other authors) did in the mental disorder classification task?

2. While I do appreciate you willingness to demonstrate the superiority of your model over other solutions I feel a bit perplexed. Just purely from the statistical reasons, having performed so MANY additional comparisons you demonstrate that  your model beats all the other solutions, don't you think that there should be some tests where some other models perform better?  May be I missed smth.

---

> ### Author Response · Authors · 2024-12-03
> **Response to Reviewer k8cX**
>
> Dear Reviewer k8cX,
>
> Thank you for your insightful and thought-provoking feedback, which has greatly contributed to refining and contextualizing our work. Below, we provide responses to your new questions.
>
> **Q1:** Section E.4 where you classify depressed vs. normal subjects. What will happen to the performance of all (or some of) the models when you pre-filter the data in the 0.3-30 Hz range rather than 0.3-75 Hz as you (and perhaps other authors) did in the mental disorder classification task?
>
>
> **R1:** We conducted additional experiments as suggested, comparing the performance of EEGConformer, LaBraM, and CBraMod under the 0.3-30 Hz pre-filtering condition on the Mumtaz2016 dataset. The results are summarized below:
>
> ||Mumtaz2016, 2-class|||
> |-|-|-|-|
> |Methods|Balanced Acc.|AUC-PR|AUROC|
> |EEGConformer(0.3-75Hz)|0.9308 $\pm$ 0.0117| 0.9684 $\pm$ 0.0105|0.9702 $\pm$ 0.0101|
> |EEGConformer(0.3-30Hz)|0.9199 $\pm$ 0.0116|0.9668 $\pm$ 0.0108|0.9562 $\pm$ 0.0121|
> |LaBraM(0.3-75Hz)|0.9409 $\pm$ 0.0079|0.9798 $\pm$ 0.0093|0.9782 $\pm$ 0.0057|
> |LaBraM(0.3-30Hz)|0.9280 $\pm$ 0.0087|0.9771 $\pm$ 0.0102|0.9770 $\pm$ 0.0086|
> |CBraMod(0.3-75Hz)|0.9560 $\pm$ 0.0056|0.9923 $\pm$ 0.0032|0.9921 $\pm$ 0.0025|
> |CBraMod(0.3-30Hz)|0.9268 $\pm$ 0.0099|0.9839 $\pm$ 0.0057|0.9826 $\pm$ 0.0065|
>
> As shown, filtering the data to 0.3-30 Hz generally leads to a performance degradation (expecially on balanced accuracy) across all models. This is likely due to the loss of high-frequency information (30-75 Hz), which may still carry relevant features for mental disorder classification tasks.
>
>
> **Q2:** While I do appreciate you willingness to demonstrate the superiority of your model over other solutions I feel a bit perplexed. Just purely from the statistical reasons, having performed so MANY additional comparisons you demonstrate that your model beats all the other solutions, don't you think that there should be some tests where some other models perform better? May be I missed smth.
>
> **R2:** We are delighted that the reviewer has raised such a question, as it suggests that our model’s performance appears to have exceeded expectations and sparked significant interest. However, we must clarify that CBraMod does not consistently outperform all existing methods under all conditions. On a few datasets, our model performs on par with other models.
>
> For example, in the performance evaluation on the TUEV and TUAB datasets excluding potential data leakage (shown in **Response to Reviewer iYYg (7/n)**), our CBraMod achieves competitive performance with state-of-the-art (SOTA) methods but does not always significantly surpass them. Specifically:
>
> ||TUEV, 6-class|||
> |-|-|-|-|
> |Methods|Balanced Acc.|Coken’s Kappa|Weighted F1|
> |LaBraM|0.6616 $\pm$ 0.0170| **0.6745** $\pm$ 0.0195| 0.8329 $\pm$ 0.0086|
> |CBraMod (excluding TUEV)|**0.6659** $\pm$ 0.0124|0.6744 $\pm$ 0.0121|**0.8331** $\pm$ 0.0071|
>
> ||TUAB, 2-class|||
> |-|-|-|-|
> |Methods|Balanced Acc.|AUC-PR|AUROC|
> |LaBraM|**0.8258** $\pm$ 0.0011| 0.9204 $\pm$ 0.0011| **0.9162** $\pm$ 0.0016|
> |CBraMod (excluding TUAB)|0.8249 $\pm$ 0.0025|**0.9221** $\pm$ 0.0015|0.9156 $\pm$ 0.0017|
>
> In these cases, CBraMod performs similarly to LaBraM, with LaBraM even showing slightly better balanced accuracy on TUAB.
>
> Additionally, as noted in the manuscript **(Line 1470-1479, Page 28)**, our CBraMod does not consistently outperform existing EEG foundation model when the pre-trained weights of all the foundation models are completely frozen during fine-tuning on some downstream datasets.
>
> Finally, prompted by your thoughtful question, we wish to acknowledge a specific limitation of CBraMod: as a patch-based EEG foundation model with a patch size of 1 second, it may not be suitable for EEG signals shorter than 1 second in duration. While we do not consider this a significant limitation for an EEG foundation model (this limitation  also exists on LaBraM), we will include this clarification in the limitations section of the manuscript to ensure full transparency.
>
> We greatly appreciate your feedback, which has helped us refine our understanding of CBraMod’s performance and limitations.
>
> Best regards,
>
> The authors

---

> ### Comment · Reviewer_k8cX · 2024-12-03
> **Thanks**
>
> Thanks! Very interesting! The reason I asked to test with a different band is that I was interested in how much EMG activity contributes to decoding. Seems like not much! However your model seems to have the largest drop in performance as compared to the other models which may suggest its greater reliance on the relatively high frequency features to squeeze the high scores. Most of the power in the regular EEG situated above 30-40 Hz comes from the muscles. See the works by Witham et al. (or smth like this). Also, in your response  the depression dataset posits a 2-class problem and not a 9-class as you have copy-pasted) .

---

> > ### Author Response · Authors · 2024-12-03
> > **Thank you for your new feedback!**
> >
> > Dear Reviewer k8cX,
> >
> > Thank you for your thoughtful feedback. We appreciate your insights into the reliance of our model on high-frequency features.
> >
> > We apologize for the "9-class" typo in our previous response, which was due to time constraints. This has now been corrected to reflect the 2-class classification task for the depression dataset.
> >
> > Thank you again for your valuable comments.
> >
> > Best regards,
> >
> > The authors

---

### Meta-Review · Area_Chair_rcC7 · 2024-12-13

**Metareview:**

This works proposes a masked autoencoder strategy for model pretraining on a large collection of EEG data. Evaluation is also quite comprehensive with 10 downstream tasks. Importantly authors engaged with reviewer iYYg who did a very careful check of the reported performances and their comparison with the literature.

The work has convinced that it contributes to the state-of-the-art for the field.

As an AC, I would highly encourage authors to contribute to shared evaluation benchmarks to facilitate tracking the progress of the field on all these tasks that go beyond NLP and computer vision.

**Additional Comments On Reviewer Discussion:**

Authors have engaged with reviewers and led to increased ratings (although m8H6 remains partly convinced). Constructive discussion with reviewer iYYg justify the accept decision.

---

### Decision · Program_Chairs · 2025-01-22

Accept (Poster)